# BRANCHES: A FAST DYNAMIC PROGRAMMING AND BRANCH & BOUND ALGORITHM FOR OPTIMAL DECISION TREES

## ABSTRACT

Decision Tree (DT) Learning is a fundamental problem in Interpretable Machine Learning, yet it poses a formidable optimisation challenge. Despite numerous efforts dating back to the early 1990's, practical algorithms have only recently emerged, primarily leveraging Dynamic Programming (DP) and Branch & Bound (B&B) techniques. These methods fall into two categories: algorithms like DL8.5, MurTree and STreeD utilise an efficient DP strategy but lack effective bounds for pruning the search space; while algorithms like OSDT and GOSDT employ more efficient pruning bounds but at the expense of a less refined DP strategy. We introduce BRANCHES, a new algorithm that combines the strengths of both approaches. Using DP and B&B with a novel analytical bound for efficient pruning, BRANCHES offers both speed and sparsity optimisation. Unlike other methods, it also handles non-binary features. Theoretical analysis shows its lower complexity compared to existing methods, and empirical results confirm that BRANCHES outperforms the state-of-the-art in speed, iterations, and optimality.

## 1 INTRODUCTION

Black-box models are ill-suited for contexts where decisions carry substantial ramifications. In healthcare, for instance, an erroneous negative diagnosis prediction could delay crucial treatment, leading to severe outcomes for patients. Likewise, in the criminal justice system, black-box models may obscure biases associated with factors such as race or gender, potentially resulting in unjust and discriminatory rulings. These considerations underscore the importance of adopting interpretable models in sensitive domains.

Decision Trees (DTs) are valued for their ability to generate simple decision rules from data, making them highly interpretable models. Unfortunately, DT optimization poses a significant challenge due to its NP-completeness, as established by Laurent & Rivest (1976). Consequently, heuristic methods, such as ID3 (Quinlan, 1986), C4.5 (Quinlan, 2014) and CART (Breiman et al., 1984), have been favoured historically. These methods construct DTs greedily by maximising some local purity metric for each chosen split, however, while they are fast and scalable, their greedy nature often leads to suboptimal and overly complex DTs, detracting from their interpretability.

This suboptimality issue spurred researchers into investigating alternatives since the early 1990's, these alternatives are mainly based on *Mathematical Programming*, they range from *Continuous Optimisation* (Bennett & Blue, 1996; Norouzi et al., 2015; Blanquero et al., 2021) to *Mixed Integer Programming (MIP)* (Bertsimas & Dunn, 2017; Verwer & Zhang, 2019; Günlük et al., 2021), *Satisfiability (SAT)* (Bessiere et al., 2009; Narodytska et al., 2018). However, solving these Mathematical Programs scales poorly with large datasets and many features. Moreover, these approaches often fix the DT structure and only optimise the internal splits and leaf predictions, which is significantly less challenging than optimising both accuracy and DT structure (sparsity). Nonetheless, breakthroughs based on Dynamic Programming (DP) and Branch & Bound (B&B) have emerged recently (Hu et al., 2019; Aglin et al., 2020; Lin et al., 2020; Demirović et al., 2022; McTavish et al., 2022; van der Linden et al., 2024) and they provided the first practical algorithms for DT optimisation. These methods fall into two categories, algorithms like DL8.5, MurTree and STreeD operate at the level of the nodes, and consequently have an efficient DP strategy. However, they lack effective

bounds pruning the search space. On the other hand, methods like OSDT and GOSDT operate at the level of DTs, this confers them better pruning bounds but at the expense of a less refined DP strategy.

In this work, we bridge the gap between the two categories. Our new DP and B&B algorithm, BRANCHES, utilises an efficient DP strategy similarly to DL8.5, and employs a novel and more efficient analytical pruning bound than OSDT's and GOSDT's, called Purification Bound. For a comprehensive presentation of our approach, we frame it within a Reinforcement Learning (RL) framework (Sutton & Barto, 2018), capitalizing on its convenient terminology for defining our recursive DP strategy. We analyze BRANCHES's computational complexity and demonstrate its superiority over existing literature. Furthermore, we extensively compare BRANCHES with the state-of-the-art. BRANCHES not only achieves faster optimal convergence in most cases, it also always terminates in fewer iterations, thus validating our theoretical analysis. Our contributions are summarized below:

- We derive a novel analytical bound to prune the search space effectively.
- We develop BRANCHES within a RL framework, its search strategy utilises DP and B&B with our novel pruning bound, called Purification bound.
- BRANCHES is not exclusively applicable to binary features.
- We analyse BRANCHES's computational complexity and show its superiority compared to the complexity bounds derived in the literature.
- We show that BRANCHES outperforms state of the art methods on various real-world datasets with regard to optimal convergence, speed and number of iterations.

## 2 RELATED WORK

To seek optimal DTs, a significant body of literature was devoted to the *Mathematical Programming* approach. We first review these approaches before delving into DP and B&B methods. Early approaches tackled the problem within a *Continuous Optimization* framework. Bennett (1992; 1994); Bennett & Blue (1996) formulated a Multi-Linear Program to optimize a non-linear and non-convex objective function over a polyhedral region. Norouzi et al. (2015) derived a smooth convex-concave upper bound on the empirical loss, which serves as a surrogate objective amenable to minimization via Stochastic Gradient Descent. In a recent development, Blanquero et al. (2021) introduced soft (randomized) decision rules at internal nodes and formulated a Non-Linear Program for which they minimise the expected misclassification cost. However, except for (Bennett & Blue, 1996), the solvers employed by these methods are locally optimal. Furthermore, *Continuous Optimization* lacks the flexibility needed to model univariate Decision Trees (DTs), where each internal split tests only one feature. These DTs are of particular interest because they display better interpretability than multi-variate DTs. To address this limitation, a *Mixed Integer Programming (MIP)* framework was rather considered in a multitude of research papers (Bertsimas & Dunn, 2017; Verwer & Zhang, 2017; 2019; Zhu et al., 2020; Günlük et al., 2021), and alternatively, some studies have explored the *Satisfiability (SAT)* framework (Bessiere et al., 2009; Narodytska et al., 2018; Avellaneda, 2020). Despite the rich literature of *Mathematical Programming* approaches, they suffer from serious limitations. The number of variables involved in the Mathematical Programs increases with the size of the dataset and the number of features, slowing down the solvers and severely limiting scalability. In addition, these methods often fix a DT structure a priori and only optimise its internal splits and leaf predictions. While this simplifies the problem, it misses the true optimal DT unless the optimal structure has been fixed in advance, which is highly unlikely. And finally, *SAT* methods seek DTs that perfectly classify the dataset, as such, they are especially prone to overtraining.

In the last five years, DP and B&B offered the first practical algorithms for Optimal DTs, and as such, triggered a paradigm shift from *Mathematical Programming*. The first of these algorithms is OSDT (Hu et al., 2019), it seeks to minimise a regularised misclassification error objective with a penalty on the number of leaves. To achieve this, OSDT employs a series of analytical bounds to prune the space of DTs (its search space). OSDT was followed shortly after by GOSDT (Lin et al., 2020) to generalise the approach to other objective functions. In contrast, DL8.5 (Aglin et al., 2020) is a fundamentally different approach, it is based on ideas from the earlier DL8 algorithm (Nijssen & Fromont, 2007; 2010). DL8 operates on a lattice of itemsets as its search space, from which it mines the optimal DT, this is fundamentally distinct from the search space of DTs employed by OSDT and

GOSDT. However, DL8 is a purely DP algorithm, and as such it is computationally and memory costly. DL8.5 addressed this issue by incorporating B&B to DL8, which offers higher speed and better scalability, albeit without addressing sparsity (DL8.5 fixes a maximum depth but does not actively minimise the depth or the number of leaves). Additionally, DL8.5's B&B strategy is based on the best solution found so far rather than more sophisticated analytical bounds, hindering its pruning capacity of the search space. Meanwhile, OSDT and GOSDT solve for sparsity but are comparatively slower due to their less refined DP strategy. Our work is motivated by this landscape, aiming to leverage the speed and scalability of methods like DL8.5 while addressing sparsity concerns and improving on the pruning efficiency of OSDT and GOSDT.

Additional recent advancements in the field include MurTree (Demirović et al., 2022), which enhances DL8.5 with similarity bounds and a tailored method for handling DTs of depth 2. McTavish et al. (2022) introduce a guessing strategy to navigate the search space, seeking solutions with performance akin to a reference ensemble model. van der Linden et al. (2024) investigate separable objectives and constraints and introduce a generalised DP framework called STreeD.

## 3 PROBLEM FORMULATION

We consider classification problems with categorical features $X = \left(X^{(1)}, \ldots, X^{(q)}\right)$ and class variable $Y \in \{1, \ldots, K\}$ such that:

$$\forall i \in \{1, \ldots q\} : X^{(i)} \in \{1, \ldots, C_i\}, \ C_i \geq 2$$

where $q \geq 2, K \geq 2$. We are provided with a dataset $\mathcal{D} = \{(X_m, Y_m)\}_{m=1}^n$ of $n \geq 1$ examples. In the following sections, we define the notions of branches and sub-DTs that are key to our formulation.

### 3.1 BRANCHES

A branch $l$ is a conjunction of clauses on the features of the following form:

$$l = \bigwedge_{v=1}^{\mathcal{S}(l)} \mathbb{1}\left\{X^{(i_v)} = j_v\right\}$$

such that $\forall v \in \{1, \ldots, \mathcal{S}(l)\} : i_v \in \{1, \ldots, q\}, \ j_v \in \{1, \ldots, C_{i_v}\}$ and:

$$\forall v, v' \in \{1, \ldots \mathcal{S}(l)\} : v \neq v' \implies i_v \neq i_{v'}$$

This condition ensures that no feature is used in more than one clause within $l$. We refer to these clauses as rules or splits. $\mathcal{S}(l)$ is the number of splits in $l$.

For any datum $X$, the valuation of $l$ for $X$ is denoted $l(X) \in \{0, 1\}$ and defined as follows:

$$l(X) = 1 \iff \bigwedge_{v=1}^{\mathcal{S}(l)} \mathbb{1}\left\{X^{(i_v)} = j_v\right\} = 1$$

When $l(X) = 1$, we say that $X$ is in $l$ or that $l$ contains $X$. The branch containing all possible data is called the root and denoted $\Omega$. Since the valuation of $l$ for any datum remains invariant when reordering its splits, we represent $l$ uniquely by ordering its splits from the smallest feature index to the highest, i.e. we impose $1 \leq i_1 < \ldots < i_{\mathcal{S}(l)} \leq q$. This unique representation is at the core of our DP memoisation.

In the following, we define the notion of splitting a branch. Let $i \in \{1, \ldots, q\} \setminus \{i_1, \ldots, i_{\mathcal{S}(l)}\}$ be an unused feature in the splits of $l$. We define the children of $l$ that stem from splitting $l$ with respect to $i$ as the set $\text{Ch}(l, i) = \{l_1, \ldots, l_{C_i}\}$ where:

$$\forall j \in \{1, \ldots, C_i\} : l_j = l \wedge \mathbb{1}\left\{X^{(i)} = j\right\} \tag{1}$$

The dataset $\mathcal{D}$ provides an empirical distribution of the data. The probability that a datum is in $l$ is:

$$\mathbb{P}[l(X) = 1] = \frac{n(l)}{n}$$

where $n(l) = \sum_{m=1}^{n} l(X_m)$ is the number of data in $l$. Likewise, we want to define the probability that a datum is in $l$ and correctly classified. For this purpose, we define the predicted class in $l$ as:

$$k^*(l) = \text{Argmax}_{1 \le k \le K} \{n_k(l)\}$$

Where $n_k(l) = \sum_{m=1}^{n} l(X_m) \mathbb{1}\{Y_m = k\}$ is the number data in $l$ that are of class $k$. $k^*(l)$ is the majority class in $l$. Then the probability that a datum is in $l$ and correctly classified is:

$$\mathcal{H}(l) = \mathbb{P}[l(X) = 1, Y = k^*(l)] = \frac{n_{k^*(l)}(l)}{n} \tag{2}$$

### 3.2 DECISION TREES

Let $l$ be a branch, a sub-DT rooted in $l$ is a collection of branches $T = \{l_1, \ldots, l_{|T|}\}$ that stem from a series of successive splittings of $l$, its children and so on. $T$ partitions $l$ in the following sense:

$$\begin{cases} l = \bigvee_{u=1}^{|T|} l_u \\ \forall u, u' \in \{1, \ldots, |T|\} : u \ne u' \implies l_u \wedge l_{u'} = 0 \end{cases}$$

We denote $\mathcal{S}(T)$ the number of splitting steps it took to construct $T$ from $l$. For any datum $X$ in $l$, $T$ predicts the majority class of the branch $l_u$ containing $X$:

$$T(X) = \sum_{u=1}^{|T|} l_u(X) k^*(l_u)$$

Now we can define the proportion of data in $\mathcal{D}$ that is in $l$ and is correctly classified by $T$:

$$\mathcal{H}(T) = \mathbb{P}[l(X) = 1, T(X) = Y] = \sum_{u=1}^{|T|} \mathcal{H}(l_u)$$

The additivity property is due to $\{l_1, \ldots, l_{|T|}\}$ forming a partition of $l$. A DT is a sub-DT that is rooted in $\Omega$. Let $T$ be a DT, since $\Omega(X) = 1$ for any datum $X$, then:

$$\mathcal{H}(T) = \mathbb{P}[\Omega(X) = 1, T(X) = Y] = \mathbb{P}[T(X) = Y]$$

which is the accuracy of $T$. Maximising accuracy is not a suitable objective, it overlooks sparsity. To incorporate sparsity, we rather consider the following regularised objective:

$$\mathcal{H}_\lambda(T) = -\lambda \mathcal{S}(T) + \mathcal{H}(T) \tag{3}$$

with $\lambda \in [0, 1]$ a penalty parameter penalising DTs with too many splits. This objective is employed by CART during the pruning phase, it was also considered by Bertsimas & Dunn (2017) and recently by Chaouki et al. (2024). Hu et al. (2019); Lin et al. (2020) use a slightly different version, where the total number of leaves is penalised instead.

### 3.3 MARKOV DECISION PROCESS (MDP)

To frame the problem within a Reinforcement Learning framework, we define the following MDP.

**State space:** The set of all possible sub-DTs. A state with only one branch $T = \{l\}$ is called a unit-state. To make the notation lighter, we just denote it $l$. There are special types of states called absorbing states. A state is absorbing if all actions transition back to it and yield 0 reward. The initial state is always the root $\Omega$.

**Action space:** At every state $T$, we denote $\mathcal{A}(T)$ the set of permissible actions at $T$. We first define this set of actions for unit-states, then we generalise it to all states. Let $l = \bigwedge_{v=1}^{\mathcal{S}(l)} \mathbb{1}\{X^{(i_v)} = j_v\}$ be a unit-state, there are two types of actions:

- The terminal action $\bar{a}$. It transitions from $l$ to an absorbing state $\bar{l}$. We denote $l \xrightarrow{\bar{a}} \bar{l}$.
- Split actions. The set of possible split actions at $l$ is $\{1, \ldots, q\} \setminus \{i_1, \ldots, i_{\mathcal{S}(l)}\}$ the set of unused features by $l$. Let $i$ be a split action, taking $i$ transitions $l$ to state $T = \text{Ch}(l, i)$, defined in Eq. (1). We denote the transition with $l \xrightarrow{i} T = \text{Ch}(l, i)$.

Thus $\mathcal{A}(l) = \{\overline{a}\} \cup \{1, \ldots, q\} \setminus \{i_1, \ldots, i_{\mathcal{S}(l)}\}$. When $\mathcal{S}(l) = q$, then $\mathcal{A}(l) = \{\overline{a}\}$ and we can only transition to $\overline{l}$. We can now generalise the set of permissible actions to any state $T = \{l_1, \ldots l_{|T|}\}$ as $\mathcal{A}(T) = \mathcal{A}(l_1) \times \ldots \times \mathcal{A}(l_{|T|})$. Taking action $a = (a_1, \ldots, a_{|T|}) \in \mathcal{A}(T)$ in $T$ is equivalent to taking each action $a_u$ in $l_u$ for $1 \leq u \leq |T|$, thus performing the transition:

$$T \xrightarrow{a} T' = \bigcup_{u=1}^{|T|} T'_u, \ \ \forall u \in \{1, \ldots, |T|\} : l_u \xrightarrow{a_u} T'_u$$

**Reward function:** For any state $T$ and action $a \in \mathcal{A}(T)$, $r(T, a)$ is the reward of taking action $a$ in $T$. Similarly to the definition of the actions, we first define the reward for unit-states and then we generalise it to all states. Let $l$ be a unit-state and $a \in \mathcal{A}(l)$ then we have:

- If $a$ is a split action, then $r(l, a) = -\lambda$ regardless of $l$ (except if $l$ is absorbing).

- If $a = \overline{a}$, then $r(l, \overline{a}) = \mathcal{H}(l)$ as per Eq. (2).

- If $l = \overline{l}$, i.e. $l$ is an absorbing state, then $r(\overline{l}, a) = 0$.

For any state $T = \{l_1, \ldots, l_{|T|}\}$ and action $a = (a_1, \ldots, a_{|T|}) \in \mathcal{A}(T)$, we define the reward as:

$$r(T, a) = \sum_{u=1}^{|T|} r(l_u, a_u)$$

A policy $\pi$ maps each state $T$ to one of its actions $\pi(T) \in \mathcal{A}(T)$. From a state $T$, the return of policy $\pi$ is defined as the cumulative reward of following $\pi$ starting from $T$:

$$\mathcal{R}^{\pi}(T) = \sum_{t=0}^{\infty} r(T_t, \pi(T_t))$$

where $T_0 = T$ and $\forall t \geq 0 : T_t \xrightarrow{\pi(T_t)} T_{t+1}$. Each policy is evaluated by its return from the initial state $\Omega$, our objective is to find the optimal policy as we shall justify shortly. First, we need to ensure that there are no divergence issues related to the infinite sum in the definition of the return.

**Proposition 1.** *Let $\pi$ be a policy, $l$ a unit-state and consider $T_0 = l$ and $\forall t \geq 0 : T_t \xrightarrow{\pi(T_t)} T_{t+1}$. Then, there exists $\tau \geq 0$ such that for any $t \geq \tau$, $T_t = \{\overline{l_1}, \ldots, \overline{l_{|T_\tau|}}\}$ is an absorbing state. In which case we call $T_l^{\pi} = \{l_1, \ldots, l_{|T_\tau|}\}$ the sub-DT of $\pi$ rooted in $l$. If $l = \Omega$ we abbreviate the notation $T_{\Omega}^{\pi} \equiv T^{\pi}$ and call $T^{\pi}$ the DT of $\pi$.*

Proposition 1 states that all policies eventually arrive in an absorbing state after a finite number of steps, regardless of where they start. Therefore all policies have finite returns. Now let us justify why we seek the optimal policy.

**Proposition 2.** *For any policy $\pi$ and unit-state $l$, the return of $\pi$ from $l$ satisfies:*

$$\mathcal{R}^{\pi}(l) = \mathcal{H}_{\lambda}(T_l^{\pi}) = -\lambda \mathcal{S}(T_l^{\pi}) + \mathcal{H}(T_l^{\pi})$$

*In particular $\mathcal{R}^{\pi}(\Omega) = \mathcal{H}_{\lambda}(T^{\pi}) = -\lambda \mathcal{S}(T^{\pi}) + \mathcal{H}(T^{\pi})$.*

Proposition 2 links the return of a policy to the regularised accuracy of its sub-DT. On the other hand, since any DT $T$ is constructed with successive splittings starting from $\Omega$, there always exists a policy $\pi$ such that $T^{\pi} = T$, and therefore $\mathcal{R}^{\pi}(\Omega) = \mathcal{H}_{\lambda}(T)$. This result provides the equivalence between finding the optimal DT and the optimal policy:

$$T^* = \text{Argmax}_T \{\mathcal{H}_{\lambda}(T)\}, \ \pi^* = \text{Argmax}_{\pi} \{\mathcal{R}^{\pi}(\Omega)\}$$

in which case the optimal DT is the DT of $\pi^*$, i.e. $T^* = T^{\pi^*}$. To conclude this section, our objective is now is to find $\pi^*$ and then deduce $T^*$ as $T^{\pi^*}$. We abbreviate the notation $\mathcal{R}^{\pi^*} \equiv \mathcal{R}^*$.

## 4    THE ALGORITHM: BRANCHES

BRANCHES is a Value Iteration algorithm (Sutton & Barto, 2018) that is enhanced with a structured B&B search. To describe this algorithm, it is convenient to further introduce the state-action return quantity. For any policy $\pi$, state $T$ and action $a \in \mathcal{A}(T)$, let $T \xrightarrow{a} T_1$ and $\forall t \geq 1 : T_t \xrightarrow{\pi(T_t)} T_{t+1}$. Then the state-action return $\mathcal{Q}^\pi(T, a)$ is the cumulative reward of taking action $a$ first, then following $\pi$:

$$\mathcal{Q}^\pi(T, a) = r(T, a) + \sum_{t=1}^\infty r(T_t, \pi(T_t)) = r(T, a) + \mathcal{R}^\pi(T_1)$$

Given the optimal state-action returns $\mathcal{Q}^*(T, a) = \mathcal{Q}^{\pi^*}(T, a)$, we can deduce the optimal policy:

$$\pi^*(T) = \text{Argmax}_{a \in \mathcal{A}(T)} \mathcal{Q}^*(T, a)$$

In the next section, we show how BRANCHES estimates these optimal state-action returns.

### 4.1    ESTIMATING THE OPTIMAL STATE-ACTION RETURNS $\mathcal{Q}^*(l, a)$

Let $l$ be a non-absorbing unit-state and $a \in \mathcal{A}(l)$, we denote $\mathcal{Q}(l, a)$ the estimate of $\mathcal{Q}^*(l, a)$. For the terminal action $\overline{a}$, $\mathcal{Q}^*(l, \overline{a})$ is directly accessible from the data via:

$$\mathcal{Q}(l, \overline{a}) = \mathcal{Q}^*(l, \overline{a}) = r(l, \overline{a}) = \mathcal{H}(l) = \frac{n_{k^*(l)}(l)}{n} \tag{4}$$

For a split action $a \in \mathcal{A}(l) \setminus \{\overline{a}\}$, such that $l \xrightarrow{a} T = \{l_1, \ldots, l_{|T|}\}$, $\mathcal{Q}(l, a)$ is defined according to the Bellman Optimality Equations below.

**Proposition 3.** *Let $l$ be a non-absorbing unit-state, $a \in \mathcal{A}(l) \setminus \{\overline{a}\}$ a split action such that $l \xrightarrow{a} T = \{l_1, \ldots, l_{|T|}\}$. Then we have:*

$$\mathcal{Q}^*(l, a) = -\lambda + \mathcal{R}^*(T) = -\lambda + \sum_{u=1}^{|T|} \mathcal{R}^*(l_u)$$

$$\forall u \in \{1, \ldots, |T|\} : \mathcal{R}^*(l_u) = \mathcal{Q}^*(l_u, \pi^*(l_u)) = \max_{a \in \mathcal{A}(l_u)} \mathcal{Q}^*(l_u, a)$$

Proposition 3 suggests the following recursive definitions of the estimates:

$$\mathcal{Q}(l, a) = -\lambda + \sum_{u=1}^{|T|} \mathcal{R}(l_u) \tag{5}$$

$$\forall u \in \{1, \ldots, |T|\} : \mathcal{R}(l_u) = \max_{a \in \mathcal{A}(l_u)} \mathcal{Q}(l_u, a) \tag{6}$$

The estimate $\mathcal{Q}(l, a)$ in Eq. (5) can only be calculated if the estimates $\mathcal{R}(l_u)$ in Eq. (6) are available. Otherwise we initialise $\mathcal{Q}(l, a)$ with Eq. (7) according to Proposition 4.

**Proposition 4** (Purification Bound). *For any non-absorbing unit-state $l$ and split action $a \in \mathcal{A}(l) \setminus \{\overline{a}\}$, we define the Purification Bound estimates:*

$$\mathcal{Q}(l, a) = -\lambda + \mathbb{P}[l(X) = 1] = -\lambda + \frac{n(l)}{n} \tag{7}$$

$$\mathcal{R}(l) = \max\{\mathcal{H}(l), -\lambda + \mathbb{P}[l(X) = 1]\} = \max\left\{\frac{n_{k^*(l)}(l)}{n}, -\lambda + \frac{n(l)}{n}\right\} \tag{8}$$

*Then the estimates $\mathcal{Q}(l, a)$ and $\mathcal{R}(l)$ are upper bounds on $\mathcal{Q}^*(l, a)$ and $\mathcal{R}^*(l)$ respectively.*

In the following, we provide an intuition behind the Purification Bound. If the split action $a$ yields $l \xrightarrow{a} T = \{l_1, \ldots, l_{|T|}\}$ such that all the data in the resulting children branches $l_u$ are correctly classified (in which case, the branches $l_u$ are called pure), then:

$$\mathcal{Q}^*(l, a) = -\lambda + \mathcal{H}(T) = -\lambda + \mathbb{P}[T(X) = Y, l(X) = 1] = -\lambda + \mathbb{P}[l(X) = 1]$$

Thus the bound Eq. (7) coincides exactly with the optimal state-action value of an action that *purifies* $l$ (if it exists), hence the name *Purification Bound*.

Now we can straightforwardly define $\mathcal{Q}(T, a)$ for any state $T$. Consider a state $T = \{l_1, \ldots, l_{|T|}\}$ and an action $a = (a_1, \ldots, a_{|T|}) \in \mathcal{A}(T)$ such that:

$$\begin{cases} T \xrightarrow{a} T' = \bigcup_{u=1}^{|T|} T'_u \\ \forall u \in \{1, \ldots, |T|\} : l_u \xrightarrow{a_u} T'_u \end{cases}$$

Then we have the following:

$$\mathcal{Q}^*(T, a) = r(T, a) + \mathcal{R}^*(T')$$
$$= \sum_{u=1}^{|T|} r(l_u, a_u) + \sum_{u=1}^{|T|} \mathcal{R}^*(T'_u) = \sum_{u=1}^{|T|} (r(l_u, a_u) + \mathcal{R}^*(T'_u)) = \sum_{u=1}^{|T|} \mathcal{Q}^*(l_u, a_u)$$

Therefore, this suggests defining the estimate $\mathcal{Q}(T, a)$ directly with:

$$\mathcal{Q}(T, a) = \sum_{u=1}^{|T|} \mathcal{Q}(l_u, a_u) \tag{9}$$

**Summary:** For any unit-state $l$, the estimate $\mathcal{Q}(l, \overline{a})$ for the terminal action $\overline{a}$ is known in advance and calculated with Eq. (4). For any split action $a \in \mathcal{A}(l) \setminus \{\overline{a}\}$, $\mathcal{Q}(l, a)$ is calculated with Eq. (5) when estimates for the children are available, otherwise it is initialised with Eq. (7). For a general state $T$, the estimate is deduced straightforwardly via Eq. (9).

### 4.2 THE SEARCH STRATEGY

Initially, all the non-terminal unit-states $l$ are labelled as unvisited and incomplete, which means that $\mathcal{R}^*(l)$ are still unknown. The absorbing states are labelled as complete on the other hand. Moreover, the state-action pairs $(l, a)$ (for non-absorbing unit-states $l$) are also labelled as incomplete since we do not know $\mathcal{Q}^*(l, a)$ either. We initialise an empty memo where the encountered state values estimates $\mathcal{R}(l)$ are stored. Each iteration of BRANCHES follows the Value Iteration pipeline below:

- **Selection:** Initialise an empty list *path*. Starting from the root $l = \Omega$, choose the action maximising the optimal state-action value estimate:

$$a = \text{Argmax}_{a' \in \mathcal{A}(l)} \mathcal{Q}(l, a')$$

Append $(l, a)$ to *path* and transition $l \xrightarrow{a} T = \{l_1, \ldots, l_{|T|}\}$. Choose an incomplete unit-state $l_u \in T$ and make it the current state $l = l_u$, *this choice can be arbitrary or according to some heuristic*. Repeat this process until reaching an unvisited or absorbing unit-state $l$. Note that the *path* list does not include this final state $l$.

- **Expansion:** If $l$ is absorbing, then we move to the Backpropagation step. Otherwise we estimate $\mathcal{Q}(l, a)$ for all $a \in \mathcal{A}(l)$ as explained below.
  For the terminal action, we set $\mathcal{Q}(l, \overline{a}) = \mathcal{H}(l)$ as per Eq. (4) and we label $(l, \overline{a})$ as complete. For any split action $a \in \mathcal{A}(l) \setminus \{\overline{a}\}$, let $l \xrightarrow{a} T = \{l_1, \ldots, l_{|T|}\}$. We calculate $\mathcal{Q}(l, a)$ according to Eq. (5):

$$\mathcal{Q}(l, a) = -\lambda + \sum_{u=1}^{|T|} \mathcal{R}(l_u)$$

where for each $l_u \in T$, $\mathcal{R}(l_u)$ is retrieved from the memo in case $l_u$ is labelled as visited, otherwise $\mathcal{R}(l_u)$ is initialised with Eq. (8):

$$\mathcal{R}(l_u) = \max\left\{\frac{n_{k^*(l_u)}(l_u)}{n}, -\lambda + \frac{n(l_u)}{n}\right\}$$

Table 1: Comparing the complexity bounds of BRANCHES and OSDT.

| $\lambda$ | $q = 10$ | | $q = 15$ | | $q = 20$ | |
|---|---|---|---|---|---|---|
| | BRANCHES | OSDT | BRANCHES | OSDT | BRANCHES | OSDT |
| 0.1 | $\mathbf{5.70 \times 10^4}$ | $5.61 \times 10^{13}$ | $\mathbf{5.80 \times 10^5}$ | $6.86 \times 10^{16}$ | $\mathbf{2.82 \times 10^6}$ | $8.35 \times 10^{18}$ |
| 0.05 | $\mathbf{3.94 \times 10^5}$ | $7.52 \times 10^{271}$ | $\mathbf{7.53 \times 10^7}$ | $1.53 \times 10^{473}$ | $\mathbf{3.01 \times 10^9}$ | $5.69 \times 10^{576}$ |
| 0.01 | $\mathbf{3.94 \times 10^5}$ | $1.64 \times 10^{392}$ | $\mathbf{1.43 \times 10^8}$ | INF | $\mathbf{4.65 \times 10^{10}}$ | INF |

and we store $\mathcal{R}(l_u)$ in the memo. If all children $l_u$ are complete, then we label $(l, a)$ as complete and we have $\mathcal{Q}(l, a) = \mathcal{Q}^*(l, a)$. Once we have calculated $\mathcal{Q}(l, a)$ for all actions $a \in \mathcal{A}(l)$, we deduce the state value estimate of $l$ as follows:

$$\mathcal{R}(l) = \max_{a \in \mathcal{A}(l)} \mathcal{Q}(l, a)$$

If $a^* = \mathrm{Argmax}_{a \in \mathcal{A}(l)} \mathcal{Q}(l, a)$ is such that $(l, a^*)$ is complete, then we label $l$ as complete and we have $\mathcal{R}^*(l) = \mathcal{R}(l) = \mathcal{Q}(l, a^*) = \mathcal{Q}^*(l, a^*)$.

- **Backpropagation:** Update $\mathcal{Q}(l, a)$ and $\mathcal{R}(l)$ for all $(l, a)$ in *path* via Backward recursion. For $j = length(path) - 1, \ldots, 0$, let $(l, a) = path[j]$ with $l \xrightarrow{a} T = \{l_1, \ldots, l_{|T|}\}$, then we update $\mathcal{Q}(l, a)$ and $\mathcal{R}(l)$ with Eq. (5) and Eq. (6) respectively. We update $\mathcal{R}(l)$ in the memo. If all children $l_u$ are complete, we label $(l, a)$ as complete. If $a^* = \mathrm{Argmax}_{a \in \mathcal{A}(l)} \mathcal{Q}(l, a)$ is such that $(l, a^*)$ is complete, then we label $l$ as complete.

BRANCHES terminates when the root $\Omega$ is complete. Algorithm 1 in Appendix E summarises these steps in a pseudocode, and Appendix D provides a detailed implementation description.

## 5 THEORETICAL ANALYSIS

In this section, we prove the optimality of BRANCHES in Theorem 5 and we analyse its computational complexity in Theorem 6 and Corollary 7.

**Theorem 5** (Optimality of BRANCHES). *When BRANCHES terminates, the optimal policy is the greedy policy with respect to the estimated state-action values $\mathcal{Q}(l, a)$, which means that for any state $T$:*

$$\pi^*(T) = \mathrm{Argmax}_{a \in \mathcal{A}(T)} \mathcal{Q}(T, a)$$

To the best of our knowledge, Hu et al. (2019) are the only authors providing a complexity analysis of their algorithm in the DP and B&B literature of optimal DTs. (Hu et al., 2019, Theorem E.2) derives an upper bound on the total number of DT evaluations performed by OSDT. There is an inaccuracy in the result, the sum should be up to the maximum depth of the optimal DT rather than the maximum number of its leaves. We provide a corrected version and discuss it in Theorem 9. To compare the computational complexities of OSDT and BRANCHES, we analyse the number of branch evaluations, i.e. calculations of $\mathcal{H}(l)$, performed by BRANCHES to reach termination.

**Theorem 6** (Problem-dependent complexity of BRANCHES). *Let $\Gamma(q, C, \lambda)$ denote the total number of branch evaluations performed by BRANCHES for an instance of the classification problem with $q \geq 2$ features, $0 < \lambda \leq 1$ the penalty parameter, and $C \geq 2$ the number of categories per feature. Then, $\Gamma(q, C, \lambda)$ satisfies the following bound:*

$$\Gamma(q, C, \lambda) \leq \sum_{h=0}^{\kappa} (q - h) C^{h+1} \binom{q}{h}; \quad \kappa = \min \left\{ \left\lfloor \mathcal{S}(T^*) - 1 + \frac{1 - \mathcal{H}(T^*)}{\lambda} \right\rfloor, q \right\}$$

**Corollary 7** (Problem-independent complexity of BRANCHES). *Let $\Gamma(q, \lambda, C)$ be defined as in Theorem 6, then it satisfies:*

$$\Gamma(q, C, \lambda) \leq \sum_{h=0}^{\kappa} (q - h) C^{h+1} \binom{q}{h}; \quad \kappa = \min \left\{ \left\lfloor \frac{1}{K\lambda} \right\rfloor - 1, q \right\}$$

Table 2: Comparing BRANCHES with OSDT, PyGOSDT and GOSDT; objective here refers to the regularised objective $\mathcal{H}_\lambda$. $TO$ refers to timeout.

| Dataset | OSDT | | | PyGOSDT | | | GOSDT | | | BRANCHES | | |
|---|---|---|---|---|---|---|---|---|---|---|---|---|
| | objective | time (s) | iterations | objective | time (s) | iterations | objective | time (s) | iterations | objective | time (s) | iterations |
| monk1-l | **0.93** | 71 | 2e6 | **0.93** | 181 | 3e6 | **0.93** | 0.71 | 3e4 | **0.93** | 0.11 | **617** |
| monk1-f | 0.97 | $TO$ | 2e4 | 0.97 | $TO$ | 2e3 | **0.983** | 4.02 | 9e4 | **0.983** | 1.07 | **1e4** |
| monk1-o | — | — | — | — | — | — | — | — | — | **0.9** | 0.02 | **64** |
| monk2-l | 0.95 | $TO$ | 7e4 | 0.95 | $TO$ | 400 | **0.968** | 10 | 1e5 | **0.968** | 2.7 | **3e4** |
| monk2-f | 0.90 | $TO$ | 4e4 | 0.90 | $TO$ | 3e4 | **0.933** | 11.1 | 1e5 | **0.933** | 5.29 | **7e4** |
| monk2-o | — | — | — | — | — | — | — | — | — | **0.955** | 0.10 | **1e3** |
| monk3-l | 0.979 | $TO$ | 596 | 0.979 | $TO$ | 123 | **0.981** | 7.38 | 8e4 | **0.981** | 1.11 | **9e3** |
| monk3-f | 0.975 | $TO$ | 1e4 | 0.973 | $TO$ | 9e3 | **0.983** | 2.13 | 5e4 | **0.983** | 1.14 | **9e3** |
| monk3-o | — | — | — | — | — | — | — | — | — | **0.987** | 0.04 | **156** |
| tic-tac-toe | 0.765 | $TO$ | 40 | 0.808 | $TO$ | 37 | **0.850** | 41 | 1.6e6 | **0.850** | 61 | **2.5e5** |
| tic-tac-toe-o | — | — | — | — | — | — | — | — | — | **0.773** | 0.90 | **3339** |
| car-eval | — | — | — | — | — | — | **0.799** | 18 | 9e5 | **0.799** | 56 | **3e5** |
| car-eval-o | — | — | — | — | — | — | — | — | — | **0.812** | 0.10 | **579** |
| nursery | — | — | — | — | — | — | 0.765 | $TO$ | 7e5 | **0.772** | 144 | **2e5** |
| nursery-o | — | — | — | — | — | — | — | — | — | **0.822** | 0.26 | **195** |
| mushroom | **0.945** | $TO$ | 4e6 | **0.945** | $TO$ | 2e6 | 0.925 | $TO$ | 1e6 | 0.938 | $TO$ | 2e4 |
| mushroom-o | — | — | — | — | — | — | — | — | — | **0.975** | 0.17 | **6** |
| kr-vs-kp | **0.900** | $TO$ | 6e4 | **0.900** | $TO$ | 2e4 | 0.815 | $TO$ | 4e5 | **0.900** | $TO$ | 8e4 |
| kr-vs-kp-o | — | — | — | — | — | — | — | — | — | **0.900** | $TO$ | 8e4 |
| zoo | — | — | — | — | — | — | **0.992** | 34 | 3e5 | **0.992** | 15 | **3e4** |
| zoo-o | — | — | — | — | — | — | — | — | — | **0.993** | 0.94 | **1456** |
| lymph | — | — | — | — | — | — | 0.784 | $TO$ | 1e6 | **0.808** | $TO$ | **1e5** |
| lymph-o | — | — | — | — | — | — | — | — | — | **0.852** | 12 | **1e4** |
| balance | **0.693** | $TO$ | 1e5 | **0.693** | $TO$ | 3e4 | **0.693** | 21 | 1e6 | **0.693** | 54 | **3e5** |
| balance-o | — | — | — | — | — | — | — | — | — | **0.671** | 0.02 | **126** |

It is difficult to analytically compare the the bound in Corollary 7 with the bound in (Hu et al., 2019, Theorem E.2). For this reason, we compare them numerically on some reasonable instances of the problem. Table 1 clearly shows the vast computational gains that BRANCHES offers over OSDT. This claim is further validated in our experiments. We note however, that the immense numbers upper bounding the complexity of OSDT *do not reflect OSDT's true complexity* but rather that the bound is too loose. Indeed, the reasoning behind (Hu et al., 2019, Theorem E.2) pertains to counting all the possible DTs which depths do not exceed the maximum depth of the optimal DT, it does not analyse OSDT's pruning capacity. In fact, it is unclear how such analysis could be performed with OSDT's pruning bounds. On the other hand, the Purification bound we provide in Proposition 4 offers a natural pruning strategy that allows for such analysis.

# 6 EXPERIMENTS

We compare BRANCHES with the state of the art based on the following metrics: **optimal convergence**, **execution time** and **number of iterations**. We provide the source code of our implementation in the supplementary material.

We employ 11 datasets from the UCI repository, which we chose because of their frequent use in benchmarking optimal Decision Tree algorithms. For each dataset, different types of encodings are considered: Suffix -l indicates a One-Hot Encoding where the last category of each feature is dropped, likewise -f drops the first category, -o is for an Ordinal Encoding. We chose different encodings because they yield problems with varying degrees of difficulty. Moreover, the state of the art algorithms exclusively consider binary features, thus necessitating a preliminary binary encoding. This seemingly benign detail can significantly harm performance by introducing unnecessary splits as we explain in Appendix C. BRANCHES can sidestep this issue since it is directly applicable to an Ordinal Encoding of the data. We set a time limit of 5 minutes for all experiments. Table 5 summarises the characteristics of the datasets we consider.

Table 2 shows that BRANCHES outperforms OSDT, PyGOSDT and GOSDT on almost all the experiments, with GOSDT being the most competitive method. We especially notice the large computational gains achieved by applying BRANCHES to the datasets in their original form through Ordinal Encoding. On the monk datasets, while both GOSDT and BRANCHES are always optimal, BRANCHES is faster, sometimes significantly. On the other hand, OSDT and Py-GOSDT are only optimal for monk1-l, and they are prohibitively slow. There are a few datasets where BRANCHES does not perform the best. On Mushroom, OSDT and PyGOSDT surpris-

Table 3: Comparing BRANCHES with CART, DL8.5, MurTree and STreeD; objective here refers to the regularised objective $\mathcal{H}_\lambda$. $TO$ refers to timeout.

| Dataset | CART objective | CART time (s) | DL8.5 objective | DL8.5 time (s) | MurTree objective | MurTree time (s) | STreeD objective | STreeD time (s) | BRANCHES objective | BRANCHES time (s) |
|---|---|---|---|---|---|---|---|---|---|---|
| monk1-l | 0.863 | 0.002 | 0.270 | 0.01 | **0.930** | **0.10** | **0.930** | 2.80 | **0.930** | 0.11 |
| monk1-f | 0.971 | 0.002 | 0.925 | 0.007 | 0.968 | 0.36 | **0.983** | 6.11 | **0.983** | 1.07 |
| monk1-o | — | — | — | — | — | — | — | — | **0.9** | **0.02** |
| monk2-l | 0.950 | 0.002 | 0.870 | 0.01 | 0.967 | 2.67 | **0.968** | 135 | **0.968** | 2.7 |
| monk2-f | 0.915 | 0.004 | 0.894 | 0.01 | 0.928 | 2.96 | — | TO | **0.933** | 5.29 |
| monk2-o | — | — | — | — | — | — | — | — | **0.955** | **0.10** |
| monk3-l | 0.979 | 0.002 | 0.938 | 0.02 | 0.970 | 0.79 | **0.981** | 9.77 | **0.981** | 1.11 |
| monk3-f | 0.980 | 0.003 | 0.957 | 0.009 | 0.966 | 0.02 | **0.983** | 3.98 | **0.983** | 1.14 |
| monk3-o | — | — | — | — | — | — | — | — | **0.987** | **0.04** |
| tic-tac-toe | 0.835 | 0.003 | −1.05 | 0.05 | **0.850** | **20** | 0.850 | 169 | **0.850** | 61 |
| tic-tac-toe-o | — | — | — | — | — | — | — | — | **0.773** | **0.90** |
| car-eval | 0.793 | 0.003 | −3.19 | 0.127 | **0.799** | 65 | — | TO | **0.799** | 56 |
| car-eval-o | — | — | — | — | — | — | — | — | **0.812** | **0.10** |
| nursery | 0.769 | 0.013 | −126 | 7.08 | **0.772** | 151 | — | TO | **0.772** | 144 |
| nursery-o | — | — | — | — | — | — | — | — | **0.822** | **0.26** |
| mushroom | 0.933 | 0.022 | 0.270 | 0.08 | **0.945** | 148 | **0.945** | 116 | 0.938 | TO |
| mushroom-o | — | — | — | — | — | — | — | — | **0.975** | **0.17** |
| kr-vs-kp | 0.888 | 0.004 | 0.434 | 28 | 0.583 | 122 | — | TO | **0.900** | TO |
| kr-vs-kp-o | — | — | — | — | — | — | — | — | **0.900** | TO |
| zoo | **0.992** | 0.002 | 0.983 | 0.36 | 0.989 | 0.36 | **0.992** | 21 | **0.992** | 15 |
| zoo-o | — | — | — | — | — | — | — | — | **0.993** | **0.94** |
| lymph | 0.779 | 0.003 | 0.24 | 0.01 | — | — | — | TO | **0.808** | TO |
| lymph-o | — | — | — | — | — | — | — | — | **0.852** | **12** |
| balance | 0.649 | 0.003 | −2.30 | 0.05 | 0.692 | 42 | — | TO | **0.693** | 54 |
| balance-o | — | — | — | — | — | — | — | — | **0.671** | **0.02** |

ingly outperform both BRANCHES and GOSDT; on tic-tac-toe, car-eval and balance, GOSDT and BRANCHES terminate but GOSDT is faster. However, we suspect that this is mainly due to GOSDT's optimised C++ implementation, which confers it an advantage over BRANCHES's Python implementation. In fact, the difference in speed between these programming languages is very evident from the large gap in execution times between GOSDT and PyGOSDT. We note however, that BRANCHES always converges in fewer iterations than GOSDT, around 10 times less in many cases. This corroborates our complexity analysis in Section 5, indicating that our Purification bound improves the pruning efficiency of the search algorithms. A future C++ implementation of BRANCHES will further improve BRANCHES's scalability, especially since it is amenable to true parallel computing. Indeed, the Algorithmic steps Selection, Expansion and Backpropagation can all be run through parallel synchronous threads. Unfortunately, Python is limited in its parallel computing capacity, it does not permit multithreading, and its multiprocessing module requires copying the data, which greatly slows down the algorithm and loses all the computational benefits of parallel computing.

In Table 3, for a fair comparison, since BRANCHES and GOSDT do not constrain the depth of the searched solutions, we impose a similar condition on the DL8.5, MurTree and STreeD. In the implementation of STreeD (as of version v1.3.1), a maximum depth lower than 20 has to be specified. For this reason, we set the maximum depth to 20, we further impose a maximum number of nodes of 80 to avoid memory issues. Table 3 shows that only BRANCHES and STreeD truly solve for sparsity, which means that upon terminating they return the optimal solution with respect to $\mathcal{H}_\lambda$. The second take-away from Table 3 is that BRANCHES outperforms STreeD on almost all the datasets except mushroom. Moreover, due to its heuristic nature, CART never achieves optimality in these experiments.

## 7 CONCLUSION, LIMITATIONS AND FUTURE WORK

For now BRANCHES is limited to categorical features. In fact, all the cited optimal DT methods were developed for categorical features and are applied to numerical features through discretisation. Furthermore, BRANCHES is currently implemented in Python, which hinders its execution times sometimes compared to the C++ implementations. The large gap in performance between PyGOSDT and GOSDT motivates a future implementation of BRANCHES in C++. Since BRANCHES far outperforms the existing Python methods and is even competitive and often better than GOSDT, we believe that a future C++ implementation of BRANCHES will yield further great improvements over the state of the are, especially in scalability.

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

## A    TABLE OF NOTATION

Table 4: Table of Notation

| | | |
|---:|:---:|:---|
| $X$ | $=$ | $\left(X^{(1)}, \ldots, X^{(q)}\right)$, an input of features. |
| $X^{(i)}$ | $\in$ | $\{1, \ldots, C_i\}$, an feature. |
| $Y$ | $\in$ | $\{1, \ldots, K\}$, a class. |
| $\mathcal{D}$ | $=$ | $\{(X_m, Y_m)\}_{m=1}^n$, Dataset of examples. |
| $l$ | $=$ | $\bigwedge_{v=1}^{\mathcal{S}(l)} \mathbb{1}\left\{X^{(i_v)} = j_v\right\}$ a branch. Also, a unit-state in our MDP. |
| $\mathcal{S}(l)$ | $\triangleq$ | The number of splits in $l$, the number of clauses in $l$. |
| $l(X)$ | $\triangleq$ | Valuation of $l$ for input $X$. When $l(X) = 1$, we say that $X$ is in $l$. |
| $\Omega$ | $\triangleq$ | The root. Branch that valuates to 1 for all possible inputs. |
| $\mathrm{Ch}(l, i)$ | $\triangleq$ | Children of $l$ when splitting with respect to feature $i$. |
| $\mathrm{Ch}(l, i)$ | $=$ | $\{l_1, \ldots, l_{C_i}\}, l_j = l \wedge \mathbb{1}\left\{X^{(i)} = j\right\}$ |
| $n(l)$ | $\triangleq$ | Number of examples in $l$. |
| $n_k(l)$ | $\triangleq$ | Number of examples in $l$ of class $k$. |
| $k^*(l)$ | $=$ | $\mathrm{Argmax}_{1 \le k \le K}\{n_k(l)\}$, majority class in $l$ |
| $\mathbb{P}[l(X) = 1]$ | $\triangleq$ | Empirical probability that $X$ is in $l$. |
| $\mathcal{H}(l)$ | $=$ | $\mathbb{P}[l(X) = 1, Y = k^*(l)]$, probability that an example is in $l$ and is correctly classified. |
| sub-DT rooted in $l$ | $\triangleq$ | Collection of branches partitioning $l$, it stems from a series of splits of $l$. Also a state in our MDP. |
| DT | $\triangleq$ | A sub-DT rooted in $\Omega$. |
| $T(X)$ | $\triangleq$ | Predicted class of $X$ by $T$. Majority class of the branch containing $X$. |
| $\mathcal{H}(T)$ | $=$ | $\mathbb{P}[T(X) = Y]$, accuracy of DT $T$. |
| $\mathcal{H}_\lambda(T)$ | $\triangleq$ | Regularized Objective function of DT $T$. |
| $\mathcal{H}_\lambda(T)$ | $=$ | $\mathbb{P}[T(X) = Y] - \lambda \mathcal{S}(T)$. |
| $\mathcal{S}(T)$ | $\triangleq$ | Number of splits to construct sub-DT $T$ from the branch where it is rooted. |
| $\lambda$ | $\in$ | $[0, 1]$, penalty parameter. |
| $T^*$ | $=$ | $\mathrm{Argmax}_T\{\mathcal{H}_\lambda(T)\}$, optimal DT. |
| $\mathcal{A}(T)$ | $\triangleq$ | Action space at state $T$. |
| $\overline{a}$ | $\triangleq$ | Terminal action. |
| $T \xrightarrow{a} T'$ | $\triangleq$ | Transition from $T$ to $T'$ through action $a$. |
| $\overline{T}$ | $\triangleq$ | Absorbing state, $T \xrightarrow{\overline{a}} \overline{T}$. |
| $r(T, a)$ | $\triangleq$ | Reward of taking action $a$ in state $T$. |
| $\pi$ | $\triangleq$ | Policy, maps each state $T$ to an action $\pi(T) \in \mathcal{A}(T)$. |
| $\mathcal{R}^\pi(T)$ | $\triangleq$ | Return of policy $\pi$ starting from $T$. |
| $\mathcal{Q}^\pi(T, a)$ | $\triangleq$ | State-action value of policy $\pi$ at state-action pair $(T, a)$. |
| $T_l^\pi$ | $\triangleq$ | Sub-DT of $\pi$ rooted in $l$. See Proposition 1. |
| $T^\pi$ | $\equiv$ | $T_\Omega^\pi$ |
| $\pi^*$ | $=$ | $\mathrm{Argmax}_\pi \mathcal{R}^\pi(\Omega)$, the optimal policy. |
| $T^*$ | $=$ | $T^{\pi^*}$ |
| $\mathcal{R}^*$ | $\equiv$ | $\mathcal{R}^{\pi^*}$ |
| $\mathcal{Q}^*$ | $\equiv$ | $\mathcal{Q}^{\pi^*}$ |
| $\mathcal{R}(T)$ | $\triangleq$ | Estimated upper bound on $\mathcal{R}^*(T)$. |
| $\mathcal{Q}(T, a)$ | $\triangleq$ | Estimated upper bound on $\mathcal{Q}^*(T, a)$ |

## B  Proofs

**Proposition 1.** *Let $\pi$ be a policy, $l$ a unit-state and consider $T_0 = l$ and $\forall t \geq 0 : T_t \xrightarrow{\pi(T_t)} T_{t+1}$. Then, there exists $\tau \geq 0$ such that for any $t \geq \tau$, $T_t = \{\overline{l_1}, \ldots, \overline{l_{|T_\tau|}}\}$ is an absorbing state. In which case we call $T_l^\pi = \{l_1, \ldots, l_{|T_\tau|}\}$ the sub-DT of $\pi$ rooted in $l$. If $l = \Omega$ we abbreviate the notation $T_\Omega^\pi \equiv T^\pi$ and call $T^\pi$ the DT of $\pi$.*

*Proof.* Let $l$ be a unit-state, $\pi$ a policy, and $(T_t)_{t=0}^\infty$ such that:

$$\begin{cases} T_0 = l \\ \forall t \geq 0 : T_t \xrightarrow{\pi(T_t)} T_{t+1} \end{cases}$$

The proof is conducted by induction on $q - \mathcal{S}(l) \in \{0, \ldots, q\}$, where we recall that $q$ is the number of features.

If $q - \mathcal{S}(l) = 0$, then $\mathcal{A}(l) = \{\overline{a}\}$ and $\pi(l) = \overline{a}$. Therefore $T_1 = \overline{l}$ and we deduce that the proposition holds with $\tau = 1$.

**Inductive hypothesis:** Suppose that the proposition is true for $q - \mathcal{S}(l') = n \in \{0, \ldots, q-1\}$, and let us show that it is true for $q - \mathcal{S}(l) = n + 1$.

If $\pi(l) = \overline{a}$, then $T_1 = \overline{l}$ and the proposition holds. Otherwise $\pi(l)$ is a split action, and we have $l \xrightarrow{\pi(l)} T_1 = \{l_1, \ldots, l_{|T_1|}\}$ where:

$$\forall u \in \{1, \ldots, |T_1|\} : q - \mathcal{S}(l_u) = n$$

Therefore, the proposition is true for all $l_u$.
Let us now denote the following:

$$\begin{cases} T_1^{(u)} = l_u \\ \forall t \geq 1 : T_t^{(u)} \xrightarrow{\pi} T_{t+1}^{(u)} \end{cases}$$

According to the proposition:

$$\exists \tau_u \geq 0, \forall t \geq \tau_u : T_t^{(u)} = \left\{ \overline{l_1^{(u)}}, \ldots, \overline{l_{|T_{\tau_u}^{(u)}|}^{(u)}} \right\}$$

By taking $\tau = \max_{1 \leq u \leq |T_1|}\{\tau_u\}$, we get:

$$\forall t \geq \tau, \forall u \in \{1, \ldots, |T_1|\} : T_t^{(u)} = \left\{ \overline{l_1^{(u)}}, \ldots, \overline{l_{|T_{\tau_j}^{(u)}|}^{(u)}} \right\} = \left\{ \overline{l_1^{(u)}}, \ldots, \overline{l_{|T_\tau^{(u)}|}^{(u)}} \right\}$$

On the other hand $\forall t \geq 1 : T_t = \bigcup_{u=1}^{|T_1|} T_t^{(u)}$, thus:

$$\forall t \geq \tau : T_t = \bigcup_{u=1}^{|T_1|} \left\{ \overline{l_1^{(j)}}, \ldots, \overline{l_{|T_\tau^{(u)}|}^{(u)}} \right\}$$

Which concludes the inductive proof. $\square$

**Proposition 2.** *For any policy $\pi$ and unit-state $l$, the return of $\pi$ from $l$ satisfies:*

$$\mathcal{R}^\pi(l) = \mathcal{H}_\lambda(T_l^\pi) = -\lambda \mathcal{S}(T_l^\pi) + \mathcal{H}(T_l^\pi)$$

*In particular $\mathcal{R}^\pi(\Omega) = \mathcal{H}_\lambda(T^\pi) = -\lambda \mathcal{S}(T^\pi) + \mathcal{H}(T^\pi)$.*

*Proof.* Let $l$ be a unit-state, $\pi$ a policy, and $(T_t)_{t=0}^\infty$ such that:

$$\begin{cases} T_0 = l \\ \forall t \geq 0 : T_t \xrightarrow{\pi(T_t)} T_{t+1} \end{cases}$$

By Induction on $\mathcal{S}(T_l^\pi)$:

If $\mathcal{S}\left(T_l^\pi\right) = 0$, then $\pi\left(l\right) = \bar{a}$ and $\forall t \geq 1 : T_t = \bar{l}$. Thus:

$$\mathcal{R}^\pi\left(l\right) = \underbrace{r\left(l, \bar{a}\right)}_{=\mathcal{H}(l)} + \sum_{t \geq 1} \underbrace{r\left(\bar{l}, \pi\left(\bar{l}\right)\right)}_{=0} = \mathcal{H}\left(l\right)$$

On the other hand $T_l^\pi = l$, therefore:

$$\mathcal{H}_\lambda\left(T_l^\pi\right) = -\lambda \underbrace{\mathcal{S}\left(T_l^\pi\right)}_{=0} + \mathcal{H}\left(l\right) = \mathcal{H}\left(l\right)$$

Hence $\mathcal{R}^\pi\left(l\right) = \mathcal{H}_\lambda\left(T_l^\pi\right)$

**Inductive hypothesis:** Suppose the proposition is true up to $\mathcal{S}\left(T_l^\pi\right) = n \geq 0$ and let us prove it for $\mathcal{S}\left(T_l^\pi\right) = n + 1$

If $\pi\left(l\right) = \bar{a}$ then we have again:

$$\mathcal{R}^\pi\left(l\right) = \underbrace{r\left(l, \bar{a}\right)}_{=\mathcal{H}(l)} + \sum_{t \geq 1} \underbrace{r\left(\bar{l}, \pi\left(\bar{l}\right)\right)}_{=0} = \mathcal{H}\left(l\right)$$

Since $T_l^\pi = l$, then:

$$\mathcal{H}_\lambda\left(T_l^\pi\right) = -\lambda \underbrace{\mathcal{S}\left(T_l^\pi\right)}_{=0} + \mathcal{H}\left(l\right) = \mathcal{H}\left(l\right)$$

Hence $\mathcal{R}^\pi\left(l\right) = \mathcal{H}_\lambda\left(T_l^\pi\right)$

Now suppose that $\pi\left(l\right)$ is a split action. We have the following:

$$\mathcal{R}^\pi\left(l\right) = r\left(l, \pi\left(l\right)\right) + \sum_{t=1}^\infty r\left(T_t, T_{t+1}\right)$$

$$= r\left(l, \pi\left(l\right)\right) + \mathcal{R}^\pi\left(T_1\right)$$

$$= r\left(l, \pi\left(l\right)\right) + \sum_{u=1}^{|T_1|} \mathcal{R}^\pi\left(l_u\right)$$

Where $T_1 = \{l_1, \ldots, l_{|T_1|}\}$. We know that:

$$\forall u \in \{1, \ldots, |T_1|\} : \mathcal{R}^\pi\left(l_u\right) = \mathcal{H}_\lambda\left(T_{l_u}^\pi\right)$$

$$\implies \mathcal{R}^\pi\left(l\right) = -\lambda + \sum_{u=1}^{|T_1|} \left\{ -\lambda \mathcal{S}\left(T_{l_u}^\pi\right) + \mathcal{H}\left(T_{l_u}^\pi\right) \right\}$$

$$= -\lambda\left\{1 + \sum_{u=1}^{|T_1|} \mathcal{S}\left(l_u\right)\right\} + \mathcal{H}\left(T_l^\pi\right)$$

We know that the total number of splits to construct $T_l^\pi$ is 1 (corresponding to the split $\pi\left(l\right)$) plus the sum of the number of splits required to construct each sub-DT $T_{l_u}^\pi$, i.e.

$$\mathcal{S}\left(T_l^\pi\right) = 1 + \sum_{u=1}^{|T_1|} \mathcal{S}\left(T_{l_u}^\pi\right)$$

Therefore we deduce that:

$$\mathcal{R}^\pi\left(l\right) = -\lambda \mathcal{S}\left(T_l^\pi\right) + \mathcal{H}\left(T_l^\pi\right) = \mathcal{H}_\lambda\left(T_l^\pi\right)$$

Which concludes the inductive proof. $\qquad\square$

**Proposition 4** (Purification Bound)**.** *For any non-absorbing unit-state $l$ and split action $a \in \mathcal{A}\left(l\right) \setminus \{\bar{a}\}$, we define the Purification Bound estimates:*

$$\mathcal{Q}\left(l, a\right) = -\lambda + \mathbb{P}\left[l\left(X\right) = 1\right] = -\lambda + \frac{n\left(l\right)}{n} \tag{7}$$

$$\mathcal{R}\left(l\right) = \max\{\mathcal{H}\left(l\right), -\lambda + \mathbb{P}\left[l\left(X\right) = 1\right]\} = \max\left\{\frac{n_{k^*\left(l\right)}\left(l\right)}{n}, -\lambda + \frac{n\left(l\right)}{n}\right\} \tag{8}$$

*Then the estimates $\mathcal{Q}\left(l, a\right)$ and $\mathcal{R}\left(l\right)$ are upper bounds on $\mathcal{Q}^*\left(l, a\right)$ and $\mathcal{R}^*\left(l\right)$ respectively.*

*Proof.* Let $l$ be a non-terminal unit-state, $a \in \mathcal{A}(l) \setminus \{\overline{a}\}$ and:

$$\mathcal{Q}(l, a) = -\lambda + \mathbb{P}[l(X) = 1]$$

Let us show that $\mathcal{Q}(l, a) \geq \mathcal{Q}^*(l, a)$. Consider $l \xrightarrow{\pi(l)} T_1 = \{l_1, \ldots, l_{|T_1|}\}$, we have the following:

$$\mathcal{Q}^*(l, a) = -\lambda + \sum_{u=1}^{|T_1|} \mathcal{R}^*(l_u)$$

According to Proposition 2, we have:

$$\forall u \in \{1, \ldots, |T_1|\} : \mathcal{R}^*(l_u) = \mathcal{H}_\lambda\left(T_{l_u}^*\right)$$

$$\mathcal{Q}^*(l, a) = -\lambda + \sum_{u=1}^{|T_1|} \mathcal{H}_\lambda\left(T_{l_u}^*\right)$$

On the other hand, we have:

$$\forall u \in \{1, \ldots, |T_1|\} : \mathcal{H}_\lambda\left(T_{l_u}^*\right) = -\lambda \mathcal{S}\left(T_{l_u}^*\right) + \mathcal{H}\left(T_{l_u}^*\right)$$
$$\leq \mathcal{H}\left(T_{l_u}^*\right)$$
$$\leq \mathbb{P}\left[l_u(X) = 1, T_{l_u}^*(X) = Y\right]$$
$$\leq \mathbb{P}[l_u(X) = 1]$$

Which implies the following:

$$\mathcal{Q}^*(l, a) \leq -\lambda + \sum_{u=1}^{|T_1|} \mathbb{P}[l_u(X) = 1] \leq -\lambda + \mathbb{P}[l(X) = 1] = \mathcal{Q}(l, a)$$

For the optimal value function, we have:

$$\mathcal{R}^*(l) = \max_{a \in \mathcal{A}(l)} \mathcal{Q}^*(l, a)$$

$$= \max_{a \in \mathcal{A}(l) \setminus \{\overline{a}\}} \left\{\mathcal{Q}^*(l, \overline{a}), \mathcal{Q}^*(l, a)\right\}$$

$$\leq \max_{a \in \mathcal{A}(l) \setminus \{\overline{a}\}} \left\{\mathcal{H}(l), \mathcal{Q}(l, a)\right\}$$

$$\leq \max\left\{\mathcal{H}(l), -\lambda + \mathbb{P}[l(X) = 1]\right\}$$

$\square$

**Lemma 8.** *For any unit-state $l$ and action $a \in \mathcal{A}(l)$, the estimate $\mathcal{Q}(l, a)$ is an upper bound on the optimal state values.*

$$\mathcal{Q}(l, a) \geq \mathcal{Q}^*(l, a)$$

*Proof.* For the terminal action, we always have:

$$\mathcal{Q}(l, \overline{a}) = \mathcal{H}(l) = \mathcal{Q}^*(l, \overline{a})$$

Let us now consider a split action $a \in \mathcal{A}(l) \setminus \{\overline{a}\}$. We have the following:

$$\mathcal{Q}(l, a) = -\lambda + \sum_{u=1}^{|T_1|} \mathcal{R}(l_u)$$

Where $l \xrightarrow{a} T_1 = \{l_1, \ldots, l_{|T_1|}\}$. It suffices to show that:

$$\forall u \in \{1, \ldots, |T_1|\} : \mathcal{R}(l_u) \geq \mathcal{R}^*(l_u)$$

We define the following policy:

$$\begin{cases} \pi(l') = \mathrm{Argmax}_{a' \in \mathcal{A}(l')} \mathcal{Q}(l', a') & \text{for } l' \text{ that have been visited.} \\ \pi(l') = \overline{a} & \text{for } l \text{ that have never been visited.} \end{cases}$$

The proof now proceeds by induction on the number of visits of $l$ which we denote here $v(l) \geq 0$.

If $v(l) = 0$, then:
$$\mathcal{R}(l) = \max\left\{\mathcal{H}(l), -\lambda + \mathbb{P}\left[l(X) = 1\right]\right\} \geq \mathcal{R}^*(l)$$

**Induction hypothesis:** Suppose that this is true for any number of visits $\leq n$ where $n \geq 0$, and let us show that the result still holds for $v(l) = n + 1$. We have

$$\mathcal{R}(l) = \max\left\{\mathcal{H}(l), -\lambda + \sum_{u=1}^{|T_1|} \mathcal{R}(l_u)\right\}$$

On the other hand

$$\forall u \in \{1, \ldots, |T_1|\} : v(l_u) \leq n$$
$$\implies \forall u \in \{1, \ldots, |T_1|\} : \mathcal{R}(l_u) \geq \mathcal{R}^*(l_u)$$

Thus

$$\mathcal{R}(l) \geq \max\{\mathcal{H}(l), -\lambda + \sum_{u=1}^{|T_1|} \mathcal{R}^*(l_u)\} = \mathcal{R}^*(l)$$

Which concludes the inductive proof, and we get that:

$$\forall u \in \{1, \ldots, |T_1|\} : \mathcal{R}(l_u) \geq \mathcal{R}^*(l_u)$$

Implying

$$\mathcal{Q}(l, a) = -\lambda + \sum_{u=1}^{|T_1|} \mathcal{R}(l_u) \geq \mathcal{Q}^*(l, a)$$

**Remark:** During the inductive reasoning, we used the fact that the number of visits of children branches is lower than the number of visits of their parent branch. However, this is not true when Dynamic Programming is considered. Indeed, due to memoisation, some children branches could have been visited more than their parents. The result still stems from a similar induction, albeit through a more technical proof. The general idea is that, for children branches $l_u$ that are visited more than $n + 1$ times, we consider their children, and so on, until we arrive at descendant branches that are either visited less than $n$ times or that are terminal. In both cases $\mathcal{R}(l_u) \geq \mathcal{R}^*(l_u)$, and we backpropagate this result to $\mathcal{R}(l)$. $\qquad\square$

**Theorem 5** (Optimality of BRANCHES). *When* BRANCHES *terminates, the optimal policy is the greedy policy with respect to the estimated state-action values* $\mathcal{Q}(l, a)$*, which means that for any state* $T$*:*
$$\pi^*(T) = \text{Argmax}_{a \in \mathcal{A}(T)} \mathcal{Q}(T, a)$$

*Proof.* Define the policy $\widetilde{\pi}(T) = \text{Argmax}_{\mathcal{A}(T)} \mathcal{Q}(T, a)$. First, we show that for any unit-state $l$, if $l$ is complete and $a^* = \text{Argmax}_{a \in \mathcal{A}(l)} \mathcal{Q}(l, a)$, then $a^* = \pi^*(l)$.

Since $l$ is complete, we have $\mathcal{Q}(l, a^*) = \mathcal{Q}^*(l, a^*)$. By Lemma 8, we get

$$\forall a \in \mathcal{A}(l) : \mathcal{Q}^*(l, a^*) = \mathcal{Q}(l, a^*) \geq \mathcal{Q}(l, a) \geq \mathcal{Q}^*(l, a)$$
$$\implies a^* = \text{Argmax}_{a \in \mathcal{A}(l)} \mathcal{Q}^*(l, a) = \pi^*(l)$$

On the other hand, $l$ is complete if and only if $(l, \pi^*(l))$ is complete, which is satisfied if and only if for all $u \in \{1, \ldots, |T|\} : l_u$ is complete, where $l \xrightarrow{\pi^*(l)} T = \{l_1, \ldots, l_{|T|}\}$.

BRANCHES terminates when $\Omega$ is complete. Let us define the following:

$$\begin{cases} T_0 = \Omega \\ \forall t \geq 0 : T_t \xrightarrow{\widetilde{\pi}(T_t)} T_{t+1}; \ T_t = \left\{l_1^{(t)}, \ldots, l_{|T_t|}^{(t)}\right\} \end{cases}$$

Since $\Omega$ is complete, we have shown $\widetilde{\pi}\left(\Omega\right) = \pi^*\left(\Omega\right)$, and it follows that:

$$\forall u \in \{1, \ldots, |T_1|\} : l_u^{(1)} \text{ is complete}$$

$$\implies \forall u \in \{1, \ldots, |T_1|\} : \widetilde{\pi}\left(l_u^{(1)}\right) = \pi^*\left(l_u^{(1)}\right)$$

$$\vdots$$

$$\implies \forall u \in \{1, \ldots, |T_t|\} : l_u^{(t)} \text{ is complete}$$

$$\implies \forall u \in \{1, \ldots, |T_t|\} : \widetilde{\pi}\left(l_u^{(t)}\right) = \pi^*\left(l_u^{(t)}\right)$$

$$\vdots$$

Thus $\widetilde{\pi}$ is optimal:

$$\mathcal{R}^*\left(\Omega\right) = \sum_{t=0}^{\infty} r\left(T_t, \pi^*\left(T_t\right)\right)$$

$$= \sum_{t=0}^{\infty} \sum_{u=1}^{|T_t|} r\left(l_u^{(t)}, \pi^*\left(l_u^{(t)}\right)\right)$$

$$= \sum_{t=0}^{\infty} \sum_{u=1}^{|T_t|} r\left(l_u^{(t)}, \widetilde{\pi}\left(l_u^{(t)}\right)\right)$$

$$= \sum_{t=0}^{\infty} r\left(T_t, \widetilde{\pi}\left(T_t\right)\right)$$

$$= \mathcal{R}^{\widetilde{\pi}}\left(\Omega\right)$$

$\square$

**Theorem 9.** *Let $\Gamma_{\text{OSDT}}\left(q, \lambda\right)$ denote the total number of evaluations that OSDT performs for an instance of the binary classification problem with $q \geq 2$ binary features and penalty parameter $0 \leq \lambda \leq 1$, then we have:*

$$\Gamma_{\text{OSDT}}\left(q, \lambda\right) \leq 1 + \sum_{h=1}^{\kappa} \left\{ N_h + \binom{q}{h} - P\left(q, h\right) \right\}$$

*Where $P\left(q, h\right)$ is the number $h-$permutations of $q$, $N_h$ is the number of possible binary DTs of depth $h$ defined in ([Hu et al., 2019](#), Formula (1)) and:*

$$\kappa = \min\left\{ \left\lfloor \frac{1}{2\lambda} \right\rfloor - 1, q \right\}$$

The difference with ([Hu et al., 2019](#), Theorem E.2) is in the term $\kappa$, the authors write it as:

$$\kappa = \min\left\{ \left\lfloor \frac{1}{2\lambda} \right\rfloor, 2^q \right\}$$

The term $2^q$ is the maximum number of leaves that any DT can have, however, following the authors' reasoning, it should be the maximum possible depth, which is $q$. Furthermore, the term $\left\lfloor \frac{1}{2\lambda} \right\rfloor$ is an upper bound on the maximum number of leaves the optimal solution can have. Such solution has at most a depth of $\left\lfloor \frac{1}{2\lambda} \right\rfloor - 1$. Indeed, the maximum depth of a DT $T$ with $|T|$ leaves is $|T| - 1$, this corresponds to a DT that only splits one node at each depth (for example, always splitting the right child node).

**Lemma 10.** *A branch $l$ can be chosen for Expansion only if there exists a DT $T$ such that:*

$$\begin{cases} l \in T \setminus L \\ -\lambda \mathcal{S}\left(T\right) + \sum_{l' \in L} \mathcal{H}\left(l'\right) + \sum_{l' \in T \setminus L} \left\{ -\lambda + \mathbb{P}\left[l'\left(X\right) = 1\right] \right\} \geq -\lambda \mathcal{S}\left(T^*\right) + \mathcal{H}\left(T^*\right) \end{cases}$$

*Where $L = \{l' \in L : \mathcal{H}\left(l'\right) \geq -\lambda + \mathbb{P}\left[l'\left(X\right) = 1\right]\}$.*

*Proof.* Let $\widetilde{\pi}$ be the Selection policy, i.e. for any unit-state $l$:

$$\widetilde{\pi}(l) = \begin{cases} \overline{a} \text{ If } l \text{ has never been visited} \\ \text{Argmax}_{a \in \mathcal{A}(l)} \mathcal{Q}(l, a) \text{ Otherwise.} \end{cases}$$

For the current Selection policy $\widetilde{\pi}$, a branch $l$ is chosen for Expansion only if $l \in T^{\widetilde{\pi}}$, thus let us analyse the properties of $T^{\widetilde{\pi}}$.

By the definition of $\widetilde{\pi}$, $T^{\widetilde{\pi}}$ maximising $\mathcal{R}(T)$:

$$\forall \text{DT } T : \mathcal{R}(T) \leq \mathcal{R}\left(T^{\widetilde{\pi}}\right)$$

$$\implies \mathcal{R}(T^*) \leq \mathcal{R}\left(T^{\widetilde{\pi}}\right)$$

$$\implies \mathcal{R}^*(T^*) \leq \mathcal{R}\left(T^{\widetilde{\pi}}\right)$$

$$\implies -\lambda\mathcal{S}(T^*) + \mathcal{H}(T^*) \leq \mathcal{R}\left(T^{\widetilde{\pi}}\right)$$

On the other hand we have:

$$\mathcal{R}\left(T^{\widetilde{\pi}}\right) = \sum_{l \in T^{\widetilde{\pi}}} \mathcal{R}(l)$$

Let $L = \{l \in L : \mathcal{H}(l) \geq -\lambda + \mathbb{P}[l(X) = 1]\}$. For any $l \in L$ we have $\mathcal{R}(l) = \mathcal{H}(l)$ and for any $l \in T \setminus L$ we have $\mathcal{R}(l) = -\lambda + \mathbb{P}[l(X) = 1]$. Therefore we deduce that:

$$-\lambda\mathcal{S}\left(T^{\widetilde{\pi}}\right) + \sum_{l' \in L} \mathcal{H}(l') + \sum_{l' \in T^{\widetilde{\pi}} \setminus L} \left\{-\lambda + \mathbb{P}[l'(X) = 1]\right\} \geq -\lambda\mathcal{S}(T^*) + \mathcal{H}(T^*)$$

The first condition for a branch $l$ to be considered for Expansion is $l \in T^{\widetilde{\pi}}$. For the second condition, $l$ cannot be in $L$, because all branches in $L$ are complete and satisfy $\overline{a} = \text{Argmax}_{a \in \mathcal{A}(l)} \mathcal{Q}^*(l, a)$. Indeed this is due to the following:

$$\mathcal{Q}^*(l, \overline{a}) = \mathcal{H}(l) \geq -\lambda + \mathbb{P}[l(X) = 1] \geq \mathcal{Q}^*(l, a) \ \forall a \in \mathcal{A}(l)$$

where the last inequality comes from Proposition 4. Now we deduce that the second condition for $l$ to be considered for Expansion is $l \in T \setminus L$. $\qquad\square$

**Theorem 6** (Problem-dependent complexity of BRANCHES). *Let $\Gamma(q, C, \lambda)$ denote the total number of branch evaluations performed by* BRANCHES *for an instance of the classification problem with $q \geq 2$ features, $0 < \lambda \leq 1$ the penalty parameter, and $C \geq 2$ the number of categories per feature. Then, $\Gamma(q, C, \lambda)$ satisfies the following bound:*

$$\Gamma(q, C, \lambda) \leq \sum_{h=0}^{\kappa} (q - h) C^{h+1} \binom{q}{h}; \ \kappa = \min\left\{\left\lfloor \mathcal{S}(T^*) - 1 + \frac{1 - \mathcal{H}(T^*)}{\lambda} \right\rfloor, q\right\}$$

*Proof.* Let $l$ be a branch. According to Lemma 10, for $l$ to be considered for Expansion, there has to exist a DT $T$ such that:

$$\begin{cases} l \in T \setminus L \\ -\lambda\mathcal{S}(T) + \sum_{l' \in L} \mathcal{H}(l') + \sum_{l' \in T \setminus L} \left\{-\lambda + \mathbb{P}[l'(X) = 1]\right\} \geq -\lambda\mathcal{S}(T^*) + \mathcal{H}(T^*) \end{cases}$$

where $L = \{l' \in T : \mathcal{H}(l') \geq -\lambda + \mathbb{P}[l'(X) = 1]\}$. Suppose $l$ is such a branch, then we have:

$$-\lambda\mathcal{S}(T) + \sum_{l' \in L} \underbrace{\mathcal{H}(l')}_{\leq \mathbb{P}[l'(X)=1]} + \sum_{l' \in T \setminus L} \left\{-\lambda + \mathbb{P}[l'(X) = 1]\right\} \geq -\lambda\mathcal{S}(T^*) + \mathcal{H}(T^*)$$

$$\implies -\lambda\left\{\mathcal{S}(T) + |T \setminus L|\right\} + \sum_{l' \in T} \mathbb{P}[l'(X) = 1] \geq -\lambda\mathcal{S}(T^*) + \mathcal{H}(T^*)$$

Since $l \in T \setminus L$, then $|T \setminus L| \geq 1$ and we get:

$$-\lambda \Big\{ \mathcal{S}(T) + 1 \Big\} + 1 \geq -\lambda \mathcal{S}(T^*) + \mathcal{H}(T^*)$$

$$\implies \mathcal{S}(T) \leq \mathcal{S}(T^*) - 1 + \frac{1 - \mathcal{H}(T^*)}{\lambda}$$

$$\implies \mathcal{S}(l) \leq \mathcal{S}(T^*) - 1 + \frac{1 - \mathcal{H}(T^*)}{\lambda}$$

Let $\mathcal{C} = \Big\{ l \text{ branch} : \mathcal{S}(l) \leq \mathcal{S}(T^*) - 1 + \frac{1 - \mathcal{H}(T^*)}{\lambda} \Big\}$. Then the number of branches that are expanded is upper bounded by $|\mathcal{C}|$.

We recall that we rather seek to upper bound the number of branches that are evaluated, i.e. for which we calculate $\mathcal{H}(l)$. These evaluations happen during the Expansion step of BRANCHES. When a branch $l$ is expanded, we evaluate all of its children. There are $q - \mathcal{S}(l)$ features left to use for splitting $l$, and for each split, $C$ children branches are created. Thus, there are $(q - \mathcal{S}(l)) C$ children of $l$, hence $(q - \mathcal{S}(l)) C$ evaluations happen during the expansion of $l$. Let us now upper bound $\Gamma(q, C, \lambda)$.

For each branch $l \in \mathcal{C}$:

- We choose $\mathcal{S}(l) \in \Big\{ 0, \ldots, \min \Big\{ \Big\lfloor \mathcal{S}(T^*) - 1 + \frac{1 - \mathcal{H}(T^*)}{\lambda} \Big\rfloor, q \Big\} \Big\}$. The minimum comes from the fact that $l \in \mathcal{C}$ and $\mathcal{S}(l) \leq q$.

- For each $h = \mathcal{S}(l)$, we construct $l$ by choosing $h$ features among the total $q$ features, there are $\binom{q}{h}$ such choices.

- For each choice among the $\binom{q}{h}$ choices, for each feature among the $h$ features, there are $C$ choices of values, therefore there are $C^h \binom{q}{h}$ branches with depth $h$.

- For each branch of depth $h$, when it is expanded, $(q - h) C$ evaluations occur.

With these considerations, we deduce that:

$$\Gamma(q, C, \lambda) \leq \sum_{h=0}^{\kappa} (q - h) C^{h+1} \binom{q}{h}; \ \kappa = \min \Big\{ \Big\lfloor \mathcal{S}(T^*) - 1 + \frac{1 - \mathcal{H}(T^*)}{\lambda} \Big\rfloor, q \Big\}$$

$\square$

**Corollary 7** (Problem-independent complexity of BRANCHES)**.** *Let* $\Gamma(q, \lambda, C)$ *be defined as in* *Theorem 6, then it satisfies:*

$$\Gamma(q, C, \lambda) \leq \sum_{h=0}^{\kappa} (q - h) C^{h+1} \binom{q}{h}; \ \kappa = \min \Big\{ \Big\lfloor \frac{1}{K\lambda} \Big\rfloor - 1, q \Big\}$$

*Proof.* To make the bound problem-independent, let us upper bound $\kappa$ and make it independent of $T^*$. We know that:

$$\mathcal{H}_\lambda(T^*) = -\lambda \mathcal{S}(T^*) + \mathcal{H}(T^*) \geq \mathcal{H}_\lambda(\Omega) = \mathcal{H}(\Omega) = \mathbb{P}[Y = k^*(\Omega)] \geq \frac{1}{K}$$

$$\implies \mathcal{S}(T^*) - 1 + \frac{1 - \mathcal{H}(T^*)}{\lambda} \leq \frac{K - 1}{K\lambda} - 1$$

Which concludes the proof. $\square$

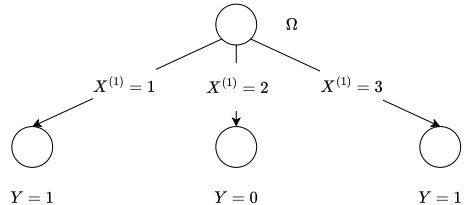

Figure 1: Optimal DT depicting the class variable that satisfies $Y = 1$ if and only if $X^{(1)} = 1$ or $X^{(1)} = 3$ on the space $\mathcal{X} = \{1, 2, 3\}$.

## C   THE DRAWBACKS OF BINARY ENCODING

Table 2 and Table 3 show that the optimal DT is always achieved significantly faster when we consider the Ordinal Encoding[1]. Interestingly, BRANCHES is the only method that can be directly applied with Ordinal Encoding, which makes it even more practical and broadly applicable than the state of the art. The central question of this section is: **Why does Ordinal Encoding provide such great leaps in efficiency compared to Binary Encoding?**

To answer this question, let us consider the following simple binary classification problem. Suppose there is only one feature $X^{(1)}$ with 3 categories, i.e. the space of features is $\mathcal{X} = \{1, 2, 3\}$, and that the class $Y$ satisfies $Y = 1$ if and only if $X^{(1)} = 1$ or $X^{(1)} = 3$. The optimal solution in this case consists of only one split, which is to split the root $\Omega$ with respect to feature $X^{(1)}$, as shown in Fig. 1. In this setting, BRANCHES only needs one iteration to terminate. Indeed, on its first iteration, it expands $\Omega$, estimates $\mathcal{Q}(\Omega, \overline{a})$ and $\mathcal{Q}(\Omega, a)$ where $a$ is the split action with respect to $X^{(1)}$. In this case, BRANCHES can already deduce that:

$$\mathcal{Q}^*(\Omega, a) = \mathcal{Q}(\Omega, a) = -\lambda + \underbrace{\mathbb{P}[\Omega(X) = 1]}_{=1} > \mathbb{P}[\Omega(X) = 1, k^*(\Omega) = Y] = \mathcal{Q}^*(\Omega, \overline{a})$$

and therefore that $\Omega$ is complete and $a = \text{Argmax}_{a' \in \mathcal{A}(\Omega)} \mathcal{Q}^*(\Omega, a')$.

Let us consider a Binary encoding of $\mathcal{X}$, this yields a new feature space $\mathcal{X}' = \{0, 1\} \times \{0, 1\} \times \{0, 1\}$ where the new features $X'^{(1)}, X'^{(2)}, X'^{(3)}$ express the existence of a category or the other:

$$\forall i \in \{1, 2, 3\} : X'^{(i)} = \mathbb{1}\{X^{(1)} = i\}$$

Fig. 2 depicts the new optimal Decision Tree on $\mathcal{X}'$. Now BRANCHES cannot deduce this solution from the first iteration, because the first iteration only explores branches of size 1 and the optimal DT includes also branches of sizes 2 and 3. Moreover, Binary encoding introduces unnecessary branches that make the search space larger than necessary, thereby wasting some of the search time. To see this, consider the branch:

$$l' = \mathbb{1}\{X'^{(1)} = 1\} \wedge \mathbb{1}\{X'^{(2)} = 1\}$$

This branch exists in the new lattice of branches constructed on $\mathcal{X}'$ and it could be explored at some point by the search algorithm. However, this would be a waste of time because $l'$ does not describe a possible subset of $\mathcal{X}$. Indeed, translating $l'$ to its corresponding branch on $\mathcal{X}$ yields:

$$l = \mathbb{1}\{X^{(1)} = 1\} \wedge \mathbb{1}\{X^{(1)} = 2\}$$

which always valuates to $0$ for any datum $X \in \mathcal{X}$. As a consequence, $l$ can never be part of the optimal solution, in fact, it can never be part of any Decision Tree on $\mathcal{X}$, $l$ is not even a proper branch as it uses the same feature in two different clauses. Therefore exploring $l'$ while solving the DT optimisation on $\mathcal{X}'$ is a waste of time.

---

[1] Except for kr-vs-kp, this is because this dataset is already in binary form.

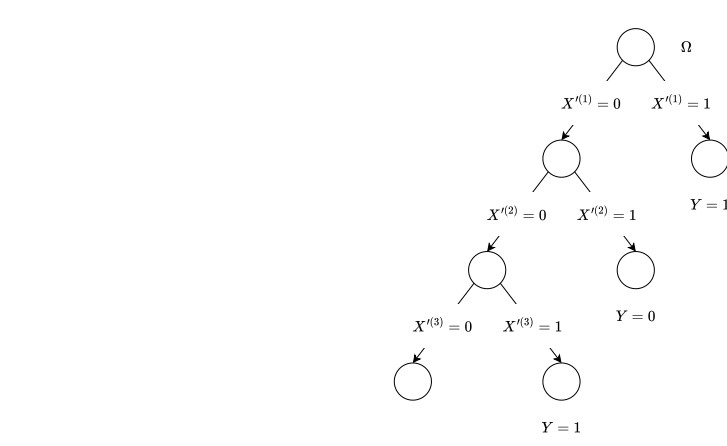

Figure 2: The new optimal Decision Tree on the full new feature space $\mathcal{X}'$.

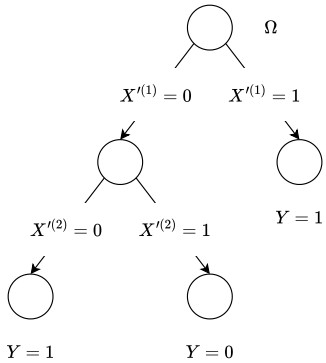

Figure 3: The new optimal Decision Tree on the reduced new feature space $\mathcal{X}'$.

The Binary Encoding we considered in the last paragraph keeps all the categories of $X^{(1)}$. In general, a more clever Binary Encoding is used based on the following:

$$X'^{(1)} = 0 \text{ and } X'^{(2)} = 0 \iff X'^{(3)} = 1$$

This allows us to drop the last feature from $\mathcal{X}'$, yielding a smaller new feature space $\mathcal{X}' = \{0, 1\} \times \{0, 1\}$ where the features $X'^{(1)}, X'^{(2)}$ are the same as before. The new optimal solution is depicted in Fig. 3 and it only includes two splits instead of three, unlike in the previous Binary Encoding, thus allowing the search to be more efficient but still less efficient than when operating on the original feature space $\mathcal{X}$. Moreover, the issue of unnecessary branches still holds here, the branch $l' = \mathbb{1}\{X'^{(1)} = 1\} \wedge \mathbb{1}\{X'^{(2)} = 1\}$ is still present in this setting. The computational inefficiency induced by Binary Encoding can be evaluated by the number of these introduced unnecessary branches. Theorem 11 provides this number for the case where all features have and equal number categories.

**Theorem 11.** *Consider a classification problem where all features share the same number of categories $C$, i.e. $\mathcal{X} = \{1, \ldots, C\}^q$. Suppose we perform a Binary Encoding on $\mathcal{X}$ where the last category of each feature is dropped, this yields the new feature space $\mathcal{X}' = \{0, 1\}^{q(C-1)}$. We define an unnecessary branch $l$ on $\mathcal{X}'$ as a branch that valuates to 0 for any input vector $X \in \mathcal{X}$:*

$$\forall X \in \mathcal{X} : l\left(X\right) = 0$$

*Then the number of these unnecessary branches that Binary Encoding introduces is equal to:*

$$\mathcal{U}\left(q, C\right) = 3^{(C-1)q} - \left[2\left(C - 1\right) + 1\right]^q$$

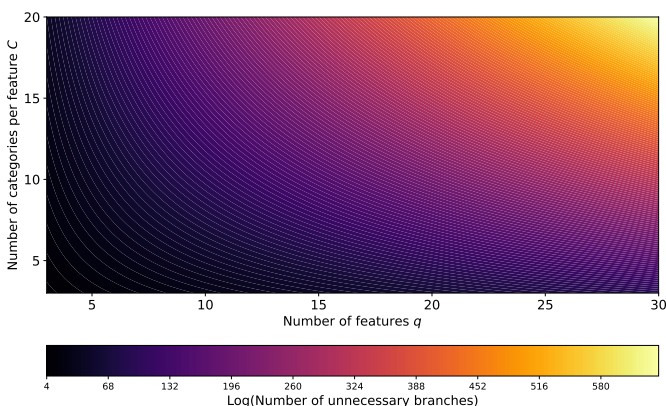

Figure 4: The number of unnecessary branches introduced by Binary Encoding.

*Proof.* The proof of this Theorem proceeds by counting the total number of branches possible on $\mathcal{X}'$ and subtracting the total number of branches that are not unnecessary.

Let us start with the total number of branches on $\mathcal{X}'$. Any branch on $\mathcal{X}'$ has the form:

$$l = \bigwedge_{v=1}^{w} \mathbb{1}\{X'^{(i_v)} = z_v\}$$

Where $X'^{(i_v)}$ are the features on the space $\mathcal{X}'$, $w \in \{0, \ldots, (C-1)q\}, i_v \in \{1, \ldots, (C-1)q\}, z_v \in \{0,1\}$. We note that $w = 0$ corresponds to $l = \Omega$ by definition.

- There $(C-1)q$ possibilities for choosing $w$.

- For each possible value $w$, there are $\binom{(C-1)q}{w}$ possible combinations $\{i_1, \ldots, i_w\}$.

- For each combination $\{i_1, \ldots, i_w\}$, there are $2^w$ possible assignments $(z_1, \ldots, z_w)$

Therefore the total number of branches on $\mathcal{X}'$ is:

$$\mathcal{A}(q, C) = \sum_{w=0}^{(C-1)q} \binom{(C-1)q}{w} 2^w = 3^{(C-1)q} \tag{10}$$

Let us now count the number of non-unnecessary branches. To do this, we consider a slightly different notation of the features on $\mathcal{X}'$.

$$\forall i \in \{1, \ldots, q\}, \forall j \in \{1, \ldots, C-1\} : X'^{(i,j)} = \mathbb{1}\{X^{(i)} = j\}$$

A branch $l = \bigwedge_{v=1}^{w} \mathbb{1}\{X'^{(i_v, j_v)} = z_v\}$ is not unnecessary if and only if $w \in \{0, \ldots, q\}, i_v \in \{1, \ldots, q\}, j_v \in \{1, \ldots, C-1\}, z_v \in \{0,1\}$.

- For each possibility value $w \in \{1, \ldots, q\}$, there are $\binom{q}{w}$ possible combinations $\{i_1, \ldots, i_w\}$.

- For each combination $\{i_1, \ldots, i_w\}$, there are $(C-1)^w$ possible assignments $(j_1, \ldots, j_w)$.

- For each assignment $(j_1, \ldots, j_w)$, there are $2^w$ possible assignments $(z_1, \ldots, z_w)$.

The total number of branches that are not unnecessary is therefore:

$$\mathcal{B}(q, C) = \sum_{w=0}^{q} \binom{q}{w} 2^w (C-1)^w = [2(C-1) + 1]^q \tag{11}$$

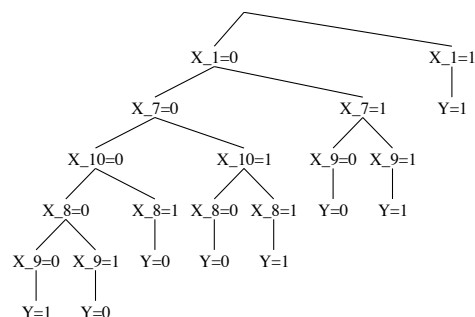

Figure 5: Optimal Decision Tree for monk1-l, it has 7 splits.

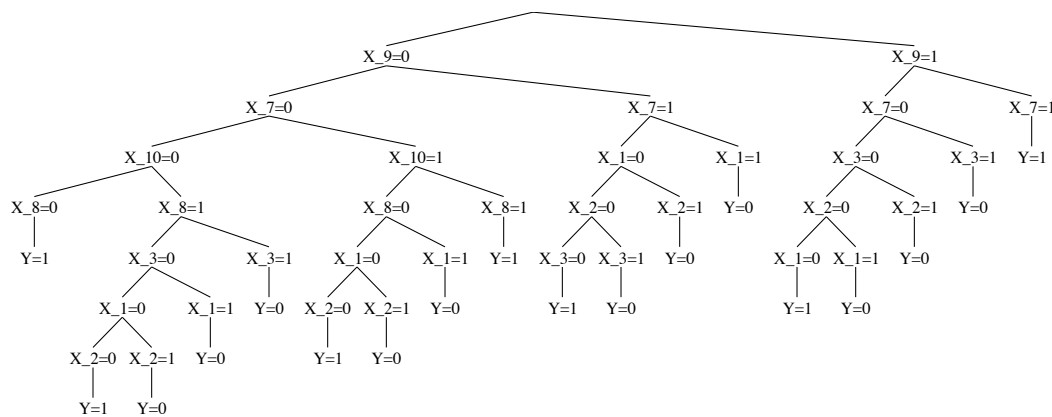

Figure 6: Optimal Decision Tree for monk1-f, it has 17 splits.

From Eq. (10) and Eq. (11) we deduce that the total number of unnecessary branches is:

$$\mathcal{U}(q, C) = \mathcal{A}(q, C) - \mathcal{B}(q, C) = 3^{(C-1)q} - [2(C-1) + 1]^q$$

$\square$

There is a subtlety here. We define $l$ on $\mathcal{X}'$, which means that it involves clauses defined with the features of $\mathcal{X}'$, and yet the definition in Theorem 11 pertains to valuating $l$ on inputs from the feature space $\mathcal{X}$. There is no mistake or lack of rigour in this definition, we are allowed to do this because the Binary Encoding is an injective map from $\mathcal{X}$ to $\mathcal{X}'$, thus implicitly, valuating $l$ on an input $X \in \mathcal{X}$ is defined as valuating $l$ on the image of $X$ in $\mathcal{X}'$ with this map.

Fig. 4 draws the number of unnecessary branches, derived in Theorem 11, as a contour function of $q$ and $C$ in Logarithmic scale. It shows how immense this number becomes as $q$ and $C$ increase. We should note that, not all of these unnecessary branches, that Binary Encoding introduces, will be explored by BRANCHES, in fact many of them (depending on the problem) will not be due to the algorithm's pruning capacity. Nevertheless, there are so many that they will inevitably hinder the search efficiency as it is clearly demonstrated in Table 2. This inefficiency is most apparent on the mushroom dataset. All algorithms that solve for sparsity hit a timeout (after 5 minutes) when applied to the binary encoded version of the data. In contrast, when applied to the Ordinal Encoding of mushroom, BRANCHES achieves an extremely fast optimal convergence in only $0.17s$ and 6 iterations.

The introduction of unnecessary branches is not the only drawback of Binary Encoding. To perform Binary Encoding, we also have to decide which category to drop from each feature, however different choices lead to different feature spaces with different optimal Decision Trees. Furthermore, these different solutions do not necessarily share similar complexities, and these choices can lead to problems with vastly different levels of challenge. A pertinent example of this is the contrast

between monk1-l and monk1-f. While all algorithms that solve for sparsity achieve optimal convergence for monk1-l, only GOSDT, STreeD and BRANCHES find the optimal solution for monk1-f. This is because monk1-l yields an optimal DT with 7 splits only while monk1-f yields an optimal solution with 17 splits, which is significantly more challenging to find. These optimal Decision Trees are depicted in Fig. 5 and Fig. 6.

# D    IMPLEMENTATION DETAILS

The search strategy we introduced in Section 4.2 is an abstract description of BRANCHES. In this section, we provide concrete elements for the implementation of the algorithm, along with micro-optimisation techniques that substantially improve its computational efficiency.

## D.1    BRANCH OBJECTS

For each branch $l = \bigwedge_{v=1}^{\mathcal{S}(l)} \mathbb{1}\{X^{(i_v)} = j_v\}$, we define an object with the following elements:

- `id_branch`: $l$ is identified with the unique string $"(i_1,j_1)(i_1,j_2)\ldots(i_{\mathcal{S}(l)},j_{\mathcal{S}(l)})"$. We recall that this string is unique because we impose the condition $i_1 < i_2 < \ldots < i_{\mathcal{S}(l)}$. We store each encountered branch in a memo dictionary using its identifier.

- `attributes_categories`: Dictionary containing the number of categories per unused feature in $l$. We recall that the set of unused features is the set of split actions.

- `bit_vector`: Vector of the indices of the data contained in $l$. This vector allows quick access to the data in $l$.

- `children`: Dictionary containing the children of $l$, i.e. the set $\mathrm{Ch}(l,i)$ for all each unused feature $i$ in $l$. Initialised with an empty dictionary.

- `attribute_opt`: The current optimal action $a^* = \mathrm{Argmax}_{a \in \mathcal{A}(l)} \mathcal{Q}(l,a)$. If $a^* = \overline{a}$, then we set `attribute_opt` to None.

- `terminal`: Boolean describing whether $l$ is terminal or not, we say that $l$ is terminal if the set of permissible actions at $l$ only includes the terminal action, i.e. $\mathcal{A}(l) = \overline{a}$.

- `complete`: Boolean describing whether $l$ is complete or not.

- `value`: The estimated $\mathcal{R}(l)$.

- `value_terminal`: The value of the terminal action at $l$.

$$Q^*(l,\overline{a}) = \mathcal{H}(l) = \mathbb{P}[l(X) = 1, k^*(l) = Y] = \frac{n_{k^*(l)}(l)}{n}$$

- `value_greedy`: Value of the current best action to take according the estimates $\mathcal{Q}(l,a)$:

$$\texttt{value\_greedy} = \mathrm{Argmax}_{a \in \mathcal{A}(l)} \mathcal{Q}(l,a) = \mathcal{Q}(l, \texttt{attribute\_opt})$$

- `freq`: Proportion of examples in $l$:

$$\texttt{freq} = \mathbb{P}[l(X) = 1] = \frac{n(l)}{n} = \frac{1}{n}\sum_{m=1}^{n} l(X_m)$$

- `pred`: Majority class at $l$:

$$\texttt{pred} = k^*(l) = \mathrm{Argmax}_{1 \le k \le K} n_k(l) = \mathrm{Argmax}_{1 \le k \le K} n_k(l)$$

- `queue`: Heap queue containing (`-value`, `value_complete`, `attribute`, `children`) tuples. For each unused feature (split action) `attribute`: `value` is the estimate:

$$\texttt{value} = \mathcal{Q}(l, \texttt{attribute}) = -\lambda + \sum_{l' \in \mathrm{Ch}(l, \texttt{attribute})} \mathcal{R}(l')$$

On the other hand, `value_complete` is the sum of the estimated values $\mathcal{R}(l')$ of the children $l' \in \mathrm{Ch}(l, \texttt{attribute})$ that are complete. By definition, the complete children $l'$ satisfy $\mathcal{R}(l') = \mathcal{R}^*(l')$, we store the sum of their values in `value_complete`,

which serves to efficiently update $\mathcal{Q}\left(l,\texttt{attribute}\right)$ during the Backpropagation step. `children` is a dictionary containing the incomplete children, it is from this dictionary that we choose the next branch to visit during the Selection step. During Backpropagation, If an incomplete branch $l'$ in `children` becomes complete, it is discarded from `children`. We note that these tuples are stored in the heap queue `queue`, thus the first element of `queue` is always the tuple with the highest `value`, i.e. `queue[0][2]` is the split action maximising $\mathcal{Q}\left(l,a\right)$. We do not need to sort all actions by their values, but rather to just keep track of the action with the highest value. As a result, $l$ becomes complete if and only if one of the following holds:

- The terminal action is the current best action:

$$\mathcal{Q}\left(l,\overline{a}\right) = \text{Argmax}_{a \in \mathcal{A}(l)}\mathcal{Q}\left(l,a\right)$$

This happens if:
$$-\texttt{queue[0][0]} \le \texttt{value\_terminal}$$

- The tuple containing $l$ and the current best split action is complete. This happens if the dictionary of incomplete children (that result from taking the current best split action in $l$) `queue[0][3]` is empty.

`queue` is initialised with an empty queue.

## D.2 THE ALGORITHM

In this section, we go over BRANCHES' search strategy, introduced in Section 4.2, and we outline it from an implementation perspective. We initialise the root $\Omega$, then we apply the search steps at each iteration as follows:

- **Selection:** Initialise the current branch $l = \Omega$ and the path list to `path = [l]`. While $l$ is incomplete and $l$.`children` is not empty, i.e. $l$ has been expanded. Consider the tuple:

  ```
  (-value, value_complete, attribute, children) = l.queue[0]
  ```

  As we have seen in Appendix D.1, `attribute` is the optimal split action with respect to the current estimates $\mathcal{Q}\left(l,a\right)$ and `children` is the subset of incomplete children in $\text{Ch}\left(l,\texttt{attribute}\right)$. Therefore, we choose the next branch $l$ from the dictionary `children`. This choice can be arbitrary or according to some scheduling policy[2]. Choosing the branch $l$ in `children` with lowest $l$.`value_greedy` is our practical choice. The reasoning behind it is to quickly prune non-promising regions of the search space. Append $l$ to `path`.

- **Expansion:** Let $l$ be the Selected branch. If $l$.`complete`, we go to the Backpropagation step. Otherwise, for each (unused) feature-category $(i,j) \in l$.`attributes_categories` let $l_{ij} = l \wedge \mathbb{1}\{X^{(i)} = j\}$ be the child branch of $l$ that corresponds to feature $i$ taking the value $j$. Our objective is to calculate $\mathcal{R}\left(l_{ij}\right)$. We first check whether $l_{ij}$.`id_branch` is in the memo, if it is, then we can directly access $\mathcal{R}\left(l_{ij}\right)$. Otherwise, we need to initialise $\mathcal{R}\left(l_{ij}\right)$ according to Eq. (8). To do this efficiently, consider a fixed feature $i$ and let us go over its categories $j \in \{1,\ldots,C_i\}$ one by one. For $l_{i1}$, we first extract the data in $l$ using $l$.`bit_vector`:

$$\mathcal{D}_l = \{X_m \in \mathcal{D} : l\left(X_m\right) = 1\} = \mathcal{D}[l.\texttt{bit\_vector}]$$

Since $l_{i1}\left(X\right) = 1 \implies l\left(X\right) = 1$, we can extract the data in $l_{i1}$ directly from the smaller set $\mathcal{D}_l$ instead of $\mathcal{D}$:

$$\mathcal{D}_{l_{i1}} = \{X_m \in \mathcal{D} : l_{i1}\left(X_m\right) = 1\} = \{X_m \in \mathcal{D}_l : l_{i1}\left(X_m\right) = 1\}$$

The indices of the data in $\mathcal{D}_{l_{i1}}$ form the vector $l_{i1}$.`bit_vector`. Now we can initialise $\mathcal{R}\left(l_{i1}\right)$ with Eq. (8) using $\mathcal{D}_{l_{i1}}$. For $l_{i2}$, if $l_{i2}$.`id_branch` is not in the memo, then to initialise $\mathcal{R}\left(l_{i2}\right)$, instead of extracting $\mathcal{D}_{l_{i2}}$ from $\mathcal{D}_l$ via:

$$\mathcal{D}_{l_{i2}} = \{X_m \in \mathcal{D}_l : l_{i2}\left(X_m\right) = 1\}$$

---

[2]The term scheduling policy is employed by Hu et al. (2019) in a similar context.

We rather use the fact that $l_{i1}$ and $l_{i2}$ are mutually exclusive, in the sense that:

$$\forall X \in \mathcal{X} : l_{i2}(X) = 1 \implies l_{i1}(X) = 0$$

Which means that we can extract $\mathcal{D}_{l_{i2}}$ from the smaller set $\mathcal{D}_l \setminus \mathcal{D}_{l_{i1}}$ instead of $\mathcal{D}_l$ and then initialise $\mathcal{R}(l_{i2})$. We repeat this process for all categories $j \in \{1, \ldots, C_i\}$ and then we do the same thing for the remaining unused features in $l$.attributes_categories. These micro-optimisations we perform allow for substantial computational efficiency.

- **Backpropagation:** For $j = length(\text{path}) - 1, \ldots, 1$ let `parent = path[j-1]` and `child = path[j]`, then we pop the heap queue `parent.queue`:

  `(-value, value_complete, attribute, children) = parent.queue.pop()`

  During the Selection step, `attribute` was the action taken at the branch `parent` to transition to the branch `child`. Now during Backpropagation, we need to update the estimates $\mathcal{Q}(\text{parent, attribute})$ and $\mathcal{R}(\text{parent})$, hence why we pop the corresponding tuple from `parent.queue`, and once we update its values, we push the tuple back in the heap queue. This rearranges the tuples so that the tuple with highest `value` will be at `parent.queue[0]`.
  If `child.complete` then we add its value to `value_complete`:

  $$\text{value\_complete} \leftarrow \text{value\_complete} + \text{child.value}$$

  and we pop `child` from the dictionary of incomplete children `children.pop(child)`. Now `parent.queue[0]` is the tuple corresponding to the best split action:

  `(-value, value_complete, attribute, children) = parent.queue[0]`

  Therefore, the value of `parent` is equal to the maximum between the value of taking this best split action and the value of taking the terminal action:

  $$\mathcal{R}(\text{parent}) = \max\left\{ \mathcal{Q}(\text{parent}, \bar{a}), \mathcal{Q}(\text{parent}, \text{attribute}) \right\}$$

  Which, in our implementation translates into:

  $$\text{parent.value} \leftarrow \max\left\{ \text{parent.value\_terminal}, \text{value} \right\}$$

  If $\mathcal{R}(\text{parent}) = \mathcal{Q}(\text{parent}, \bar{a})$, then $\bar{a} = \text{Argmax}_{a \in \mathcal{A}(\text{parent})} \mathcal{Q}(\text{parent}, a)$, and since we know that $\mathcal{Q}^*(\text{parent}, \bar{a}) = \mathcal{Q}(\text{parent}, \bar{a})$ (according to Eq. (4)), then we deduce that `parent` is complete and $\mathcal{R}^*(\text{parent}) = \mathcal{Q}^*(\text{parent}, \bar{a})$. Therefore we update:

  $$\text{parent.complete} \leftarrow \text{True}$$

  This is not the only condition that makes `parent` complete. Indeed, `parent` can also be complete if `(parent, attribute)` is complete, which happens when the dictionary `children` is empty.

# E PSEUDOCODE

---

**Algorithm 1** BRANCHES

---

1: **Input:** Dataset $\mathcal{D} = \{(X_m, Y_m)\}_{m=1}^n$, penalty parameter $\lambda \geq 0$.
2: memo $\leftarrow \{\}$      ▷ Initialise an empty memo
3: INITIALISE$(\Omega, \mathcal{D})$
4: **while** not $\Omega$.complete **do**
5:     $(l, \text{path}) \leftarrow$ SELECT()
6:     **if** $l$.complete **then**
7:         BACKPROPAGATE(path)
8:     **else**
9:         EXPAND$(l, \mathcal{D})$
10:         BACKPROPAGATE(path)
11:     **end if**
12: **end while**
13: **return** INFER()
14: **procedure** SELECT()
15:     $l \leftarrow \Omega$
16:     path $\leftarrow [l]$
17:     **while** $l$.expanded and (not $l$.complete) **do**
18:         $(\mathcal{Q}(l, i), \text{return\_complete}, i, \text{children\_incomplete}) \leftarrow l.\text{queue}[0]$
19:         $l \leftarrow \text{children\_incomplete}[0]$
20:         path.append$(l)$
21:     **end while**
22:     **return** $(l, \text{path})$
23: **end procedure**
24: **procedure** EXPAND$(l, \mathcal{D})$
25:     $l$.expanded $\leftarrow True$
26:     **for** $i \in \mathcal{A}(l) \setminus \{\bar{a}\}$ **do**
27:         SPLIT$(l, i, \mathcal{D})$
28:         $\mathcal{R}(l) \leftarrow \max\{\mathcal{Q}(l, \bar{a}), l.\text{queue}[0][0]\}$      ▷ This update comes from Eq. (6)
29:     **end for**
30:     **if** $\mathcal{R}(l) = \mathcal{Q}(l, \bar{a})$ **then**      ▷ In this case $\mathcal{R}^*(l) = \mathcal{Q}^*(l, \bar{a}) = \mathcal{H}(l)$
31:         $l$.complete $\leftarrow True$      ▷ $\mathcal{R}^*(l)$ is known
32:         $l$.terminal $\leftarrow True$      ▷ Label $l$ terminal if the optimal action at $l$ is $\pi^*(l) = \bar{a}$
33:     **end if**
34: **end procedure**
35: **procedure** BACKPROPAGATE(path)
36:     $N \leftarrow \text{length}(\text{path})$
37:     **for** $t = N - 2$ to $0$ **do**
38:         $l \leftarrow \text{path}[t]$
39:         $(\mathcal{Q}(l, i), \text{return\_complete}, i, \text{children\_incomplete}) \leftarrow l.\text{queue.pop}()$
40:         $\mathcal{Q}(l, i) \leftarrow \text{return\_complete}$      ▷ Initialise $\mathcal{Q}(l, i)$
41:         **for** $l' \in \text{children\_incomplete}$ **do**
42:             $\mathcal{Q}(l, i) \leftarrow \mathcal{Q}(l, i) + \mathcal{R}(l')$
43:             **if** $l'$.complete **then**      ▷ Check if $l'$ is complete now
44:                 children\_incomplete.discard$(l')$      ▷ Delete $l'$ from children\_incomplete
45:             **end if**
46:         **end for**
47:         $l$.queue.push$((\mathcal{Q}(l, i), \text{return\_complete}, i, \text{children\_incomplete}))$
48:         $(\mathcal{Q}(l, i^*), \text{return\_complete}, i^*, \text{children\_incomplete}) \leftarrow l.\text{queue}[0][0]$
49:         $\mathcal{R}(l) \leftarrow \mathcal{Q}(l, i^*)$
50:         **if** $(\mathcal{R}(l) = \mathcal{Q}(l, \bar{a}))$ or (children\_incomplete is empty) **then**
51:             $l$.complete $\leftarrow True$
52:             $l$.terminal $\leftarrow True$      ▷ Label $l$ terminal if the optimal action at $l$ is $\pi^*(l) = \bar{a}$
53:         **end if**
54:     **end for**
55: **end procedure**

---

```
56: procedure INITIALISE(l, D)
57:     l.expanded ← False                                              ▷ Label l as not expanded yet
58:     l.children ← dict ()                                            ▷ Initialise the dictionary of children
59:     l.queue ← queue ([])                                            ▷ Initialise the priority queue of l
60:     Q (l, ā) ← H (l)                                                ▷ H (l) is calculated with D
61:     if A (l) = {ā} then
62:         l.terminal ← True                                          ▷ Label l as terminal if it cannot be split
63:         l.complete ← True                                          ▷ R* (l) is known
64:         R (l) ← Q (l, ā)                                            ▷ In this case R* (l) = Q* (l, ā) = H (l)
65:     else
66:         l.terminal ← False
67:         Initialise R (l) according to Eq. (6) and Eq. (7)
68:         if R (l) = Q (l, ā) then
69:             l.complete ← True                                      ▷ R* (l) is known, R* (l) = Q* (l, ā) = H (l)
70:             l.terminal ← True                                      ▷ Label l terminal if the optimal action at l is π* (l) = ā
71:         else
72:             l.complete ← False                                     ▷ R* (l) is still unknown
73:         end if
74:     end if
75:     memo.add(l)                                                    ▷ Add the initialised branch to the memo
76: end procedure
77: procedure SPLIT(l, i, D)
78:     l.children[i] ← []                          ▷ Initialise the list of children that stem taking split action i in l
79:     Q (l, i) ← −λ                                                   ▷ Initialise the Upper Bound Q (l, i)
80:     return_complete ← −λ                                           ▷ Initialise the return due to complete children
81:     children_incomplete ← []                                       ▷ Initialise the list of incomplete children
82:     for j ∈ {1, . . . , C_i} do
83:         l_ij ← l ∧ 𝟙{X^(i) = j}
84:         if l_ij ∉ memo then                        ▷ Only initialise the branches that are not in the memo
85:             INITIALISE(l_ij, D)
86:         end if
87:         l.children[i].append (l_ij)
88:         Q (l, i) ← Q (l, i) + R (l_ij)                             ▷ Update the Upper Bound Q (l, i)
89:         if l_ij.complete then
90:             return_complete ← return_complete + R (l_ij)
91:         else
92:             children_incomplete.append (l_ij)
93:         end if
94:     end for
95:     l.queue.push ((Q (l, i) , return_complete, i, children_incomplete))
96: end procedure
97: procedure INFER()
98:     T ← []
99:     Q ← queue ()
100:    Q.put (Ω)
101:    while Q is not empty do
102:        l ← Q.pop ()
103:        if l.terminal then
104:            T.append (l)
105:        else
106:            (Q (l, i) , return_complete, i, children_incomplete) ← l.queue[0]
107:            for l' ∈ l.children [i] do
108:                Q.put (l')
109:            end for
110:        end if
111:    end while
112:    return T
113: end procedure
```

Table 5: Number of examples $n$, number of features $q$, number of classes $K$ and penalty parameter $\lambda$ for the different datasets used in our experiments.

| Dataset | $n$ | $q$ | $K$ | $\lambda$ |
|---|---|---|---|---|
| monk1-l | 124 | 11 | 2 | 0.01 |
| monk1-f | 124 | 11 | 2 | 0.001 |
| monk1-o | 124 | 6 | 2 | 0.01 |
| monk2-l | 169 | 11 | 2 | 0.001 |
| monk2-f | 169 | 11 | 2 | 0.001 |
| monk2-o | 169 | 6 | 2 | 0.001 |
| monk3-l | 122 | 11 | 2 | 0.001 |
| monk3-f | 122 | 11 | 2 | 0.001 |
| monk3-o | 122 | 6 | 2 | 0.001 |
| tic-tac-toe | 958 | 18 | 2 | 0.005 |
| tic-tac-toe-o | 958 | 9 | 2 | 0.005 |
| car-eval | 1728 | 15 | 4 | 0.005 |
| car-eval-o | 1728 | 6 | 4 | 0.005 |
| nursery | 12960 | 19 | 5 | 0.01 |
| nursery-o | 12960 | 8 | 4 | 0.01 |
| mushroom | 8124 | 95 | 2 | 0.01 |
| mushroom-o | 8124 | 22 | 2 | 0.01 |
| kr-vs-kp | 3196 | 37 | 2 | 0.01 |
| kr-vs-kp-o | 3196 | 36 | 2 | 0.01 |
| zoo | 101 | 20 | 7 | 0.001 |
| zoo-o | 101 | 16 | 7 | 0.001 |
| lymph | 148 | 18 | 4 | 0.01 |
| lymph-o | 148 | 41 | 4 | 0.01 |
| balance | 576 | 16 | 2 | 0.01 |
| balance-o | 576 | 4 | 2 | 0.01 |

## F    EXPERIMENTAL DETAILS

Table 5 describes the properties and the setup for each one of our experiments.

### F.1    CROSSVALIDATION RESULTS

In this section, we perform a 5 fold crossvalidation comparing BRANCHES with the other algorithms in terms of the training and test accuracies, train and test objectives, and number of splits of the proposed solutions.

Table 6 and Table 9 show that the methods that solve for sparsity display similar performance (when they terminate) on almost all the experiments, which reinforces the exactitude of their implementations being faithful to their theoretical optimality guarantee. There are however few cases where there is a discrepancy between their test accuracies, *even when they terminate*. This is the case for monk3-f and zoo for example. The three algorithms BRANCHES, GOSDT and STreeD find the same DT solutions during the crossvalidation training, the difference in test accuracies is due to different predicted classes in branches (leaves) that contain no training example, but contain some test examples. In these branches, the choice of the predicted class is arbitrary, which explain the noticed discrepancy.

The second remark from these results is that BRANCHES is robust to memory issues unlike GOSDT and MurTree. Moreover, we notice that STreeD, in Table 9, does not have an anytime property as it only suggests a DT solution if it terminates. All the other methods on the other hand suggested solutions even when they did not terminate.

Table 7 is the most prone to overfitting, it yields $100\%$ training accuracy on all experiments, yet due to the overly complicated DT solutions it suggests, it scores poorly in the other metrics. This is not

Table 6: 5 folds cross-validation train/test results for BRANCHES and GOSDT. acc refers to Accuracy, obj refers to the objective $\mathcal{H}_\lambda(T)$, splits refers to the number of splits $\mathcal{S}(T)$. The kernel dies for GOSDT on mushroom and lymph due to high memory consumption.

| Dataset | GOSDT | | | | | BRANCHES | | | | |
|---|---|---|---|---|---|---|---|---|---|---|
| | train acc | train obj | test acc | test obj | splits | train acc | train obj | test acc | test obj | splits |
| monk1-l | $1\pm0$ | $0.936\pm0.008$ | $0.844\pm0.196$ | $0.780\pm0.188$ | $6.4\pm0.8$ | $1\pm0$ | $0.936\pm0.008$ | $0.844\pm0.196$ | $0.780\pm0.188$ | $6.4\pm0.8$ |
| monk1-f | $1\pm0$ | $0.986\pm0.002$ | $0.750\pm0.108$ | $0.736\pm0.108$ | $14\pm2.3$ | $1\pm0$ | $0.986\pm0.002$ | $0.764\pm0.168$ | $0.750\pm0.166$ | $14\pm2.3$ |
| monk2-l | $1\pm0$ | $0.971\pm0.002$ | $0.812\pm0.152$ | $0.783\pm0.149$ | $28.8\pm2.4$ | $1\pm0$ | $0.971\pm0.002$ | $0.847\pm0.107$ | $0.818\pm0.105$ | $28.8\pm2.4$ |
| monk2-f | $1\pm0$ | $0.948\pm0.001$ | $0.521\pm0.065$ | $0.469\pm0.065$ | $51.8\pm1.3$ | $1\pm0$ | $0.948\pm0.001$ | $0.503\pm0.042$ | $0.451\pm0.042$ | $51.8\pm1.3$ |
| monk3-l | $1\pm0$ | $0.984\pm0.001$ | $0.796\pm0.084$ | $0.780\pm0.085$ | $16.2\pm1.1$ | $1\pm0$ | $0.984\pm0.001$ | $0.812\pm0.076$ | $0.796\pm0.075$ | $16.2\pm1.1$ |
| monk3-f | $1\pm0$ | $0.986\pm0.002$ | $0.869\pm0.041$ | $0.855\pm0.039$ | $13.8\pm1.9$ | $1\pm0$ | $0.986\pm0.002$ | $0.760\pm0.132$ | $0.747\pm0.132$ | $13.8\pm1.9$ |
| tic-tac-toe | $0.961\pm0.006$ | $0.864\pm0.005$ | $0.790\pm0.104$ | $0.693\pm0.105$ | $19.4\pm1$ | $0.961\pm0.006$ | $0.864\pm0.005$ | $0.790\pm0.103$ | $0.693\pm0.103$ | $19.4\pm1$ |
| car-eval | $0.885\pm0.005$ | $0.817\pm0.006$ | $0.647\pm0.081$ | $0.579\pm0.079$ | $13.6\pm1.8$ | $0.885\pm0.005$ | $0.817\pm0.006$ | $0.647\pm0.081$ | $0.579\pm0.079$ | $13.6\pm1.8$ |
| nursery | $0.830\pm0.016$ | $0.776\pm0.015$ | $0.758\pm0.039$ | $0.704\pm0.042$ | $5.4\pm0.5$ | $0.878\pm0.019$ | $0.794\pm0.004$ | $0.652\pm0.100$ | $0.568\pm0.108$ | $8.4\pm1.7$ |
| mushroom | — | — | — | — | — | $0.974\pm0.011$ | $0.944\pm0.011$ | $0.837\pm0.141$ | $0.807\pm0.141$ | $3\pm0$ |
| kr-vs-kp | $0.835\pm0.039$ | $0.809\pm0.035$ | $0.774\pm0.076$ | $0.748\pm0.074$ | $2.6\pm0.5$ | $0.944\pm0.011$ | $0.902\pm0.011$ | $0.929\pm0.043$ | $0.887\pm0.043$ | $4.2\pm0.4$ |
| zoo | $1\pm0$ | $0.993\pm0.001$ | $0.940\pm0.058$ | $0.933\pm0.058$ | $7.4\pm0.8$ | $1\pm0$ | $0.993\pm0.001$ | $0.960\pm0.058$ | $0.953\pm0.058$ | $7.4\pm0.8$ |
| lymph | — | — | — | — | — | $0.885\pm0.026$ | $0.819\pm0.013$ | $0.776\pm0.054$ | $0.710\pm0.059$ | $6.6\pm1.3$ |
| balance | $0.817\pm0.022$ | $0.745\pm0.009$ | $0.373\pm0.163$ | $0.301\pm0.174$ | $7.2\pm1.7$ | $0.817\pm0.022$ | $0.745\pm0.009$ | $0.373\pm0.163$ | $0.301\pm0.174$ | $7.2\pm1.7$ |

Table 7: 5 folds cross-validation train/test results for BRANCHES and DL8.5. acc refers to Accuracy, obj refers to the objective $\mathcal{H}_\lambda(T)$, splits refers to the number of splits $\mathcal{S}(T)$.

| Dataset | DL8.5 | | | | | BRANCHES | | | | |
|---|---|---|---|---|---|---|---|---|---|---|
| | train acc | train obj | test acc | test obj | splits | train acc | train obj | test acc | test obj | splits |
| monk1-l | $1\pm0$ | $0.404\pm0.020$ | $0.629\pm0.064$ | $0.033\pm0.051$ | $59.6\pm1.9$ | $1\pm0$ | $0.936\pm0.008$ | $0.844\pm0.196$ | $0.780\pm0.188$ | $6.4\pm0.8$ |
| monk1-f | $1\pm0$ | $0.939\pm0.002$ | $0.589\pm0.047$ | $0.528\pm0.046$ | $60.6\pm1.9$ | $1\pm0$ | $0.986\pm0.002$ | $0.764\pm0.168$ | $0.750\pm0.166$ | $14\pm2.3$ |
| monk2-l | $1\pm0$ | $0.900\pm0.003$ | $0.416\pm0.155$ | $0.316\pm0.152$ | $100\pm3.2$ | $1\pm0$ | $0.971\pm0.002$ | $0.847\pm0.107$ | $0.818\pm0.105$ | $28.8\pm2.4$ |
| monk2-f | $1\pm0$ | $0.914\pm0.006$ | $0.591\pm0.116$ | $0.505\pm0.112$ | $86\pm6$ | $1\pm0$ | $0.948\pm0.001$ | $0.503\pm0.042$ | $0.451\pm0.042$ | $51.8\pm1.3$ |
| monk3-l | $1\pm0$ | $0.955\pm0.003$ | $0.434\pm0.116$ | $0.388\pm0.115$ | $45.2\pm3.2$ | $1\pm0$ | $0.984\pm0.001$ | $0.812\pm0.076$ | $0.796\pm0.075$ | $16.2\pm1.1$ |
| monk3-f | $1\pm0$ | $0.943\pm0.002$ | $0.729\pm0.042$ | $0.673\pm0.041$ | $56.8\pm2$ | $1\pm0$ | $0.986\pm0.002$ | $0.760\pm0.132$ | $0.747\pm0.132$ | $13.8\pm1.9$ |
| tic-tac-toe | $1\pm0$ | $-0.562\pm0.08$ | $0.446\pm0.142$ | $-1.116\pm0.14$ | $312\pm17$ | $0.961\pm0.006$ | $0.864\pm0.005$ | $0.790\pm0.103$ | $0.693\pm0.103$ | $19.4\pm1$ |
| car-eval | $1\pm0$ | $-2.042\pm0.2$ | $0.307\pm0.206$ | $-2.735\pm0.14$ | $608.4\pm41$ | $0.885\pm0.005$ | $0.817\pm0.006$ | $0.647\pm0.081$ | $0.579\pm0.079$ | $13.6\pm1.8$ |
| nursery | $1\pm0$ | $-90.2\pm2.9$ | $0.063\pm0.125$ | $-91.143\pm2.84$ | $9120\pm290$ | $0.878\pm0.019$ | $0.794\pm0.004$ | $0.652\pm0.100$ | $0.568\pm0.108$ | $8.4\pm1.7$ |
| mushroom | $1\pm0$ | $0.336\pm0.091$ | $0.947\pm0.074$ | $0.283\pm0.077$ | $66.4\pm9$ | $0.974\pm0.011$ | $0.944\pm0.011$ | $0.837\pm0.141$ | $0.807\pm0.141$ | $3\pm0$ |
| kr-vs-kp | $1\pm0$ | $-8.25\pm0.87$ | $0.663\pm0.07$ | $-8.585\pm0.81$ | $924.8\pm87$ | $0.944\pm0.011$ | $0.902\pm0.011$ | $0.929\pm0.043$ | $0.887\pm0.043$ | $4.2\pm0.4$ |
| zoo | $1\pm0$ | $0.984\pm0$ | $0.940\pm0.02$ | $0.925\pm0.02$ | $15.8\pm0.4$ | $1\pm0$ | $0.993\pm0.001$ | $0.960\pm0.058$ | $0.953\pm0.058$ | $7.4\pm0.8$ |
| lymph | $1\pm0$ | $0.364\pm0.014$ | $0.722\pm0.044$ | $0.086\pm0.042$ | $63.6\pm1.356$ | $0.885\pm0.026$ | $0.819\pm0.013$ | $0.776\pm0.054$ | $0.710\pm0.059$ | $6.6\pm1.3$ |
| balance | $1\pm0$ | $-1.52\pm0.195$ | $0.646\pm0.032$ | $-1.87\pm0.172$ | $251\pm19$ | $0.817\pm0.022$ | $0.745\pm0.009$ | $0.373\pm0.163$ | $0.301\pm0.174$ | $7.2\pm1.7$ |

surprising due to the lack of regularisation parameter and the high maximum depth of 20 that we set for a fair comparison.

CART never achieved optimality, in terms of the training objective $\mathcal{H}_\lambda$. Furthermore, it is interesting to note that, even on experiments where BRANCHES did not terminate, and thus did not necessarily find the optimal DT within the allocated 5 minutes of time, it still found better solutions (in terms of the training $\mathcal{H}_\lambda$) than CART. However, solutions with higher training $\mathcal{H}_\lambda$ do not always induce higher test accuracies as evident from monk3-f, tic-tac-toe, car-eval, nursery and kr-vs-kp. On the other hand, they always produce significantly less complex, and thus more interpretable, DTs, which we recall is a major motivation behind employing Decision Tree models. We believe it is very likely that, with large training datasets, the objective metric $\mathcal{H}_\lambda$ is a good indicator of high out-of-sample accuracy and sparsity (number of splits).

### F.2 DEPENDENCE ON $\lambda$

Fig. 7, Fig. 8, Fig. 9, Fig. 10 and Fig. 11 show the dependence of the objective $\mathcal{H}_\lambda$, accuracy, number of splits $\mathcal{S}(T)$, execution times and number of iterations respectively on $\lambda$.

- We did not report $\mathcal{H}_\lambda$ for DL8.5 because it is significantly lower than $\mathcal{H}_\lambda$ of the other algorithms.
- MurTree is missing in some comparisons because it causes the kernel to die due to high memory consumption.
- The missing data points with regard to STreeD are due to its non-anytime behaviour, it does not suggest a DT solution for those $\lambda$ values after the 5 minutes time limit.

Overall, BRANCHES exhibits the best frontier, in terms of $\mathcal{H}_\lambda$, with GOSDT the most competitive method. The execution times frontier of BRANCHES is also better GOSDT's, albeit GOSDT

Table 8: 5 folds cross-validation train/test results for BRANCHES and MurTree. acc refers to Accuracy, obj refers to the objective $\mathcal{H}_\lambda(T)$, splits refers to the number of splits $\mathcal{S}(T)$. The kernel dies for MurTree on kr-vs-kp and lymph.

| Dataset | MurTree | | | | | BRANCHES | | | | |
|---|---|---|---|---|---|---|---|---|---|---|
| | train acc | train obj | test acc | test obj | splits | train acc | train obj | test acc | test obj | splits |
| monk1-l | $1\pm0$ | $0.870\pm0.013$ | $0.820\pm0.182$ | $0.690\pm0.170$ | $13\pm1.3$ | $1\pm0$ | $0.936\pm0.008$ | $0.844\pm0.196$ | $0.780\pm0.188$ | $6.4\pm0.8$ |
| monk1-f | $1\pm0$ | $0.972\pm0.002$ | $0.629\pm0.127$ | $0.602\pm0.126$ | $27.8\pm2$ | $1\pm0$ | $0.986\pm0.002$ | $0.764\pm0.168$ | $0.750\pm0.166$ | $14\pm2.3$ |
| monk2-l | $1\pm0$ | $0.970\pm0.003$ | $0.800\pm0.156$ | $0.770\pm0.154$ | $30\pm2.6$ | $1\pm0$ | $0.971\pm0.002$ | $0.847\pm0.107$ | $0.818\pm0.105$ | $28.8\pm2.4$ |
| monk2-f | $1\pm0$ | $0.944\pm0.002$ | $\mathbf{0.568\pm0.041}$ | $\mathbf{0.512\pm0.040}$ | $56.4\pm2.2$ | $1\pm0$ | $0.948\pm0.001$ | $0.503\pm0.042$ | $0.451\pm0.042$ | $51.8\pm1.3$ |
| monk3-l | $1\pm0$ | $0.975\pm0.005$ | $0.708\pm0.170$ | $0.683\pm0.167$ | $25\pm4.5$ | $1\pm0$ | $0.984\pm0.001$ | $0.812\pm0.076$ | $0.796\pm0.075$ | $16.2\pm1.1$ |
| monk3-f | $1\pm0$ | $0.976\pm0.002$ | $\mathbf{0.778\pm0.079}$ | $\mathbf{0.754\pm0.078}$ | $24.2\pm2$ | $1\pm0$ | $0.986\pm0.002$ | $0.760\pm0.132$ | $0.747\pm0.132$ | $13.8\pm1.9$ |
| tic-tac-toe | $0.961\pm0.006$ | $0.864\pm0.005$ | $0.790\pm0.104$ | $0.693\pm0.105$ | $19.4\pm1$ | $0.961\pm0.006$ | $0.864\pm0.005$ | $0.790\pm0.103$ | $0.693\pm0.103$ | $19.4\pm1$ |
| car-eval | $0.888\pm0.010$ | $0.817\pm0.006$ | $0.647\pm0.081$ | $0.576\pm0.074$ | $14.2\pm2.5$ | $0.885\pm0.005$ | $0.817\pm0.006$ | $0.647\pm0.081$ | $0.579\pm0.079$ | $13.6\pm1.8$ |
| nursery | $0.878\pm0.019$ | $0.794\pm0.004$ | $0.652\pm0.100$ | $0.568\pm0.108$ | $8.4\pm1.7$ | $0.878\pm0.019$ | $0.794\pm0.004$ | $0.652\pm0.100$ | $0.568\pm0.108$ | $8.4\pm1.7$ |
| mushroom | $0.992\pm0.001$ | $0.950\pm0.006$ | $0.831\pm0.172$ | $0.789\pm0.168$ | $4.2\pm0.7$ | $0.974\pm0.011$ | $0.944\pm0.011$ | $0.837\pm0.141$ | $0.807\pm0.141$ | $3\pm0$ |
| kr-vs-kp | — | — | — | — | — | $0.944\pm0.011$ | $0.902\pm0.011$ | $0.929\pm0.043$ | $0.887\pm0.043$ | $4.2\pm0.4$ |
| zoo | $1\pm0$ | $0.990\pm0.001$ | $0.930\pm0.068$ | $0.920\pm0.067$ | $10.2\pm1$ | $1\pm0$ | $0.993\pm0.001$ | $0.960\pm0.058$ | $0.953\pm0.058$ | $7.4\pm0.8$ |
| lymph | — | — | — | — | — | $0.885\pm0.026$ | $0.819\pm0.013$ | $0.776\pm0.054$ | $0.710\pm0.059$ | $6.6\pm1.3$ |
| balance | $0.821\pm0.021$ | $0.745\pm0.009$ | $0.364\pm0.176$ | $0.288\pm0.191$ | $7.6\pm1.8$ | $0.817\pm0.022$ | $0.745\pm0.009$ | $0.373\pm0.163$ | $0.301\pm0.174$ | $7.2\pm1.7$ |

Table 9: 5 folds cross-validation train/test results for BRANCHES and STreeD. acc refers to Accuracy, obj refers to the objective $\mathcal{H}_\lambda(T)$, splits refers to the number of splits $\mathcal{S}(T)$. STreeD reaches timeout and does not suggest a solution for car-eval, nursery, lymph and balance.

| Dataset | STreeD | | | | | BRANCHES | | | | |
|---|---|---|---|---|---|---|---|---|---|---|
| | train acc | train obj | test acc | test obj | splits | train acc | train obj | test acc | test obj | splits |
| monk1-l | $1\pm0$ | $0.936\pm0.008$ | $0.844\pm0.196$ | $0.780\pm0.188$ | $6.4\pm0.8$ | $1\pm0$ | $0.936\pm0.008$ | $0.844\pm0.196$ | $0.780\pm0.188$ | $6.4\pm0.8$ |
| monk1-f | $1\pm0$ | $0.986\pm0.002$ | $0.772\pm0.177$ | $0.758\pm0.176$ | $14\pm2.3$ | $1\pm0$ | $0.986\pm0.002$ | $0.764\pm0.168$ | $0.750\pm0.166$ | $14\pm2.3$ |
| monk2-l | $1\pm0$ | $0.971\pm0.002$ | $0.799\pm0.099$ | $0.771\pm0.096$ | $28.8\pm2.4$ | $1\pm0$ | $0.971\pm0.002$ | $0.847\pm0.107$ | $0.818\pm0.105$ | $28.8\pm2.4$ |
| monk2-f | $1\pm0$ | $0.948\pm0.001$ | $0.515\pm0.057$ | $0.464\pm0.057$ | $51.8\pm1.3$ | $1\pm0$ | $0.948\pm0.001$ | $0.503\pm0.042$ | $0.451\pm0.042$ | $51.8\pm1.3$ |
| monk3-l | $1\pm0$ | $0.984\pm0.001$ | $0.812\pm0.076$ | $0.796\pm0.075$ | $16.2\pm1$ | $1\pm0$ | $0.984\pm0.001$ | $0.812\pm0.076$ | $0.796\pm0.075$ | $16.2\pm1.1$ |
| monk3-f | $1\pm0$ | $0.986\pm0.002$ | $0.835\pm0.115$ | $0.822\pm0.114$ | $13.8\pm1.9$ | $1\pm0$ | $0.986\pm0.002$ | $0.760\pm0.132$ | $0.747\pm0.132$ | $13.8\pm1.9$ |
| tic-tac-toe | $0.961\pm0.006$ | $0.864\pm0.005$ | $0.802\pm0.107$ | $0.705\pm0.108$ | $19.4\pm1$ | $0.961\pm0.006$ | $0.864\pm0.005$ | $0.790\pm0.103$ | $0.693\pm0.103$ | $19.4\pm1$ |
| car-eval | — | — | — | — | — | $0.885\pm0.005$ | $0.817\pm0.006$ | $0.647\pm0.081$ | $0.579\pm0.079$ | $13.6\pm1.8$ |
| nursery | — | — | — | — | — | $0.878\pm0.019$ | $0.794\pm0.004$ | $0.652\pm0.100$ | $0.568\pm0.108$ | $8.4\pm1.7$ |
| mushroom | $0.990\pm0.003$ | $0.950\pm0.006$ | $0.830\pm0.172$ | $0.790\pm0.169$ | $4\pm0.6$ | $0.974\pm0.011$ | $0.944\pm0.011$ | $0.837\pm0.141$ | $0.807\pm0.141$ | $3\pm0$ |
| kr-vs-kp | — | — | — | — | — | $0.944\pm0.011$ | $0.902\pm0.011$ | $0.929\pm0.043$ | $0.887\pm0.043$ | $4.2\pm0.4$ |
| zoo | $1\pm0$ | $0.993\pm0.001$ | $0.940\pm0.058$ | $0.933\pm0.058$ | $7.4\pm0.8$ | $1\pm0$ | $0.993\pm0.001$ | $0.960\pm0.058$ | $0.953\pm0.058$ | $7.4\pm0.8$ |
| lymph | — | — | — | — | — | $0.885\pm0.026$ | $0.819\pm0.013$ | $0.776\pm0.054$ | $0.710\pm0.059$ | $6.6\pm1.3$ |
| balance | — | — | — | — | — | $0.817\pm0.022$ | $0.745\pm0.009$ | $0.373\pm0.163$ | $0.301\pm0.174$ | $7.2\pm1.7$ |

outperforms BRANCHES on a few: care-eval and balance. In terms of the number of iterations, BRANCHES clearly outperforms GOSDT on all experiments showing better computational efficiency and validating our computational complexity analysis of Section 5.

### F.3 CHOOSING $\lambda$

The $\lambda$ values, in Table 5, were chosen through experimentation to yield well behaved DTs in terms of accuracy and sparsity. A principled approach to choosing adequate $\lambda$ values is to estimate suitable metrics through crossvalidation and choosing $\lambda$ accordingly. Fig. 12, Fig. 13, Fig. 14, Fig. 15, Fig. 16, Fig. 17, Fig. 18, Fig. 19, Fig. 20, , Fig. 21, Fig. 22, and Fig. 23 show quartile plots of the different metrics of interest induced by the 5 fold crossvalidation, these figures can be employed to choose adequate $\lambda$ values.

Table 10: 5 folds cross-validation train/test results for BRANCHES and CART. acc refers to Accuracy, obj refers to the objective $\mathcal{H}_\lambda(T)$, splits refers to the number of splits $\mathcal{S}(T)$.

| Dataset | CART | | | | | BRANCHES | | | | |
|---|---|---|---|---|---|---|---|---|---|---|
| | train acc | train obj | test acc | test obj | splits | train acc | train obj | test acc | test obj | splits |
| monk1-l | 0.982 ± 0.027 | 0.890 ± 0.043 | 0.740 ± 0.153 | 0.648 ± 0.151 | 9.2 ± 3.1 | **1 ± 0** | **0.936 ± 0.008** | **0.844 ± 0.196** | **0.780 ± 0.188** | **6.4 ± 0.8** |
| monk1-f | **1 ± 0** | 0.978 ± 0.006 | 0.676 ± 0.109 | 0.654 ± 0.108 | 22.2 ± 6.4 | **1 ± 0** | **0.986 ± 0.002** | **0.764 ± 0.168** | **0.750 ± 0.166** | **14 ± 2.3** |
| monk2-l | **1 ± 0** | 0.952 ± 0.005 | 0.645 ± 0.103 | 0.597 ± 0.102 | 47.8 ± 5.1 | **1 ± 0** | **0.971 ± 0.002** | **0.847 ± 0.107** | **0.818 ± 0.105** | **28.8 ± 2.4** |
| monk2-f | **1 ± 0** | 0.931 ± 0.005 | 0.450 ± 0.089 | 0.381 ± 0.089 | 69.2 ± 5 | **1 ± 0** | **0.948 ± 0.001** | **0.503 ± 0.042** | **0.451 ± 0.042** | **51.8 ± 1.3** |
| monk3-l | **1 ± 0** | 0.981 ± 0.002 | 0.787 ± 0.055 | 0.768 ± 0.056 | 19.2 ± 1.8 | **1 ± 0** | **0.984 ± 0.001** | **0.812 ± 0.076** | **0.796 ± 0.075** | **16.2 ± 1.1** |
| monk3-f | **1 ± 0** | 0.983 ± 0.003 | **0.844 ± 0.093** | **0.827 ± 0.091** | 16.6 ± 3.5 | **1 ± 0** | **0.986 ± 0.002** | 0.760 ± 0.132 | 0.747 ± 0.132 | **13.8 ± 1.9** |
| tic-tac-toe | 0.960 ± 0.009 | 0.843 ± 0.008 | **0.846 ± 0.093** | **0.729 ± 0.083** | 23.4 ± 2.5 | **0.961 ± 0.006** | **0.864 ± 0.005** | 0.790 ± 0.103 | 0.693 ± 0.103 | **19.4 ± 1** |
| car-eval | **0.896 ± 0.006** | 0.800 ± 0.015 | **0.686 ± 0.083** | **0.590 ± 0.085** | 19.2 ± 3.5 | 0.885 ± 0.005 | **0.817 ± 0.006** | 0.647 ± 0.081 | 0.579 ± 0.079 | **13.6 ± 1.8** |
| nursery | **0.883 ± 0.015** | 0.783 ± 0.011 | **0.668 ± 0.106** | **0.568 ± 0.105** | 10 ± 0.6 | 0.878 ± 0.019 | **0.794 ± 0.004** | 0.652 ± 0.100 | **0.568 ± 0.108** | **8.4 ± 1.7** |
| mushroom | **0.988 ± 0.006** | 0.940 ± 0.012 | **0.870 ± 0.142** | **0.822 ± 0.143** | 4.8 ± 1.1 | 0.974 ± 0.011 | **0.944 ± 0.011** | 0.837 ± 0.141 | 0.807 ± 0.141 | **3 ± 0** |
| kr-vs-kp | **0.964 ± 0.006** | 0.890 ± 0.018 | **0.942 ± 0.046** | 0.868 ± 0.029 | 7.4 ± 1.7 | 0.944 ± 0.011 | **0.902 ± 0.011** | 0.929 ± 0.043 | **0.887 ± 0.043** | **4.2 ± 0.4** |
| zoo | **1 ± 0** | 0.992 ± 0.001 | 0.940 ± 0.049 | 0.932 ± 0.048 | 7.8 ± 1.1 | **1 ± 0** | **0.993 ± 0.001** | **0.960 ± 0.058** | **0.953 ± 0.058** | **7.4 ± 0.8** |
| lymph | **0.971 ± 0.012** | 0.809 ± 0.017 | 0.750 ± 0.018 | 0.588 ± 0.015 | 16.2 ± 1.9 | 0.885 ± 0.026 | **0.819 ± 0.013** | **0.776 ± 0.054** | **0.710 ± 0.059** | **6.6 ± 1.3** |
| balance | 0.782 ± 0.022 | 0.712 ± 0.011 | 0.347 ± 0.109 | 0.277 ± 0.120 | **7 ± 1.5** | **0.817 ± 0.022** | **0.745 ± 0.009** | **0.373 ± 0.163** | **0.301 ± 0.174** | 7.2 ± 1.7 |

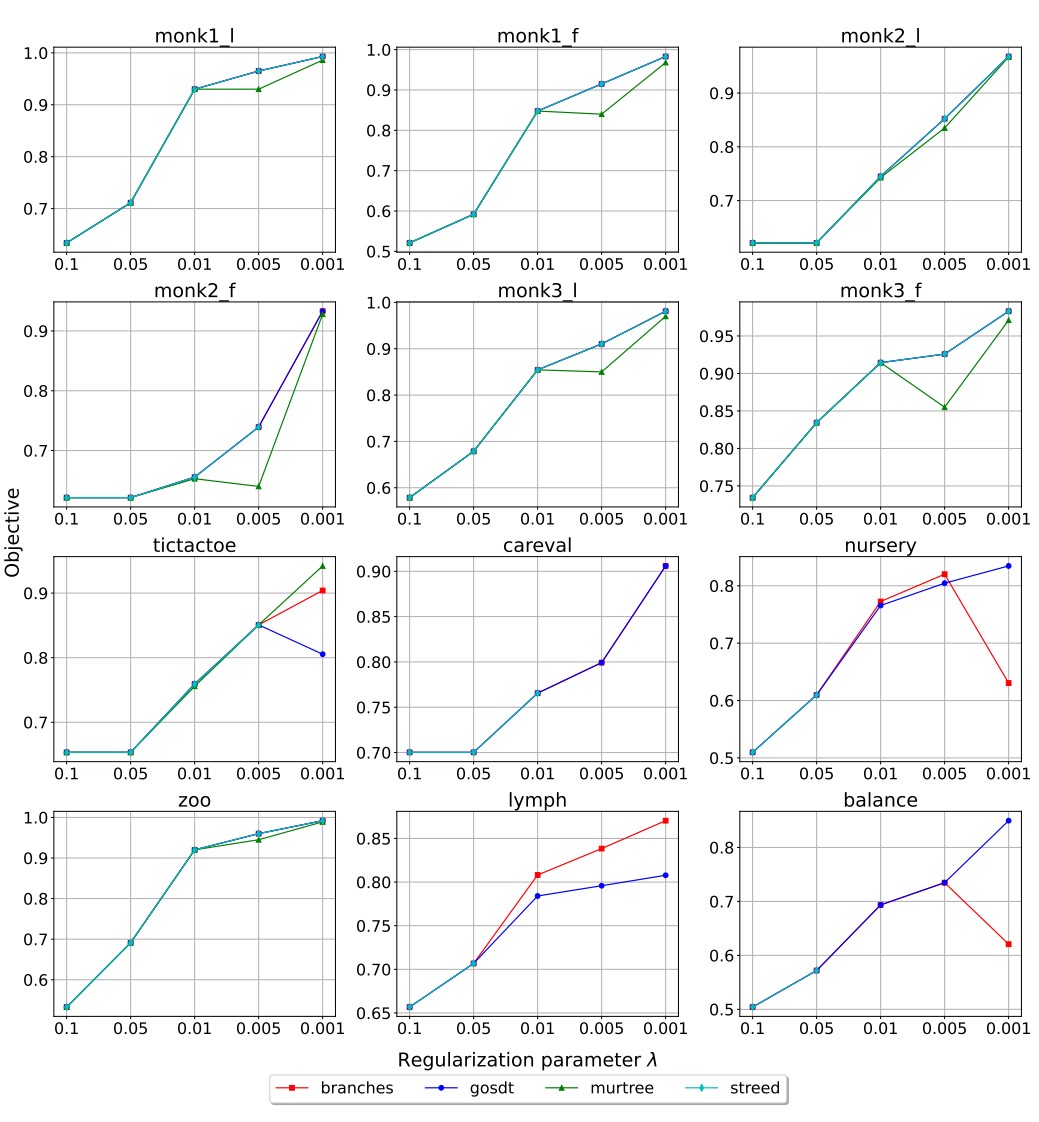

Figure 7: Dependence of the objective $\mathcal{H}_\lambda$ on $\lambda$.

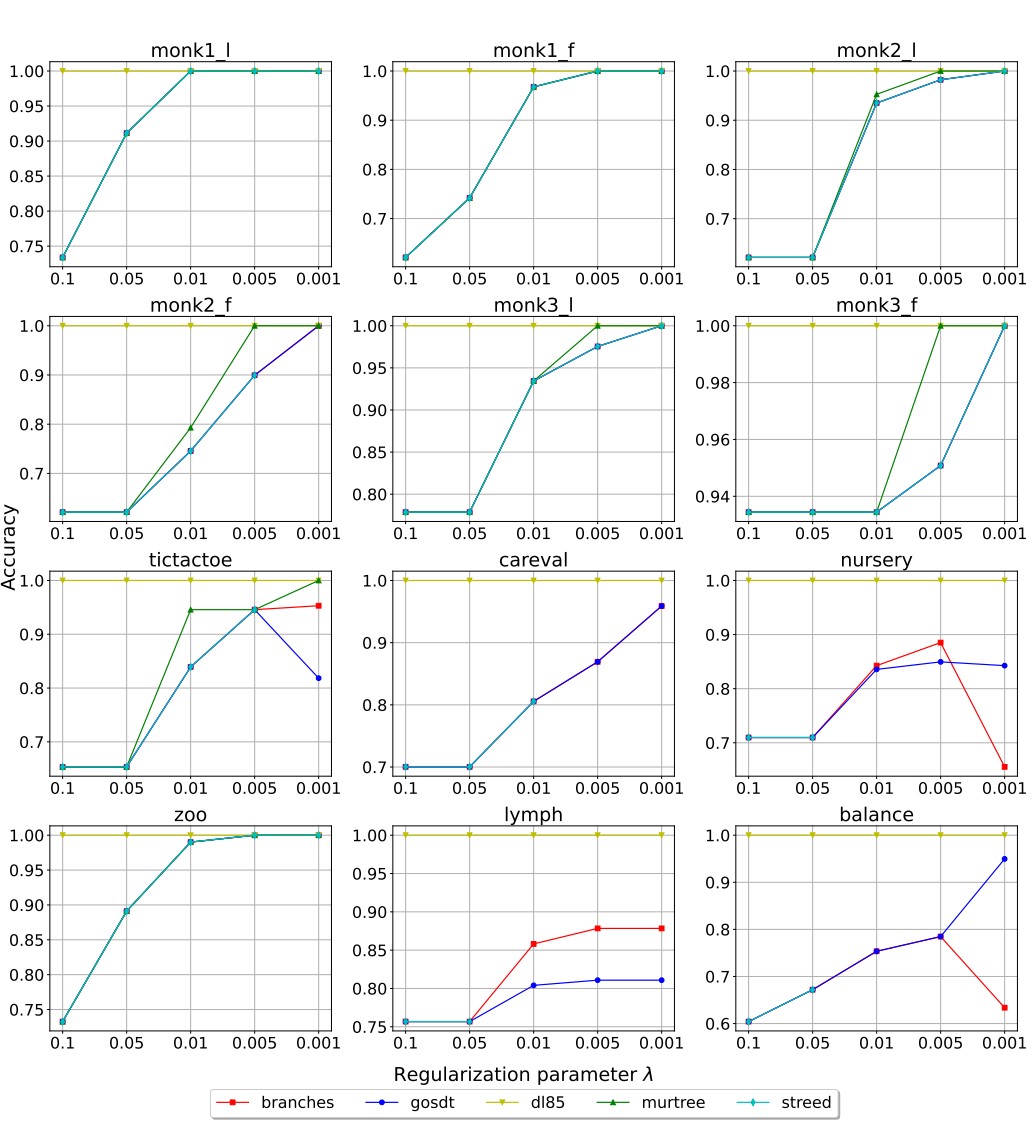

Figure 8: Dependence of the accuracy on $\lambda$.

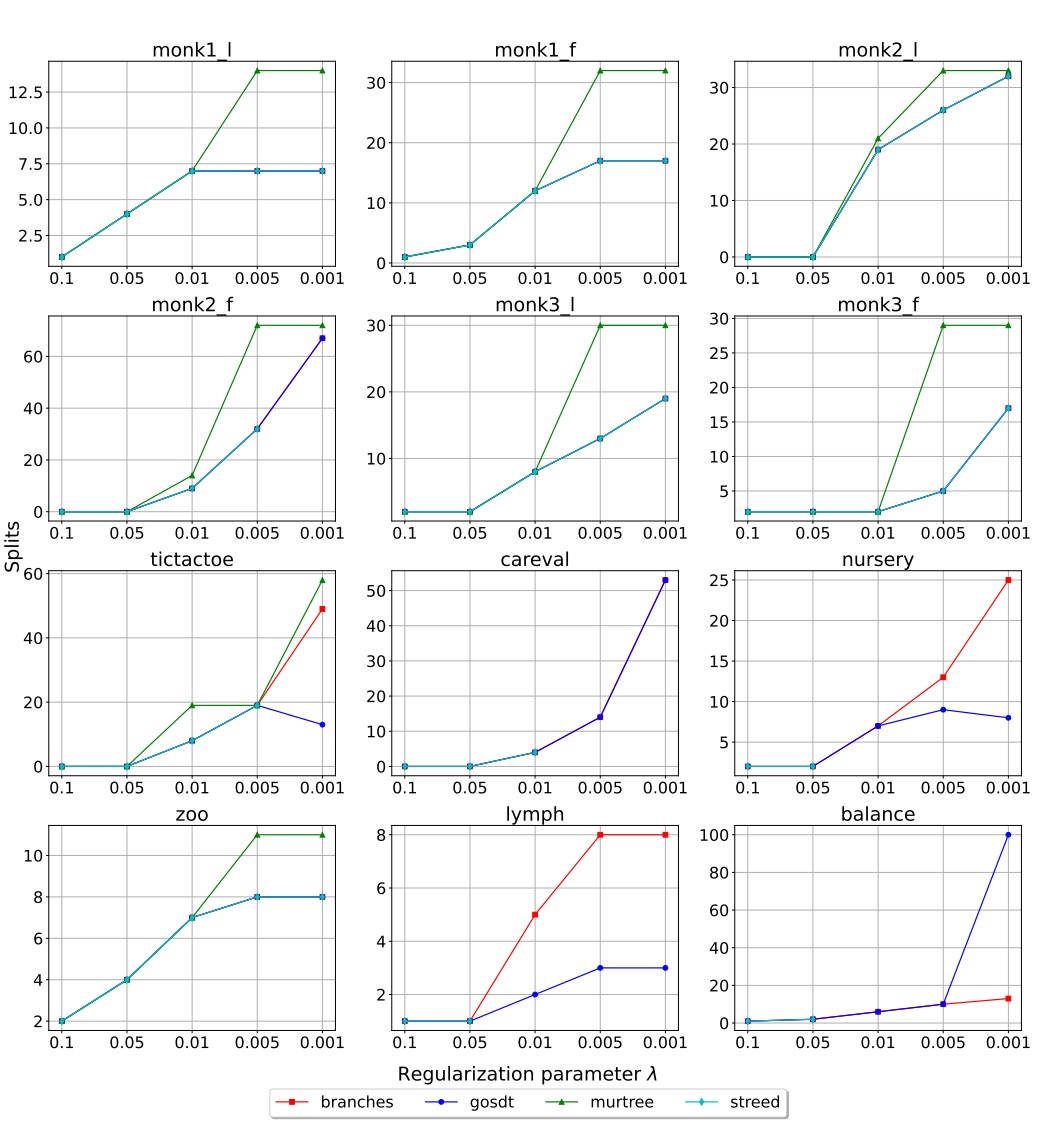

Figure 9: Dependence of the number of splits $\mathcal{S}\left(T\right)$ on $\lambda$.

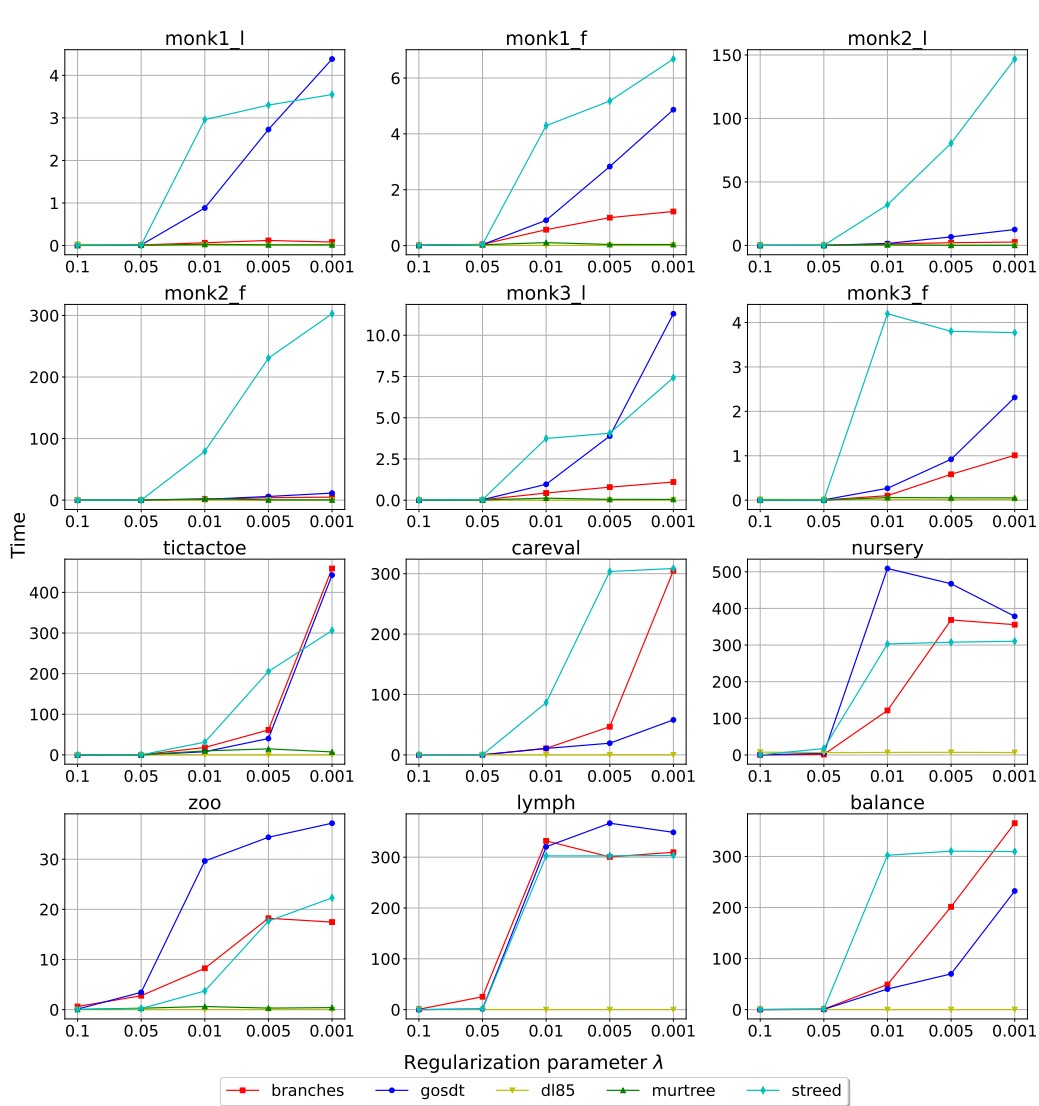

Figure 10: Dependence of the execution times on $\lambda$.

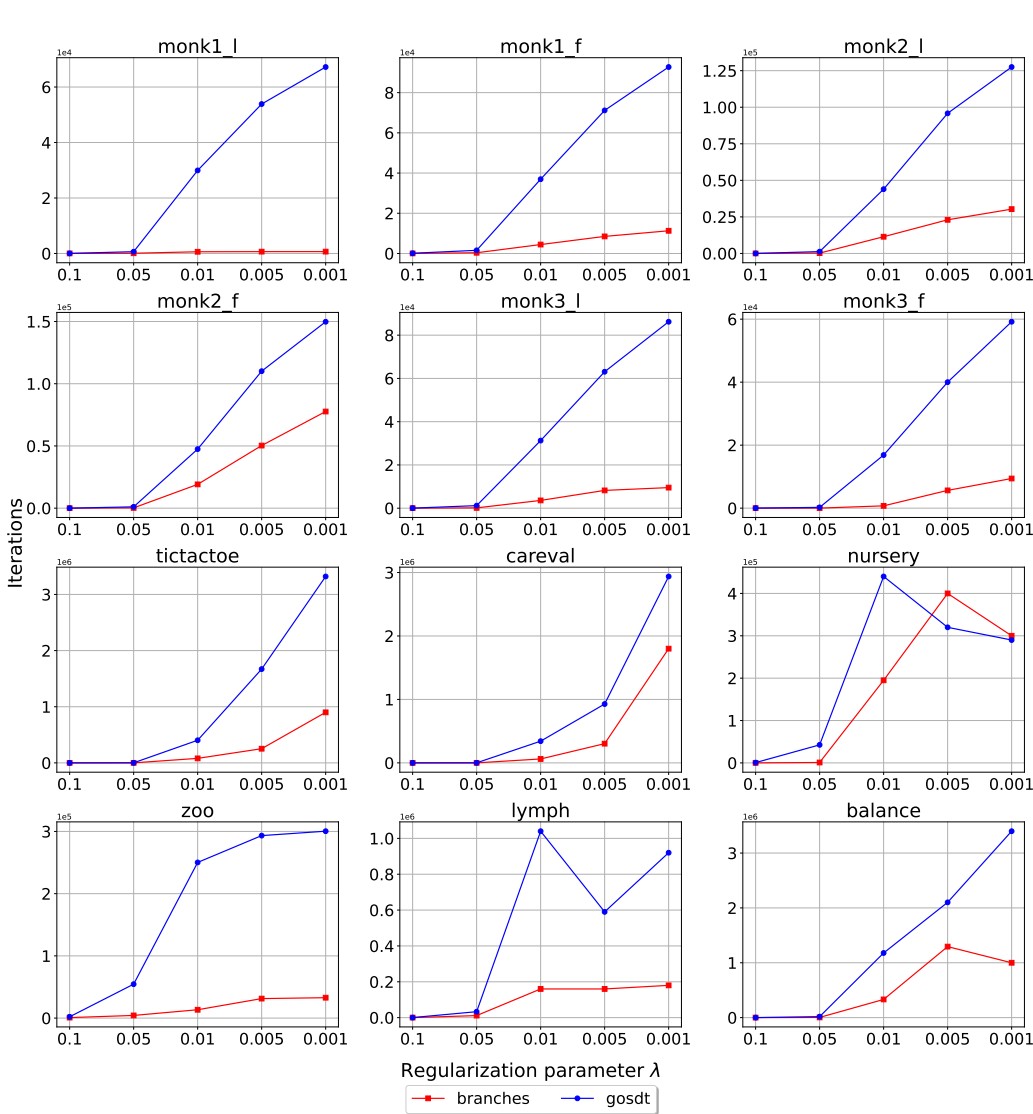

Figure 11: Dependence of the number of iterations on $\lambda$.

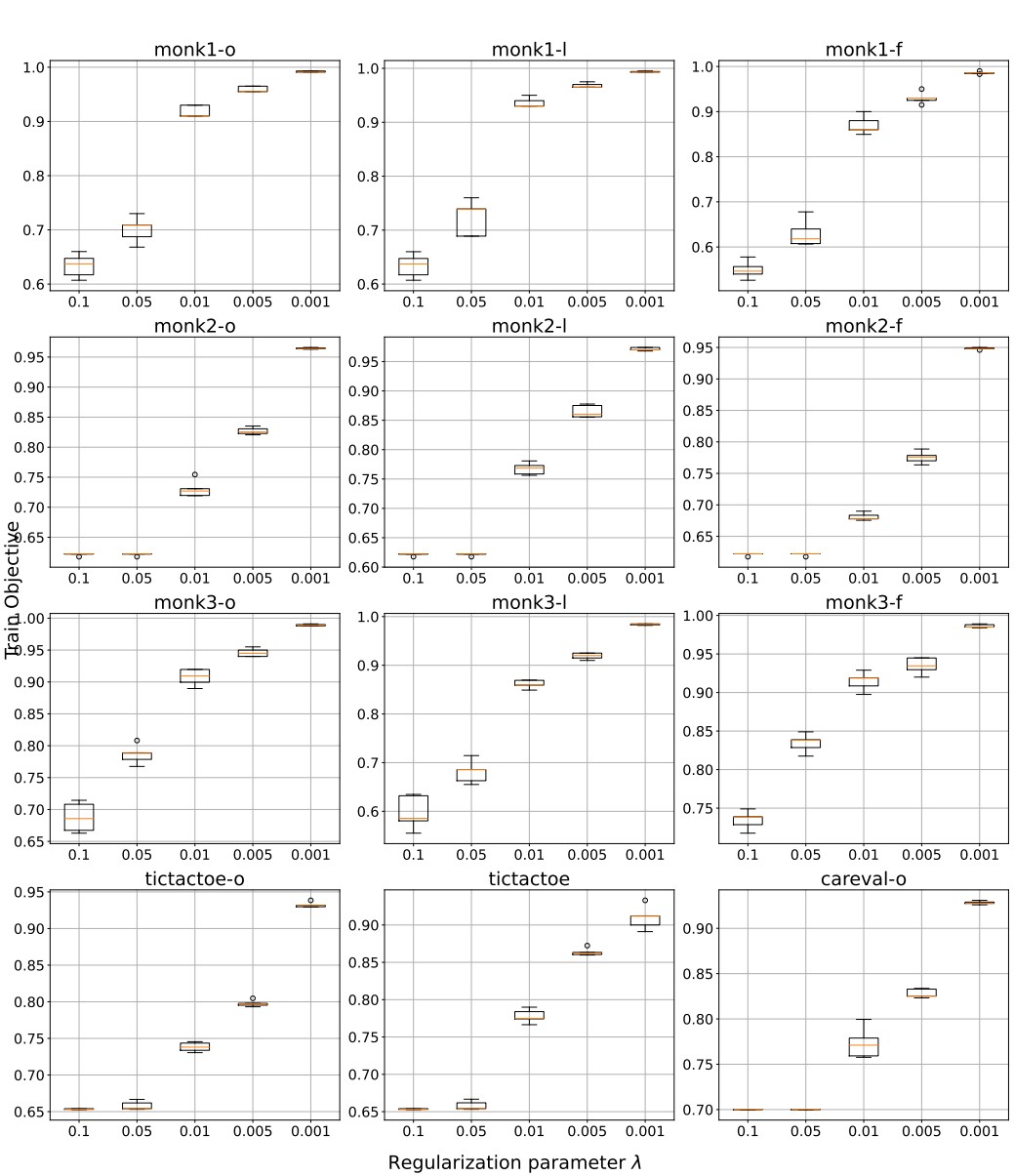

Figure 12: 5 fold crossvalidation of BRANCHES for the training objective $\mathcal{H}_\lambda$.

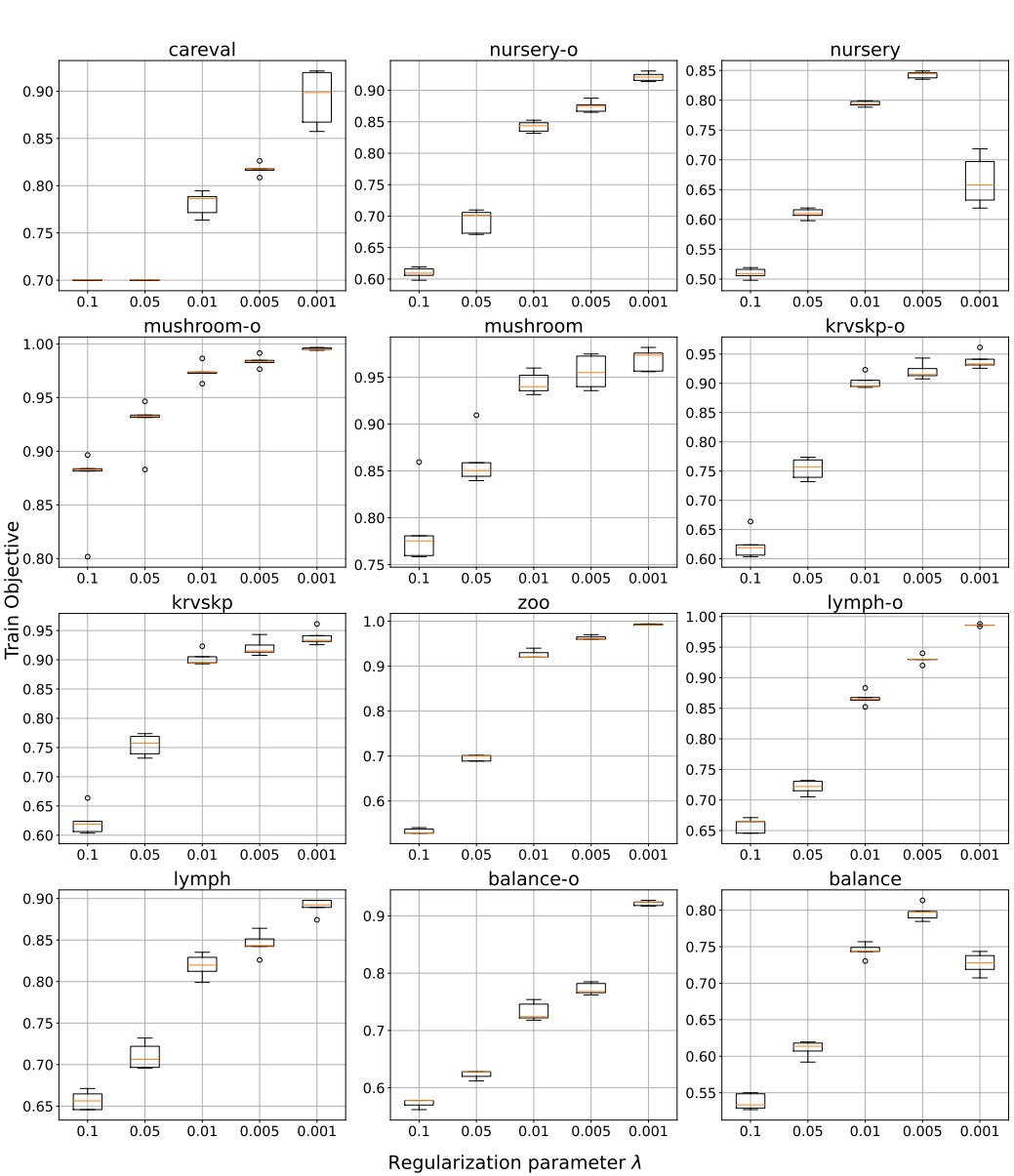

Figure 13: 5 fold crossvalidation of BRANCHES for the training objective $\mathcal{H}_\lambda$.

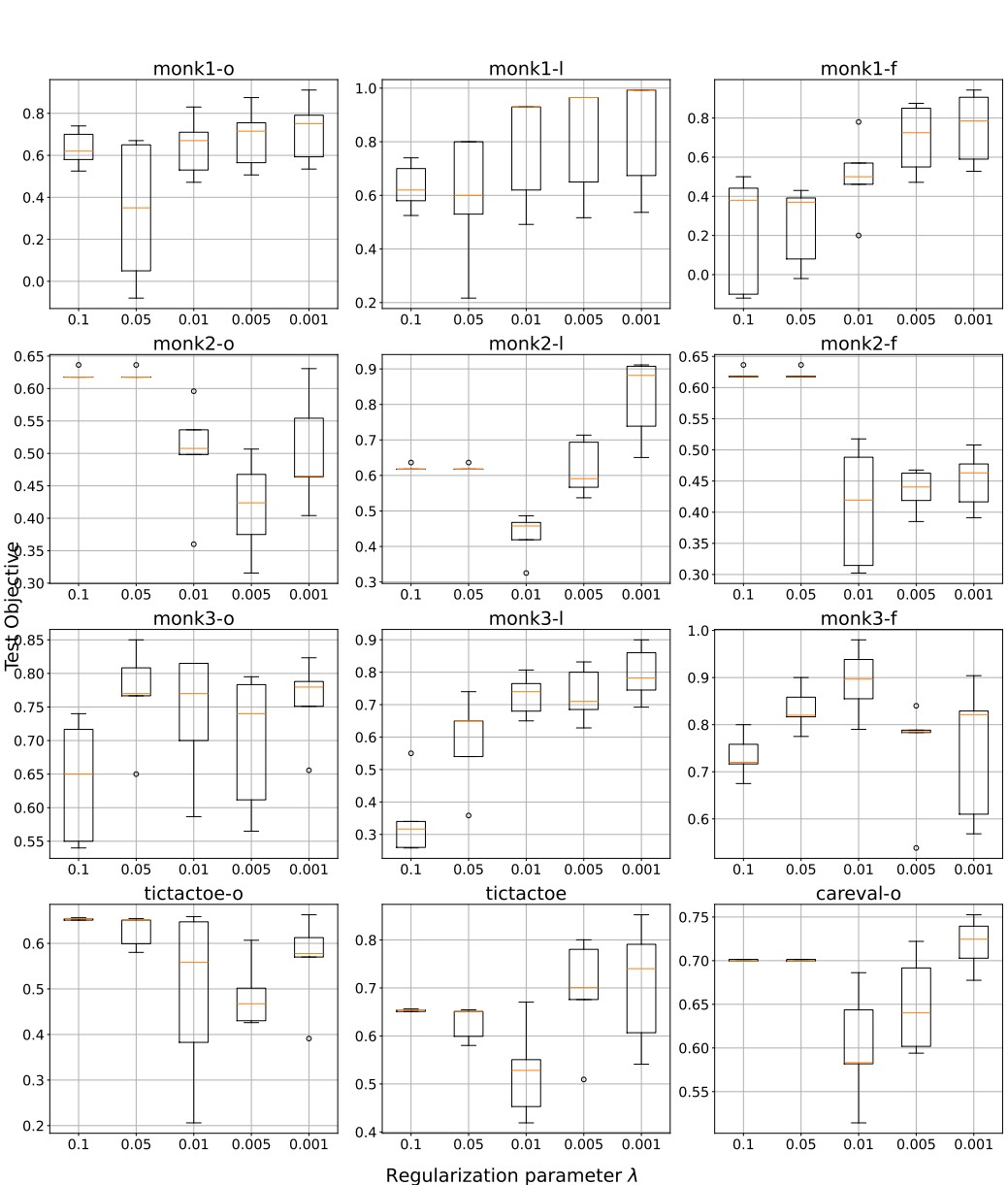

Figure 14: 5 fold crossvalidation of BRANCHES for the test objective $\mathcal{H}_\lambda$.

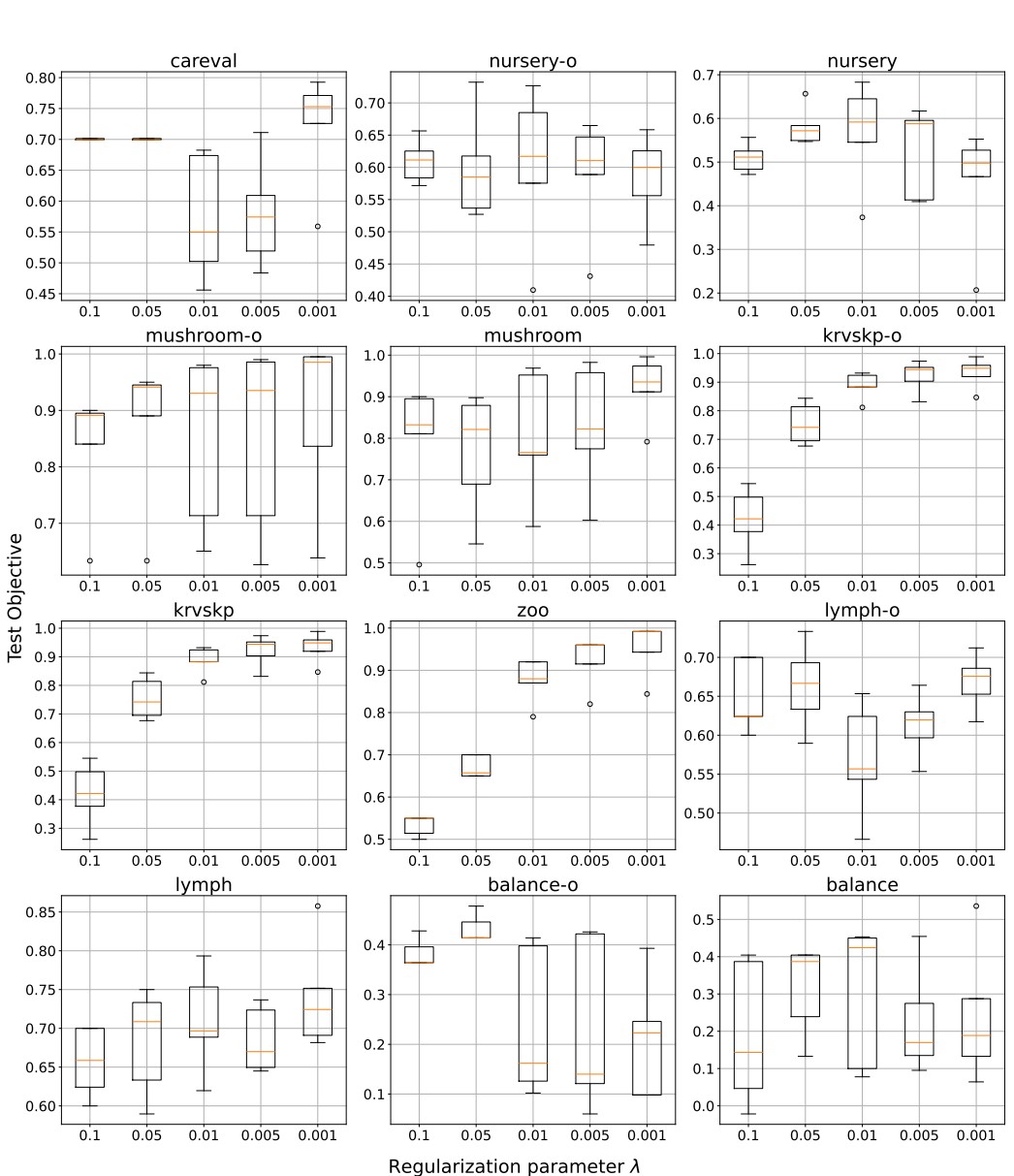

Figure 15: 5 fold crossvalidation of BRANCHES for the test objective $\mathcal{H}_\lambda$.

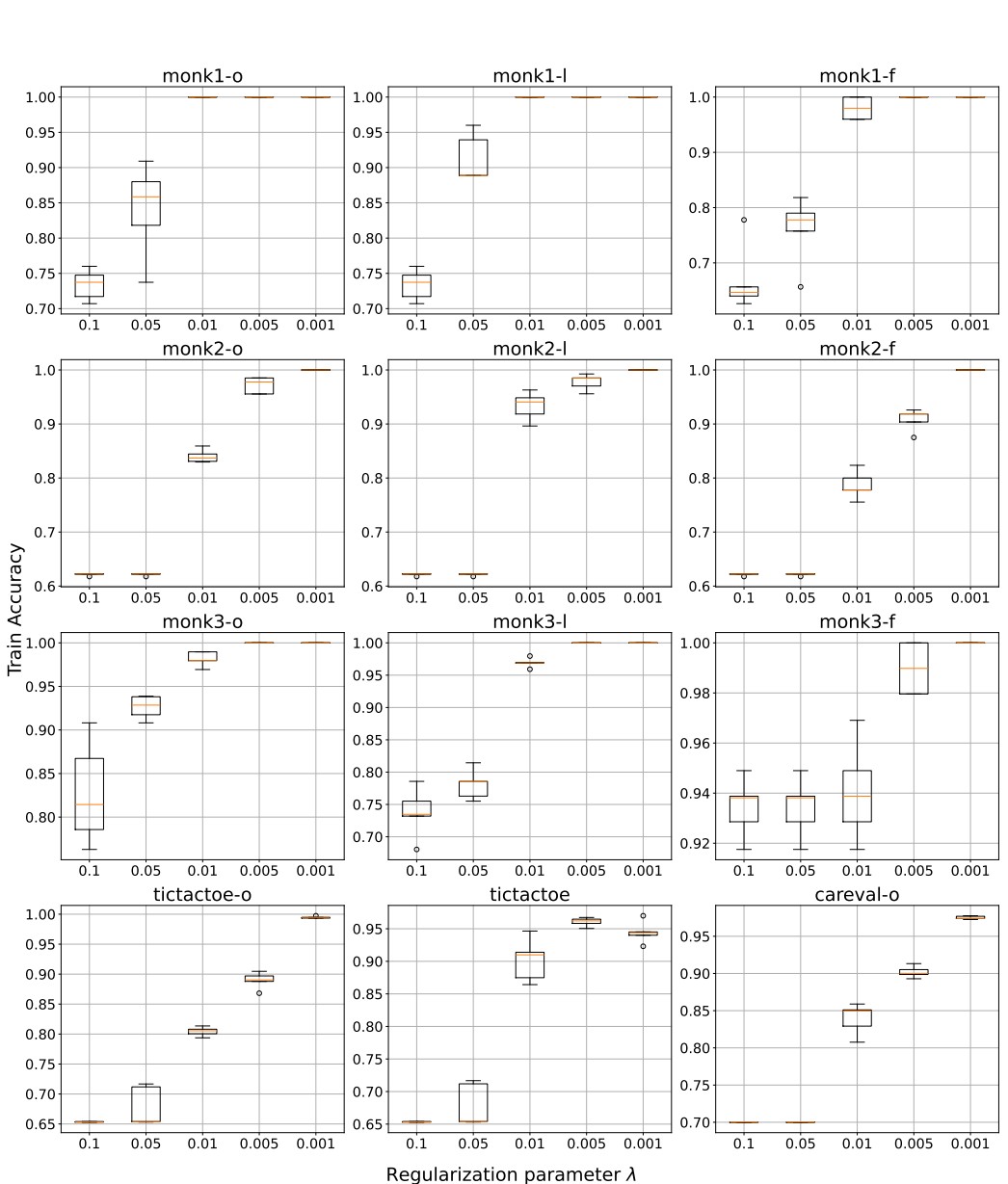

Figure 16: 5 fold crossvalidation of BRANCHES for the training accuracy.

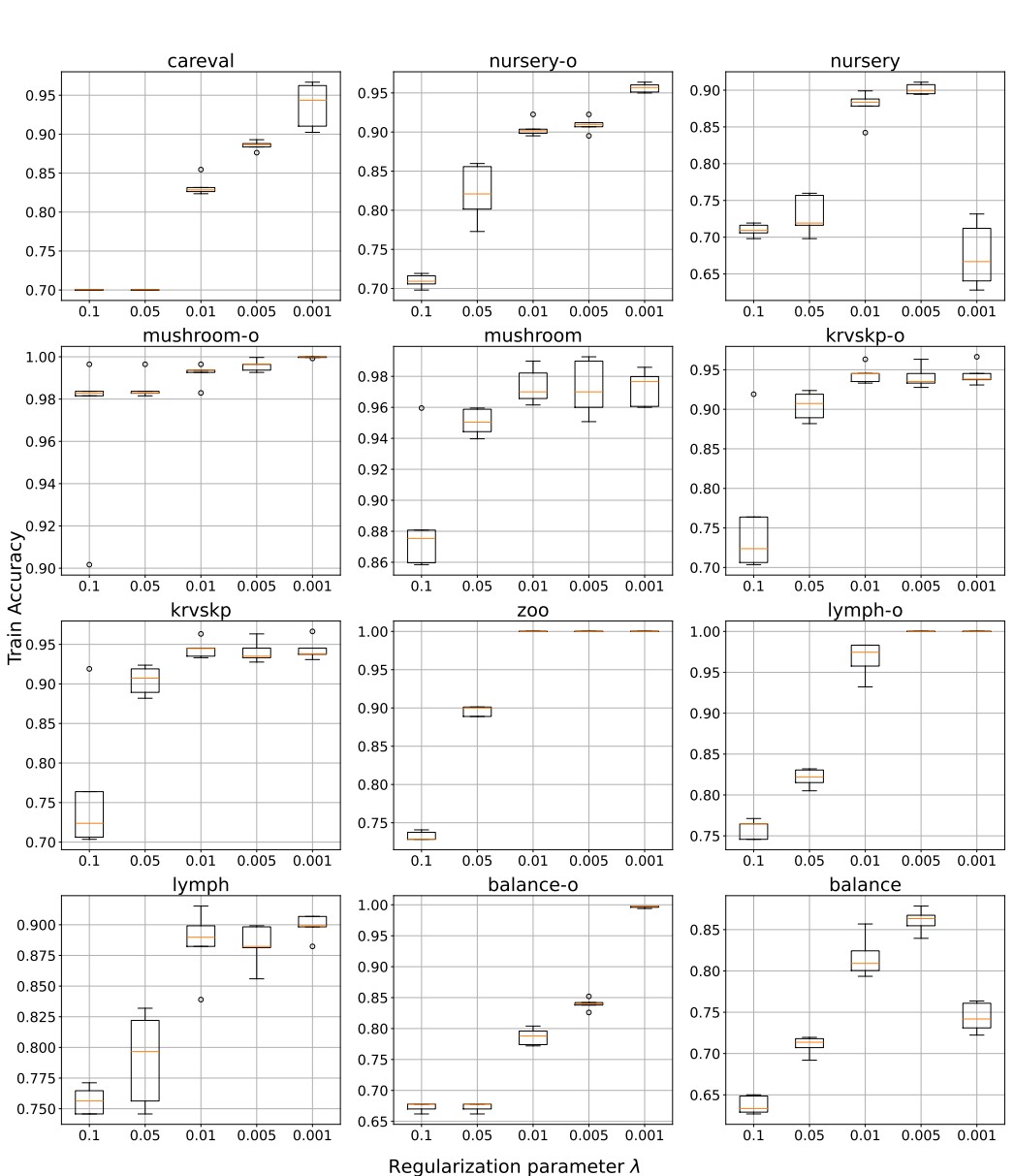

Figure 17: 5 fold crossvalidation of BRANCHES for the training accuracy.

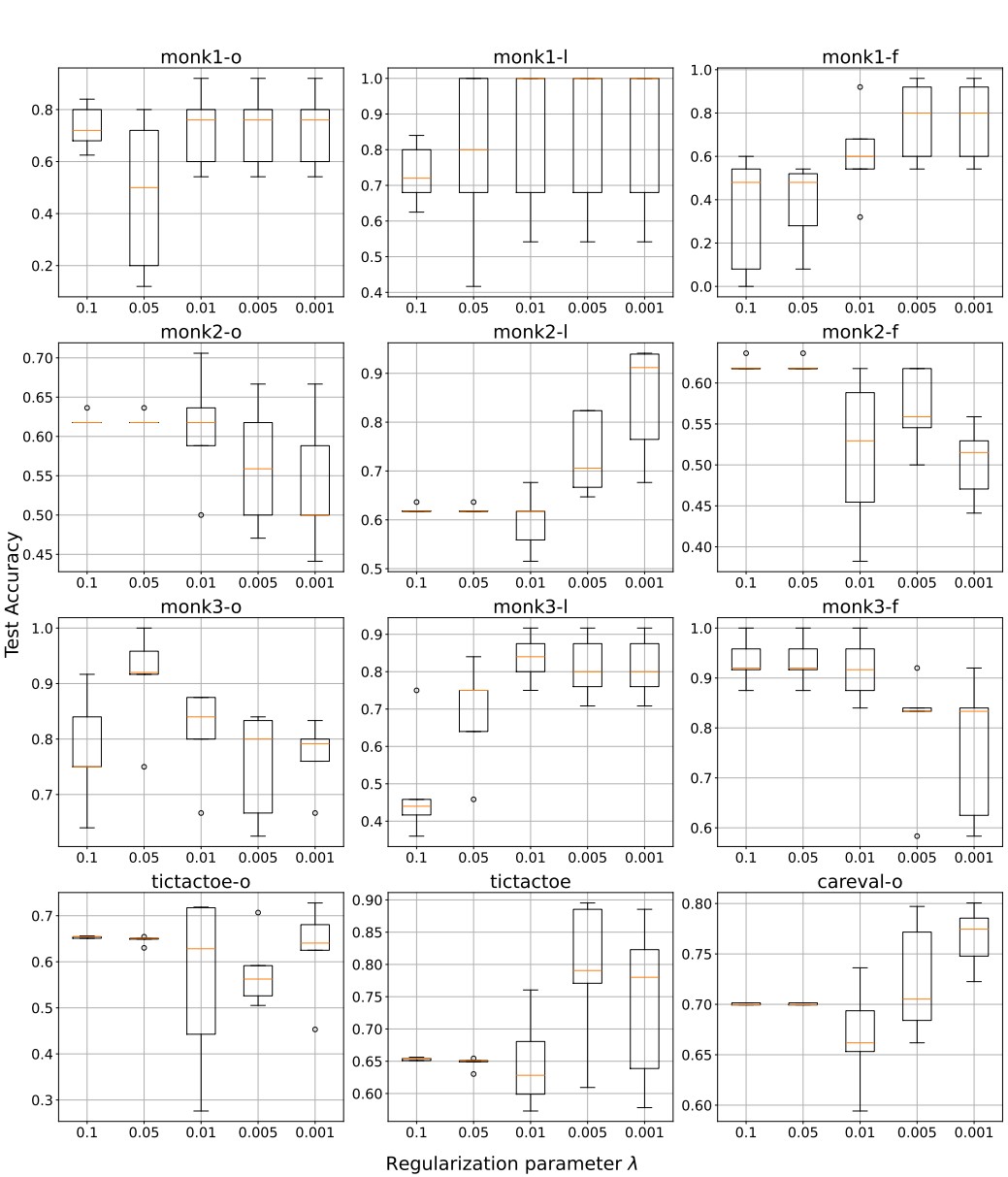

Figure 18: 5 fold crossvalidation of BRANCHES for the test accuracy.

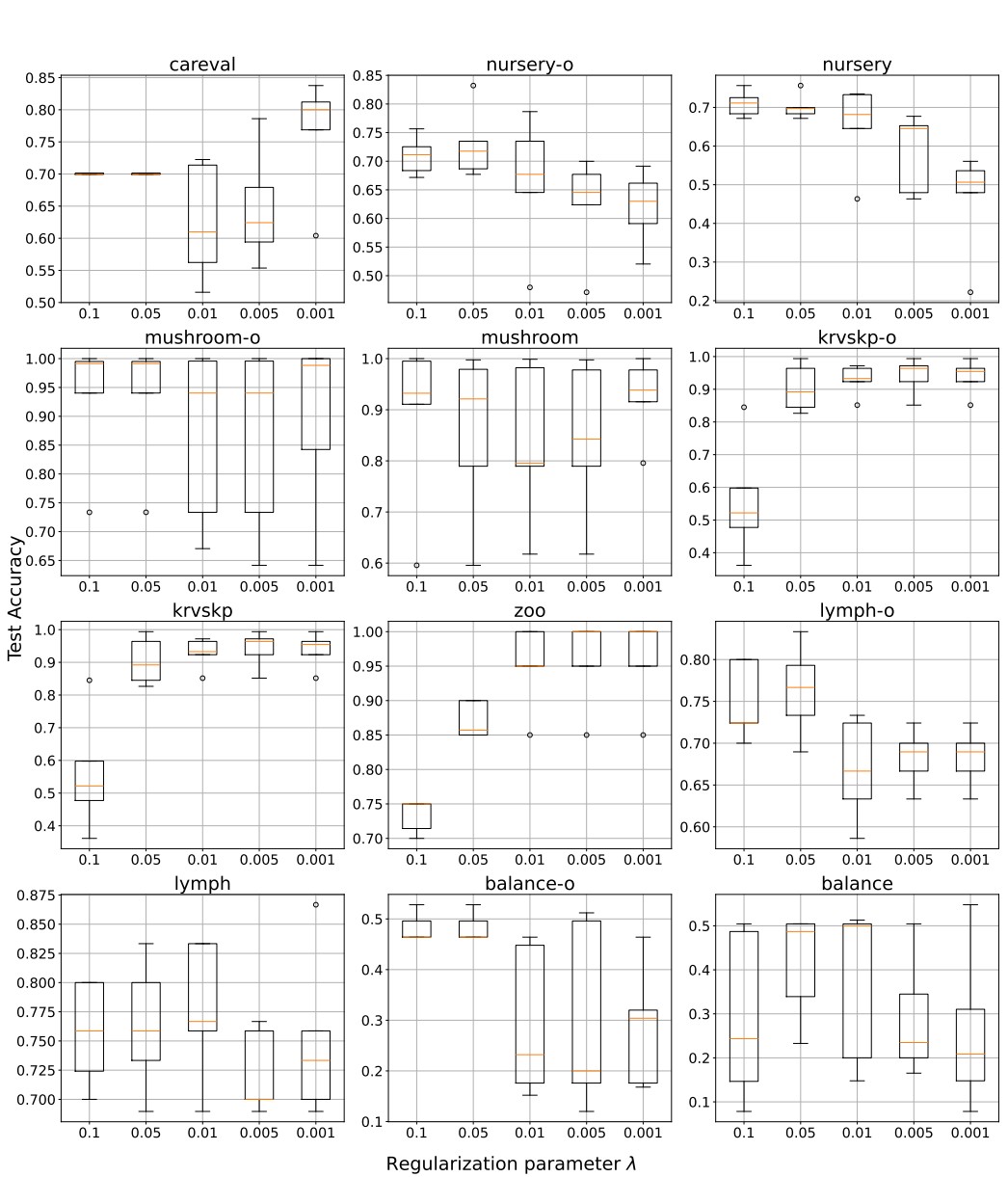

Figure 19: 5 fold crossvalidation of BRANCHES for the test accuracy.

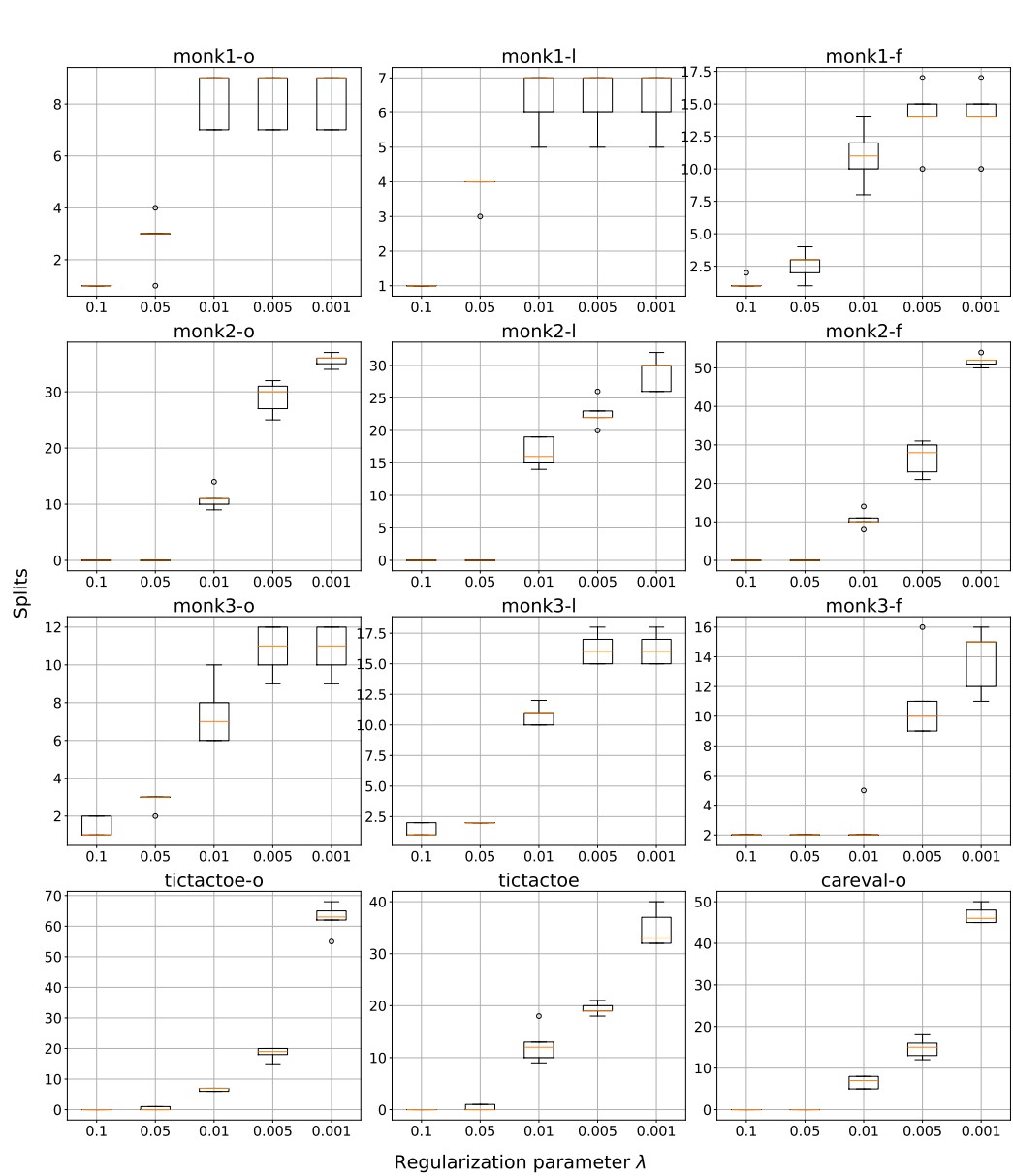

Figure 20: 5 fold crossvalidation of BRANCHES for the number of splits $\mathcal{S}(T)$.

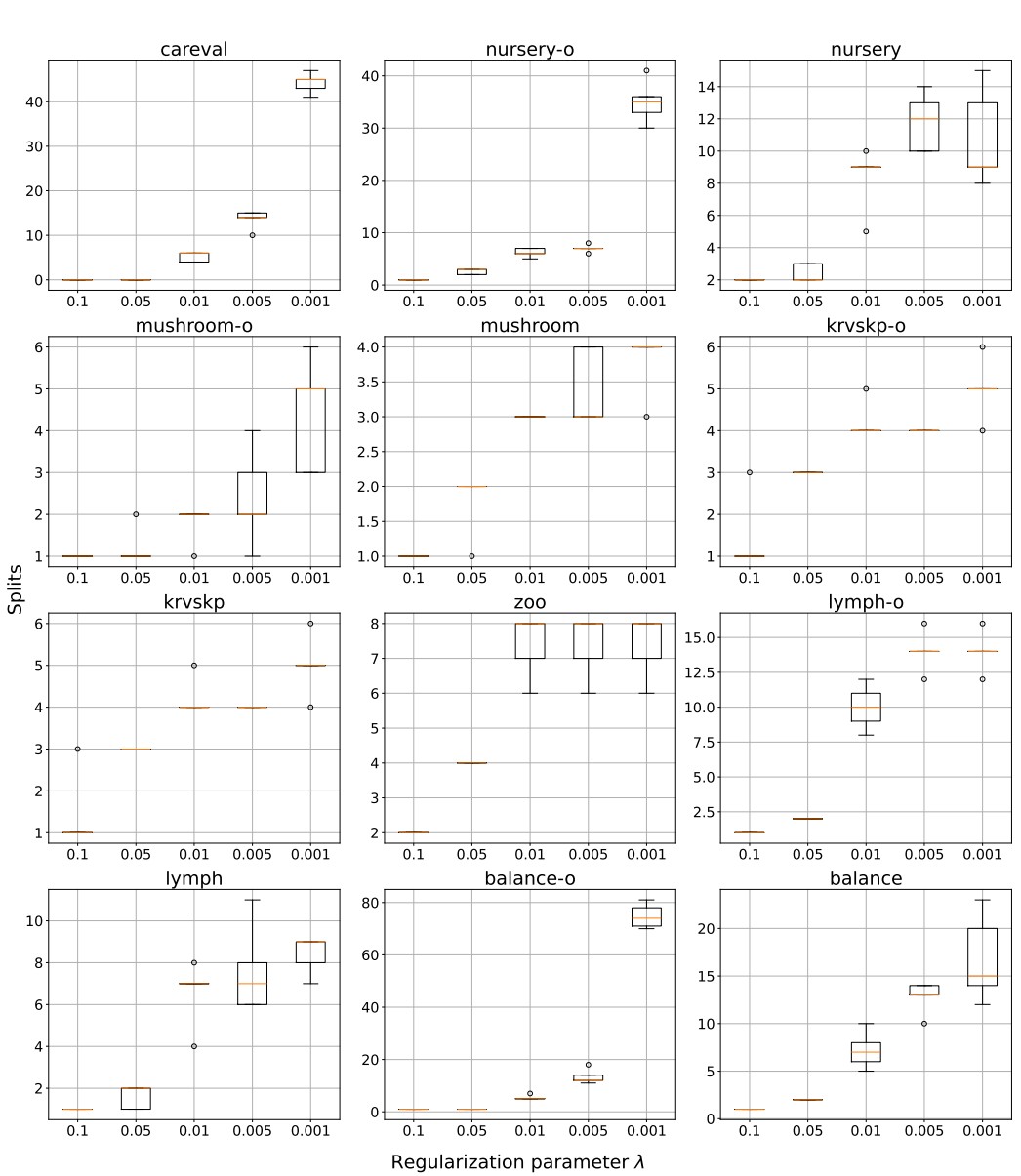

Figure 21: 5 fold crossvalidation of BRANCHES for the number of splits $\mathcal{S}(T)$.

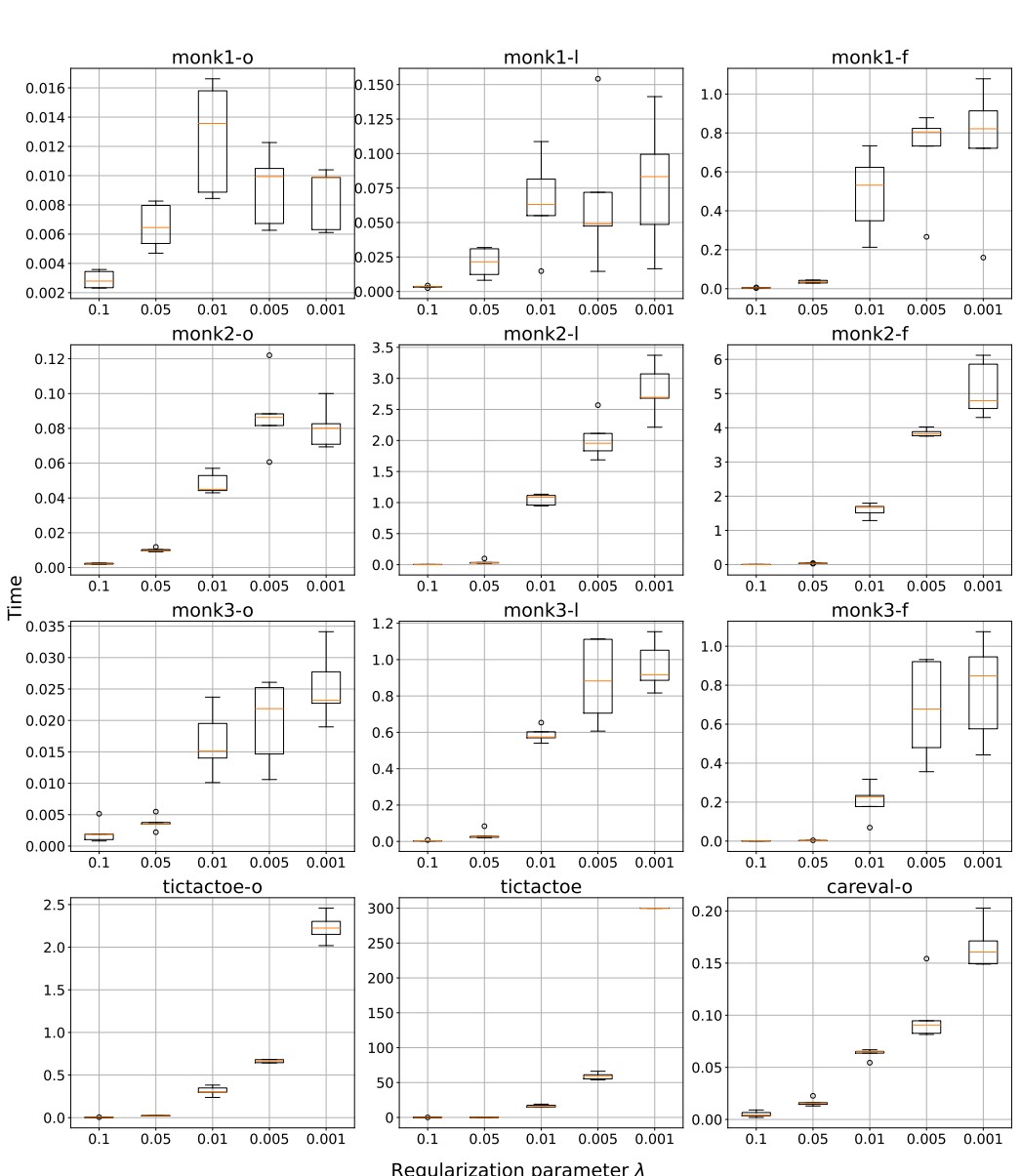

Figure 22: 5 fold crossvalidation of BRANCHES for the execution time.

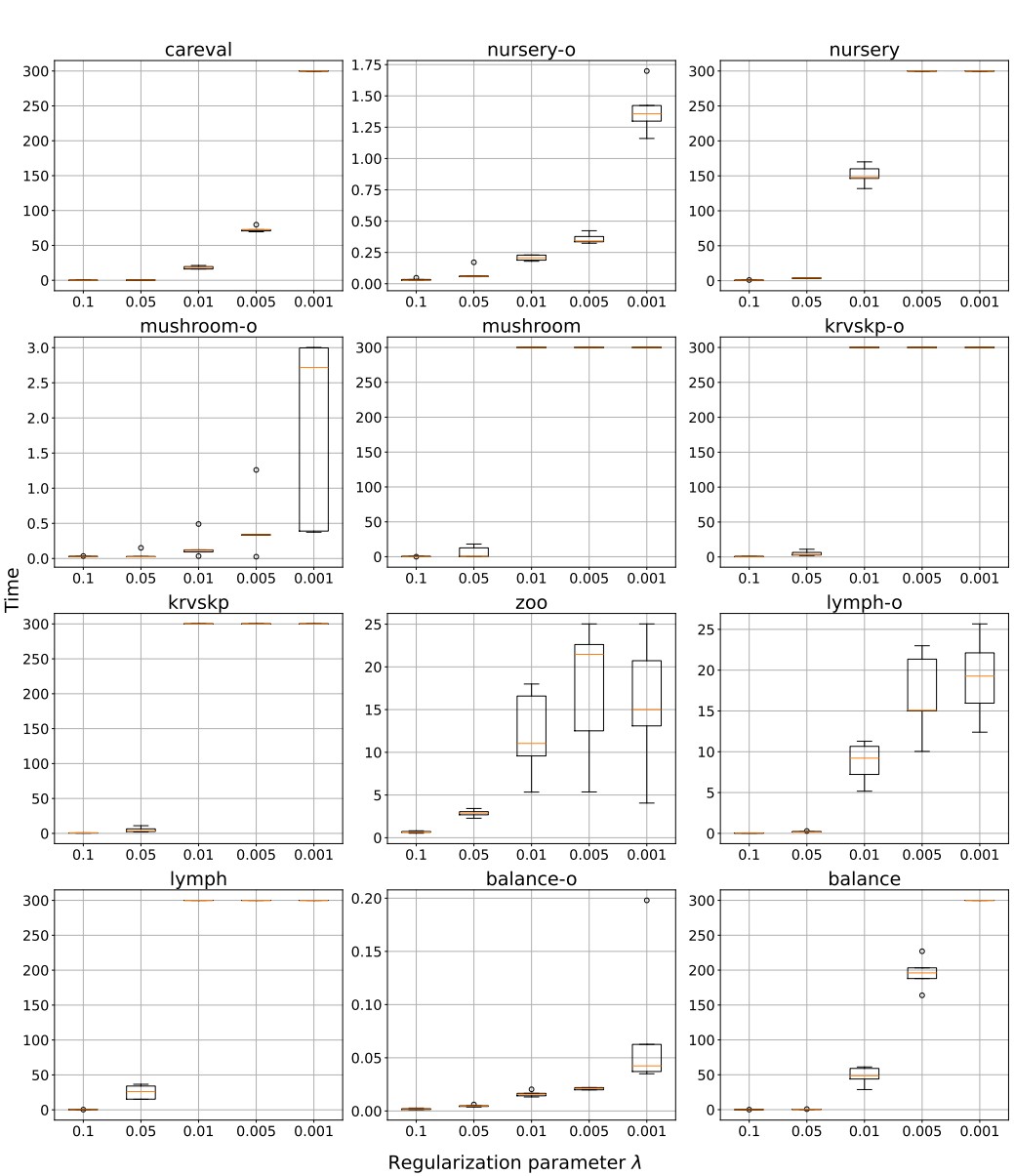

Figure 23: 5 fold crossvalidation of BRANCHES for the execution time.

