# OpenReview forum: "Branches: A Fast Dynamic Programming and Branch & Bound Algorithm for Optimal Decision Trees"
_ICLR.cc/2025/Conference — Submitted to ICLR 2025_

### Official Review · Reviewer_DXVK · 2024-11-01

**Soundness:** 3
**Presentation:** 2
**Contribution:** 3
**Rating:** 6
**Confidence:** 3

**Summary:**

This paper proposes a novel approach for learning optimal decision trees, which is a fundamental problem in interpretable machine learning field. The main idea involves modelling the decision tree learning as a Markov Decision Process (MDP) and solving it using a dynamic programming (DP) and branch & bound (B&B) algorithm. Experimental results demonstrate that the proposed method outperforms the state-of-the-art in terms of efficiency and objective.

**Strengths:**

- The concerned learning optimal decision trees is a fundamental problem and of great importance in the field of interpretable machine learning and even in general machine learning.
- The proposed DP and B&B algorithm is novel and is theoretically justified.
- Experimental results demonstrate the effectiveness of the proposed method, which outperforms the state-of-the-art in terms of efficiency and objective.

**Weaknesses:**

- The modeling of MDP for decision tree learning sounds interesting, but it provides little insights on how the proposal outperforms. Conventional MDPs benefit from the exploration-exploitation trade-off, which is not related to the ODT learning.
- There lacks a formulation of the objective of learning optimal decision trees. The paper should provide a clear definition of the objective and how the proposed method achieves it.

**Questions:**

See Weaknesses.

---

### Official Review · Reviewer_f8hT · 2024-11-02

**Soundness:** 2
**Presentation:** 1
**Contribution:** 3
**Rating:** 3
**Confidence:** 4

**Summary:**

This paper presents novel algorithms for optimizing decision trees, with the goal of enhancing both accuracy and sparsity, where sparsity refers to the tree's depth and number of leaves. Recent progress in decision tree optimization has largely been driven by dynamic programming (DP) and branch-and-bound (B&B) techniques. Current approaches, however, tend to fall into one of two categories:

1. Algorithms like DL8.5, MurTree, and STreeD employ efficient DP methods that make node-level decisions on feature splits. While effective at optimizing decisions within nodes, these methods lack strong bounds to prune the search space effectively.

2. Algorithms such as OSDT and GOSDT, on the other hand, use strong bounds to reduce the search space. However, this comes at the expense of less efficient DP methods to make node-level decisions.

This paper aims to bridge the gap between these two approaches. The proposed algorithm combines an efficient DP strategy, similar to that used in DL8.5, with a more effective pruning bound than those of OSDT and GOSDT.

**Strengths:**

- The high-level intuition behind this approach is appealing, presenting a natural and elegant combination of DP and B&B techniques.
- The empirical results look promising. (I am not an expert in decision tree learning, so I can’t definitively assess whether these are the expected baselines and benchmarks; hopefully another reviewer will be able to speak to this.) The proposed method generally performs well relative to the baselines. In most cases, it either achieves comparable or improved accuracy with a faster runtime or, in instances where it has slower runtime, yields better accuracy.

**Weaknesses:**

Unfortunately, I cannot recommend acceptance of this paper in its current form, as it requires substantial revision to improve clarity and readability.
- Many concepts seem unnecessarily complicated by dense equations without accompanying plain-English explanations. For instance, the first equation in Section 3.2 is hard to interpret without a straightforward description. Adding a plain-English explanation—or even a visual figure—could significantly clarify the concept. Subsequently, the definition of $S(T)$, which appears central to the concept of sparsity, is difficult for me to understand.
- I wonder if using the formalisms of Reinforcement Learning in Section 3.3 is unnecessarily complex and could be simplified. For example, the term "absorbing state" in the context of decision tree construction is hard for me to understand; an example here would be helpful to illustrate the idea.
- The explanation of the main algorithm also needs a rewrite for clarity. In the first sentence of Section 4.2, for example, I can’t tell whether "unvisited" and "incomplete" mean the same thing (i.e., $R^*(l)$ is unknown) or if they refer to distinct states. The "Selection" paragraph suggests they might.
- In the experiments section, I’m not sure what "iterations" refers to in Table 2 for the baseline algorithms and why this metric is not included in Table 3. Could you please define "iterations" clearly in the context of the baselines and explain why this metric is relevant for some comparisons but not others?
- Finally, the paper has numerous grammatical issues, particularly run-on sentences caused by misplaced commas instead of periods. I highly recommend running the text through software like Grammarly to address these errors.

**Questions:**

1. Could you please clarify the confusions I highlighted in the "Weaknesses" section?

---

### Official Review · Reviewer_wPqL · 2024-11-03

**Soundness:** 3
**Presentation:** 3
**Contribution:** 2
**Rating:** 5
**Confidence:** 4

**Summary:**

This paper proposes a reinforcement learning-based branch-and-bound method for optimal sparse trees, utilizing a dynamic programming strategy and a novel pruning bounds for increased efficiency. A theoretical analysis is provided to demonstrate its faster convergence. This method is applicable to datasets with binary features and can be extended to categorical features through encoding.

**Strengths:**

1. This paper integrates a more efficient pruning bound than existing dynamic programming methods and provides a theoretical analysis of the higher efficiency of the proposed algorithm.

2. The authors analyze the drawbacks of binary encoding on efficiency, a common issue for DP-based algorithms.

3. The proposed algorithm achieves similar sparsity and accuracy to GOSDT, which helps validate its correctness and effectiveness.

**Weaknesses:**

1.	In the optimization problems of machine learning models, the primary issue we care about is the testing accuracy, while maintaining interpretability. However, it seems that the proposed method places somewhat excessive emphasis on the regularized objective in the numerical results. As is shown in Table 2 and 3, the authors mainly compare this objective value among all the methods, it is a little subjective and not convincing to demonstrate superiority of proposed method to the compared methods.
2.	When comparing to DL8.5, the configuration of the maximum depth is unfair. To some extent, the overfitting observed in DL8.5 suggests its optimization potential for larger datasets.
3.	It is recommended that the authors use several continuous datasets (considered as those with many categories) to clarify the limitations of the proposed algorithm on continuous features. For example, the Skin Segmentation dataset from UCI could be used to test the scalability limitations and upper performance bounds of the proposed algorithm within a specified time limit (e.g., one day or one week).

**Questions:**

1. The authors analyze the drawbacks in the efficiency of binary encoding for certain methods. However, could this also affect the optimality of decision tree optimization problems? For example, when binary encoding categorical or continuous features, might the optimal solution of the binarized problem be worse than that of the original optimization problem?

2. In Table 6, the proposed algorithm performs similarly to GOSDT which is the most competitive method and sometimes better. Could you explain the primary factors that lead to these differences?

---

### Official Review · Reviewer_dUsF · 2024-11-04

**Soundness:** 3
**Presentation:** 2
**Contribution:** 3
**Rating:** 5
**Confidence:** 3

**Summary:**

The paper introduces a new algorithm for decision tree learning: BRANCHES. This algorithm combines the strengths of existing techniques--Dynamic Programming (DP) and Branch & Bound (B&B)--to improve both the speed and sparsity of decision tree optimization. Unlike traditional greedy approaches that yield suboptimal decision trees, BRANCHES uses an efficient recursive strategy paired with a novel analytical pruning bound called the "Purification Bound", which significantly reduces the search space.

The paper presents theoretical analyses for computational efficiency, with complimenting experimental results as compared to other baseline methods. Results indicate that BRANCHES achieves faster optimal convergence with fewer iterations.

**Strengths:**

Many portions of the paper are written well. The detailed explanation of why exactly we need both Dynamic Programming and Branch & Bound is great, as well as the actual explanation of the algorithm + intuitive explanations of proposition results. Moreover, though I'm not very familiar with the literature on this problem, the experiments seem sensible and demonstrate the improvements of this algorithm.

**Weaknesses:**

The major weakness I note, which I assume the authors can readily rectify, is the comparison of the presented algorithm with other baselines concretely. The abstract, introduction and related work use vague language to describe the improved speed and efficiency, but fail to accurately capture how much of an improvement this algorithm is. As a prominent example, the bullet points of the end of section 1 all state vaguely that "the computational complexity is superior compared to the literature" or "outperforms state of the art methods" but no concrete numbers or theoretical bounds are explicitly provided. These results do appear in the final sections as experimental results, but are provided as non-relative metrics--which makes it hard for an outsider on this problem to parse the magnitude of the improvement.

I encourage the authors to improve the writing in this regard so as to make the paper more approachable for the broader ICLR audience. With these revisions, I would be happy to increase my score.

**Questions:**

Can you state concretely how much of an improvement the method has (analytically and experimentally) as compared to the prior work?

---

### Meta-Review · Area_Chair_Dhvd · 2024-12-18

**Metareview:**

There was no engagement from the authors at rebuttal time and the reviews clearly show that the paper does not pass the bar for acceptance, so I can only encourage the authors to carefully revise their paper based on the reviews provided.

**Additional Comments On Reviewer Discussion:**

N/A

---

### Decision · Program_Chairs · 2025-01-22

Reject